# Exact Mean Square Linear Stability Analysis for SGD

## Abstract

The dynamical stability of optimization methods at the vicinity of minima of the loss has recently attracted significant attention. For gradient descent (GD), stable convergence is possible only to minima that are sufficiently flat w.r.t. the step size, and those have been linked with favorable properties of the trained model. However, while the stability threshold of GD is well-known, to date, no explicit expression has been derived for the exact threshold of stochastic GD (SGD). In this paper, we derive such a closed-form expression. Specifically, we provide an explicit condition on the step size that is both necessary and sufficient for the stability of SGD in the mean square sense. Our analysis sheds light on the precise role of the batch size $B$. Particularly, we show that the stability threshold is a monotonically non-decreasing function of the batch size, which means that reducing the batch size can only decrease stability. Furthermore, we show that SGD's stability threshold is equivalent to that of a process which takes in each iteration a full batch gradient step w.p. $1 - p$, and a single sample gradient step w.p. $p$, where $p \approx 1/B$. This indicates that even with moderate batch sizes, SGD's stability threshold is very close to that of GD's. Finally, we prove simple necessary conditions for stability, which depend on the batch size, and are easier to compute than the precise threshold. We demonstrate our theoretical findings through experiments on the MNIST dataset.

## 1 Introduction

The dynamical stability of optimization methods has been shown to play a key role in shaping the properties of trained models. For instance, gradient descent (GD) can stably converge only to minima that are sufficiently flat with respect to the step size (Cohen *et al.*, 2021), and in the context of neural networks, such minima were shown to correspond to models with favorable properties. These include smoothness of the predictor function (Ma & Ying, 2021; Nacson *et al.*, 2023; Mulayoff *et al.*, 2021), balancedness of the layers (Mulayoff & Michaeli, 2020), and arguably better generalization Hochreiter & Schmidhuber (1997); Keskar *et al.* (2016); Jastrzębski *et al.* (2017); Wu *et al.* (2017); Ma & Ying (2021). While the stability threshold of GD is well-known, that of stochastic GD (SGD) has yet to be fully understood. Several theoretical works studied SGD's dynamics using various notions of stability, including mean square (Wu *et al.*, 2018; Granziol *et al.*, 2022; Velikanov *et al.*, 2023), higher moments (Ma & Ying, 2021), and in probability (Ziyin *et al.*, 2023). However, these works either do not provide explicit stability conditions (*e.g.,* presenting the condition as an optimization problem (Wu *et al.*, 2018; Ma & Ying, 2021) or in terms of a moment generating function (Velikanov *et al.*, 2023)), or rely on strong assumptions (*e.g.,* the nature of the Hessian batching noise and infinite network widths (Granziol *et al.*, 2022), momentum parameter close to 1 and "spectrally expressible" dynamics (Velikanov *et al.*, 2023)). Moreover, several empirical works studied SGD's stability (Cohen *et al.*, 2021; Gilmer *et al.*, 2022; Jastrzębski *et al.*, 2020; 2019), yet its exact stability threshold is still unknown.

In this paper, we analyze the stability of SGD in the mean square sense. We start by considering interpolating minima, which are common in training of overparametrized models. In this case, we provide an explicit threshold on the step size $\eta$ that is both necessary and sufficient for stability. Our analysis sheds light on the precise role of the batch size $B$. Particularly, we show that the maximal step size allowing stable convergence is monotonically non-decreasing in the batch size. Namely, decreasing the batch size can only decrease the stability threshold of SGD. Moreover, we show that this threshold is equivalent to that of a process that takes in each iteration a full batch gradient step

w.p. $1 - p$, and a single sample gradient step w.p. $p$, where $p \approx 1/B$. This suggests that even with moderate batch sizes, SGD's stability threshold is very close to that of GD's. Although our result gives an explicit condition on the step size for stability, its computation may still be challenging in practical applications. Thus, we also prove simple necessary criteria for stability, which depend on the batch size and are easier to compute.

Next, we turn to study a broader class of minima which we call *regular*. Specifically, in interpolating minima, the loss of each individual sample has zero gradient and a positive semi-definite (PSD) Hessian. In regular minima, the individual Hessians are still required to be PSD, but the gradients can be arbitrary. Only the mean of the gradients over all samples has to vanish (as in any minimum). In this setting, the dynamics can wander within the null-space of the Hessian, if the gradients have nonzero components in that subspace. However, the interesting question is whether the process is stable within the orthogonal complement of the null space. Here we again provide an explicit condition on the step size that is both necessary and sufficient for stability. We further derive the theoretical limit of the covariance matrix of the dynamics, as well as the limit values of the expected squared distance to the minimum, the expected loss, and the expected squared norm of the gradient, and show how they all decrease when reducing the learning rate. This provides a theoretical explanation to the behavior encountered in common learning rate scheduling strategies.

Finally, we demonstrate our theoretical findings through experiments on the MNIST dataset (LeCun, 1998). These confirm that our theory correctly predicts the stability threshold of SGD, and its dependence on the batch size.

## 2 BACKGROUND: LINEARIZED DYNAMICS

Let $\ell_i : \mathbb{R}^d \to \mathbb{R}$ be differentiable almost everywhere for all $i \in [n]$. We consider the minimization of a loss function

$$\mathcal{L}(\boldsymbol{\theta}) = \frac{1}{n} \sum_{i=1}^{n} \ell_i(\boldsymbol{\theta}) \tag{1}$$

using the SGD iterations

$$\boldsymbol{\theta}_{t+1} = \boldsymbol{\theta}_t - \eta \nabla \hat{\mathcal{L}}_t(\boldsymbol{\theta}_t). \tag{2}$$

Here, $\eta$ is the step size and $\hat{\mathcal{L}}_t$ is a stochastic approximation of $\mathcal{L}$ obtained as

$$\hat{\mathcal{L}}_t(\boldsymbol{\theta}) = \frac{1}{B} \sum_{i \in \mathfrak{B}_t} \ell_i(\boldsymbol{\theta}), \tag{3}$$

where $\mathfrak{B}_t$ is a batch of size $B$ sampled at iteration $t$. We assume that the batches $\{\mathfrak{B}_t\}$ are drawn uniformly from the dataset, independently across iterations.

Analyzing the full dynamics of this process is intractable in most cases. Yet, near minima, accurate characterization of the stability of the iterates can be obtained via linearization (Wu *et al.*, 2018; Ma & Ying, 2021; Mulayoff *et al.*, 2021), as is common in stability analysis of nonlinear systems.

**Definition 1** (Linearized dynamics). *Let $\boldsymbol{\theta}^*$ be a twice differentiable minimum of $\mathcal{L}$, and denote*

$$\boldsymbol{g}_i \triangleq \nabla \ell_i(\boldsymbol{\theta}^*), \qquad \boldsymbol{H}_i \triangleq \nabla^2 \ell_i(\boldsymbol{\theta}^*). \tag{4}$$

*Then the linearized dynamics of SGD near $\boldsymbol{\theta}^*$ is given by*

$$\boldsymbol{\theta}_{t+1} = \boldsymbol{\theta}_t - \frac{\eta}{B} \sum_{i \in \mathfrak{B}_t} \boldsymbol{H}_i(\boldsymbol{\theta}_t - \boldsymbol{\theta}^*) - \frac{\eta}{B} \sum_{i \in \mathfrak{B}_t} \boldsymbol{g}_i. \tag{5}$$

Note that since $\boldsymbol{\theta}^*$ is a minimum point of $\mathcal{L}$ we have that

$$\nabla \mathcal{L}(\boldsymbol{\theta}^*) = \frac{1}{n} \sum_{i=1}^{n} \boldsymbol{g}_i = \mathbf{0}. \tag{6}$$

Furthermore, the Hessian of the loss, which we denote by $\boldsymbol{H}$, is given by

$$\boldsymbol{H} \triangleq \nabla^2 \mathcal{L}(\boldsymbol{\theta}^*) = \frac{1}{n} \sum_{i=1}^{n} \boldsymbol{H}_i. \tag{7}$$

Thus, the linearized dynamics are in fact SGD iterates on the second-order Taylor expansion of $\mathcal{L}$ at $\boldsymbol{\theta}^*$,

$$\tilde{\mathcal{L}}(\boldsymbol{\theta}) = \mathcal{L}(\boldsymbol{\theta}^*) + \frac{1}{2}(\boldsymbol{\theta} - \boldsymbol{\theta}^*)^{\mathrm{T}} \boldsymbol{H}(\boldsymbol{\theta} - \boldsymbol{\theta}^*). \tag{8}$$

## 3 STABILITY OF FIRST AND SECOND MOMENTS

Our focus is on the stability of SGD's dynamics. We specifically examine the dynamics within two subspaces, the null space of the Hessian $\boldsymbol{H}$ at the minimum, and its orthogonal complement. We denote the projection of any vector $\boldsymbol{v} \in \mathbb{R}^d$ onto the null space of $\boldsymbol{H}$ by $\boldsymbol{v}^{\parallel}$, and its projection onto the orthogonal complement of the null space by $\boldsymbol{v}^{\perp}$.

Multiple works studied the stability of SGD's dynamics. Commonly, this was done by analyzing the evolution of the moments of the linearized dynamics (see Sec. 2) over time, with a specific emphasis on the second moment, which is the approach we take here. However, before discussing the evolution of the second moment, let us summarize the behavior of the first moment. Specifically, it is easy to demonstrate that the first moment of SGD's linearized trajectory $\{\mathbb{E}[\boldsymbol{\theta}_t]\}$ is the same as GD's. Now, since GD is stable if and only if $\eta \le 2/\lambda_{\max}(\boldsymbol{H})$, we have the following (see proof in App. B).

**Theorem 1** (Stability of the mean). *Assume that $\boldsymbol{\theta}^*$ is a twice differentiable minimum. Consider the linear dynamics of $\{\boldsymbol{\theta}_t\}$ from Def. 1 and let*

$$\eta_{\mathrm{mean}}^* \triangleq \frac{2}{\lambda_{\max}(\boldsymbol{H})}. \tag{9}$$

*Then*

1. $\mathbb{E}[\boldsymbol{\theta}_t^{\parallel}] = \mathbb{E}[\boldsymbol{\theta}_0^{\parallel}]$ *for all $t \ge 0$;*
2. $\limsup\limits_{t\to\infty} \|\mathbb{E}[\boldsymbol{\theta}_t] - \boldsymbol{\theta}^*\|$ *is finite if and only if $\eta \le \eta_{\mathrm{mean}}^*$;*
3. $\lim\limits_{t\to\infty} \|\mathbb{E}[\boldsymbol{\theta}_t^{\perp}] - \boldsymbol{\theta}^{*\perp}\| = 0$ *if $\eta < \eta_{\mathrm{mean}}^*$.*

We next proceed to analyze the dynamics of the second moment, which determine stability in the mean square sense. Note that boundedness of the first moment is a necessary condition for boundedness of the second moment. Therefore, the condition $\eta \le \eta_{\mathrm{mean}}^*$ is a prerequisite for stability in the mean square sense. However, how much smaller than $\eta_{\mathrm{mean}}^*$ is SGD's mean square stability threshold, is not currently known in closed form. Here, we determine the precise threshold for the mean square stability of SGD's linearized dynamics. To achieve this, we leverage the approach taken by Ma & Ying (2021), who investigated the stability of SGD in the context of interpolating minima.

### 3.1 INTERPOLATING MINIMA

We begin by studying interpolating minima, which are prevalent in the training of overparametrized models. In this case, the model fits the training set perfectly[1], which means that these global minima are also minima for each sample individually. This is expressed mathematically as follows.

**Definition 2** (Interpolating minima). *A twice differentiable minimum $\boldsymbol{\theta}^*$ is said to be* interpolating *if for each sample $i \in [n]$ the gradient $\boldsymbol{g}_i = \boldsymbol{0}$ and the Hessian $\boldsymbol{H}_i$ is PSD.*

In this setting, Ma & Ying (2021) showed that the evolution of any moment of SGD over time is fully tractable. Specifically, for the second moment, they proved the following.

**Theorem 2** (Ma & Ying (2021), Thm. 1 + Cor. 3). *Assume that $\boldsymbol{\theta}^*$ is a twice differentiable interpolating minimum. Consider the linear dynamics of $\{\boldsymbol{\theta}_t\}$ from Def. 1, and let*

$$\boldsymbol{Q}(\eta, B) \triangleq (\boldsymbol{I} - \eta\boldsymbol{H}) \otimes (\boldsymbol{I} - \eta\boldsymbol{H}) + \frac{n-B}{B(n-1)}\frac{\eta^2}{n}\sum_{i=1}^{n}(\boldsymbol{H}_i \otimes \boldsymbol{H}_i - \boldsymbol{H} \otimes \boldsymbol{H}), \tag{10}$$

*where $\otimes$ denotes the Kronecker product. Then $\limsup\limits_{t\to\infty}\mathbb{E}[\|\boldsymbol{\theta}_t - \boldsymbol{\theta}^*\|^2]$ is finite if and only if*

$$\max_{\boldsymbol{\Sigma} \in \mathcal{S}_+(\mathbb{R}^{d \times d})} \frac{\|\boldsymbol{Q}(\eta, B)\,\mathrm{vec}\,(\boldsymbol{\Sigma})\|}{\|\boldsymbol{\Sigma}\|_{\mathrm{F}}} \le 1, \tag{11}$$

*where $\mathcal{S}_+(\mathbb{R}^{d \times d})$ denotes the set of all PSD matrices over $\mathbb{R}^{d \times d}$. Furthermore, if the spectral radius $\rho(\boldsymbol{Q}(\eta, B)) \le 1$ then $\limsup\limits_{t\to\infty}\mathbb{E}[\|\boldsymbol{\theta}_t - \boldsymbol{\theta}^*\|^2]$ is finite.*

---

[1]The important minima from a practical standpoint are the ones that benign overfit.

Below, we omit the dependence of $\boldsymbol{Q}$ on $\eta$ and $B$ whenever these are not essential for the discussion. In this theorem, $\boldsymbol{\Sigma}$ represents the second-moment matrix of $\boldsymbol{\theta}_t - \boldsymbol{\theta}^*$. Specifically, the matrix $\boldsymbol{\Sigma}_t = \mathbb{E}[(\boldsymbol{\theta}_t - \boldsymbol{\theta}^*)(\boldsymbol{\theta}_t - \boldsymbol{\theta}^*)^{\mathrm{T}}]$ evolves over time as $\mathrm{vec}(\boldsymbol{\Sigma}_{t+1}) = \boldsymbol{Q}\,\mathrm{vec}(\boldsymbol{\Sigma}_t)$. Therefore, the stability condition of (11) simply states that if the dynamics of the dominant initial state of the system (which is restricted to PSD matrices) is bounded, then $\boldsymbol{\Sigma}_t$ is bounded and vice versa. However, this characterization leaves us with a complex optimization problem over a high dimension ($d^2$), which is hard to solve numerically. Therefore, this approach does not reduce the problem into a condition from which we can gain any meaningful theoretical insight into the behavior of SGD.

Our first key result is that the optimization problem (11) can be reduced to an eigenvalue problem. Specifically, we establish (see Sec. 3.3) that when the eigenvectors of the $d^2 \times d^2$ matrix $\boldsymbol{Q}$ are reshaped into $d \times d$ matrices, they always correspond to either symmetric or skew-symmetric matrices[2]. Moreover, the dominant eigenvalue of $\boldsymbol{Q}$ is positive and always corresponds to a PSD matrix. Consequently, the maximizer of (11) is the top eigenvector of $\boldsymbol{Q}$, which we use, along with some algebraic manipulation, to derive the following result (see proof in App. B).

**Theorem 3** (Exact threshold for interpolating minima). *Assume that $\boldsymbol{\theta}^*$ is a twice differentiable interpolating minimum. Consider the linear dynamics of $\{\boldsymbol{\theta}_t\}$ from Def. 1, and let*

$$\boldsymbol{C} \triangleq \frac{1}{2}\boldsymbol{H} \oplus \boldsymbol{H}, \qquad \boldsymbol{D} \triangleq (1-p)\,\boldsymbol{H} \otimes \boldsymbol{H} + p\,\frac{1}{n}\sum_{i=1}^{n}\boldsymbol{H}_i \otimes \boldsymbol{H}_i, \qquad (12)$$

*where $\oplus$ denotes the Kronecker sum and $p \triangleq \frac{n-B}{B(n-1)} \in [0,1]$. Define*

$$\eta_{\mathrm{var}}^* \triangleq \frac{2}{\lambda_{\max}\left(\boldsymbol{C}^\dagger \boldsymbol{D}\right)}, \qquad (13)$$

*where $\boldsymbol{C}^\dagger$ denotes the Moore-Penrose inverse of $\boldsymbol{C}$. Then*

1. *$\boldsymbol{\theta}_t^{\parallel} = \boldsymbol{\theta}_0^{\parallel}$ for all $t \geq 0$;*
2. *$\limsup\limits_{t\to\infty} \mathbb{E}\left[\|\boldsymbol{\theta}_t^{\perp} - \boldsymbol{\theta}^{*\perp}\|^2\right]$ is finite if and only if $\eta \leq \eta_{\mathrm{var}}^*$;*
3. *$\lim\limits_{t\to\infty} \mathbb{E}\left[\|\boldsymbol{\theta}_t^{\perp} - \boldsymbol{\theta}^{*\perp}\|^2\right] = 0$ if $\eta < \eta_{\mathrm{var}}^*$.*

This result provides an explicit characterization of the mean square stability of SGD. First, we observe that the set of step sizes that are stable in the mean square sense, is an interval. This is in contrast to stability in probability, where the stable learning rates can comprise of several disjoint intervals (Ziyin *et al.*, 2023). Moreover, SGD's threshold, $\eta_{\mathrm{var}}^*$, has the same form as the threshold for GD, $2/\lambda_{\max}$, but with a different matrix. In App. I we show how Thm. 3 recovers GD's condition in full batch.

The dependence of $\eta_{\mathrm{var}}^*$ on the batch size $B$ may not be immediate to see from the theorem. However, we can prove the following (see proof in App. D).

**Proposition 1.** *Assume that $\boldsymbol{\theta}^*$ is a twice differentiable interpolating minimum. Then $\eta_{\mathrm{var}}^*$ is a non-decreasing function of $B$.*

This result implies that decreasing the batch size can only decrease the stability threshold, which settles with the empirical observations, *e.g.*, in Wu *et al.* (2018). Additionally, since $\eta_{\mathrm{var}}^*$ is non-decreasing with $B$, and for $B = n$ it equals $\eta_{\mathrm{mean}}^*$, we have that the gap between $\lambda_{\max}(\boldsymbol{C}^\dagger\boldsymbol{D})$ and $\lambda_{\max}(\boldsymbol{H})$ is non-negative for all $B \in [1, n]$ and non-increasing in $B$. For stable minima, $\lambda_{\max}(\boldsymbol{C}^\dagger\boldsymbol{D})$ is bounded from above by $2/\eta$. This suggests that training with smaller batches leads to lower $\lambda_{\max}(\boldsymbol{H})$ which results in smoother predictor functions (Mulayoff *et al.*, 2021).

At what rate does $\eta_{\mathrm{var}}^*$ increase with $B$ towards $\eta_{\mathrm{mean}}^*$? To understand this, note that $\boldsymbol{D}$ is a convex combination of two matrices, where $p$ represents the combination weight. The first matrix is $\boldsymbol{H} \otimes \boldsymbol{H}$, which is associated with full batch SGD ($B = n$), while the second matrix is $\frac{1}{n}\sum_{i=1}^{n}\boldsymbol{H}_i \otimes \boldsymbol{H}_i$, which is related to single sample SGD ($B = 1$). We can use this fact to explain the effect of the batch size on dynamical stability by presenting an equivalent stochastic process that has the same stability threshold as SGD (see proof in App. E).

---

[2]Eigenbases corresponding to eigenvalues of multiplicity greater than one, always have a basis consisting of symmetric and skew-symmetric matrices.

**Proposition 2.** *Let* $\mathtt{ALG}(p)$ *be a stochastic optimization algorithm in which*

$$\boldsymbol{\theta}_{t+1} = \begin{cases} \boldsymbol{\theta}_t - \eta \nabla \ell_{i_t}(\boldsymbol{\theta}_t) & w.p. \quad p, \\ \boldsymbol{\theta}_t - \eta \nabla \mathcal{L}(\boldsymbol{\theta}_t) & w.p. \quad 1-p, \end{cases} \tag{14}$$

*where* $\{i_t\}$ *are i.i.d. random indices distributed uniformly over the training set. Assume that* $\boldsymbol{\theta}^*$ *is a twice differentiable interpolating minimum. Then when* $p = \frac{n-B}{B(n-1)}$, $\mathtt{ALG}(p)$ *has the same stability threshold in the vicinity of* $\boldsymbol{\theta}^*$ *as SGD with batch size* $B$.

In simpler terms, in each iteration the process $\mathtt{ALG}(p)$ takes a gradient step with a batch of one sample ($B = 1$) with probability $p$ and with a full batch ($B = n$) with probability $1 - p$. This result shows that the stability conditions of SGD and of $\mathtt{ALG}(p)$ are the same for $p = \frac{n-B}{B(n-1)}$. When $n \gg B$, we have that $p \approx 1/B$. Therefore, Prop. 2 implies that, in the context of stability, even moderate values of $B$ make mini-batch SGD behave like GD. We note that while propositions 1 and 2 were presented in the context of interpolating minima, they also apply to regular minima (see Sec. 3.2).

It is worthwhile mentioning that if the stability condition is not met, then the linearized dynamics diverge. However, in practice, the full (non-linearized) dynamics can just move to a different point on the loss landscape, where the generalized sharpness $\lambda_{\max}(\boldsymbol{C}^\dagger \boldsymbol{D})$ is lower. It was shown that GD possesses such a stabilizing mechanism (Damian *et al.*, 2023). An interesting open question is whether a similar mechanism exists in SGD.

Theorem 3 gives an explicit condition on the step size. However, its computation may still be challenging in practical applications, as it requires inverting, multiplying, and computing the spectral norm of large ($d^2 \times d^2$) matrices. Yet, we can obtain necessary criteria for stability that are simple and easier to verify, and which also depend on the batch size. To do so, we note that the eigenvalues of $\boldsymbol{C}^\dagger \boldsymbol{D}$ coincide with those of $(\boldsymbol{C}^\dagger)^{\frac{1}{2}} \boldsymbol{D} (\boldsymbol{C}^\dagger)^{\frac{1}{2}}$. We therefore upper bound $\eta_{\text{var}}^*$ by evaluating $\boldsymbol{u}^{\mathrm{T}}((\boldsymbol{C}^\dagger)^{\frac{1}{2}} \boldsymbol{D} (\boldsymbol{C}^\dagger)^{\frac{1}{2}}) \boldsymbol{u}$ for interesting directions $\boldsymbol{u}$ other than the top eigenvector. Specifically, the next result corresponds to $\boldsymbol{u} = \boldsymbol{C}^{\frac{1}{2}} \text{vec}(\boldsymbol{I})/\|\boldsymbol{C}^{\frac{1}{2}} \text{vec}(\boldsymbol{I})\|$ and $\boldsymbol{u} = \boldsymbol{C}^{\frac{1}{2}} (\boldsymbol{v}_{\max} \otimes \boldsymbol{v}_{\max})/\|\boldsymbol{C}^{\frac{1}{2}} (\boldsymbol{v}_{\max} \otimes \boldsymbol{v}_{\max})\|$, where $\boldsymbol{v}_{\max}$ is the top eigenvector of $\boldsymbol{H}$ (see proof in App. F).

**Proposition 3** (Necessary conditions). *The step size* $\eta_{\text{var}}^*$ *satisfies*

$$\eta_{\text{var}}^* \leq \frac{2\lambda_{\max}(\boldsymbol{H})}{\lambda_{\max}^2(\boldsymbol{H}) + \frac{p}{n} \sum_{i=1}^n (\boldsymbol{v}_{\max}^{\mathrm{T}} \boldsymbol{H}_i \boldsymbol{v}_{\max} - \lambda_{\max}(\boldsymbol{H}))^2}, \tag{15}$$

*as well as*

$$\eta_{\text{var}}^* \leq \frac{2\text{Tr}(\boldsymbol{H})}{(1-p)\|\boldsymbol{H}\|_{\mathrm{F}}^2 + \frac{p}{n} \sum_{i=1}^n \|\boldsymbol{H}_i\|_{\mathrm{F}}^2}. \tag{16}$$

From (15), we can deduce a lower bound on the gap between the stability thresholds of GD and SGD. Specifically, when the variance of $\boldsymbol{H}_i$ over the direction of the top eigenvector of $\boldsymbol{H}$ is large, $\eta_{\text{var}}^*$ is far from $\eta_{\text{mean}}^*$ for moderate $p$. In general, this condition is expected to be quite tight when there is a clear dominant direction in $\boldsymbol{H}$ caused by some $\boldsymbol{H}_i$. In contrast, condition (16) is expected to be tight if all $\{\boldsymbol{H}_i\}$ have roughly the same spectrum but with different bases, *i.e.,* when no sample is dominant and the samples are incoherent.

## 3.2 NON-INTERPOLATING MINIMA

While for interpolating minima, we saw that $\boldsymbol{\theta}_t^\perp$ can converge to $\boldsymbol{\theta}^{*\perp}$, this is generally not the case for non-interpolating minima. In this section, we explore the dynamics of SGD in the vicinity of a broader class of minima. Particularly, we consider the following definition.

**Definition 3** (Regular minima). *A twice differentiable minimum* $\boldsymbol{\theta}^*$ *is said to be* regular *if for each sample* $i \in [n]$ *the Hessian* $\boldsymbol{H}_i$ *is PSD.*

This definition encompasses a broader class of minima than Def. 2, as it allows for arbitrary (nonzero) gradients $\boldsymbol{g}_i$. Only the mean of the gradients has to vanish (as in any minimum). Intuitively speaking, although a regular minimum does not necessarily fit all the training points, it does not involve a major disagreement among them. This can be understood through the second-order Taylor expansion for each sample, which may be unbounded from below, yet it can only go to minus infinity linearly with the parameters, and not quadratically.

Clearly, having gradients with nonzero components in the null space of the Hessian pushes the dynamics to diverge. Interestingly, for regular minima, the dynamics of SGD in the null space and in its orthogonal complement are separable. Thus, despite having a random walk in the null space, we can give a condition for stability within its orthogonal complement (see proof in App. B).

**Theorem 4** (Exact threshold for regular minima). *Assume that $\boldsymbol{\theta}^*$ is a twice differentiable regular minimum. Consider the linear dynamics of $\{\boldsymbol{\theta}_t\}$ from Def. 1. Then*

1. $\lim_{t\to\infty} \mathbb{E}\left[\|\boldsymbol{\theta}_t^\parallel - \boldsymbol{\theta}^{*\parallel}\|^2\right] = \infty$ *if and only if* $\sum_{i=1}^n \|\boldsymbol{g}_i^\parallel\|^2 > 0$;

2. *If* $\eta < \eta_{\text{var}}^*$ *then* $\limsup_{t\to\infty} \mathbb{E}\left[\|\boldsymbol{\theta}_t^\perp - \boldsymbol{\theta}^{*\perp}\|^2\right]$ *is finite;*

3. *If* $\limsup_{t\to\infty} \mathbb{E}\left[\|\boldsymbol{\theta}_t^\perp - \boldsymbol{\theta}^{*\perp}\|^2\right]$ *is finite then* $\eta \leq \eta_{\text{var}}^*$.

We see that $\eta_{\text{var}}^*$ is the stability threshold also for regular minima. Recall that when $\eta < \eta_{\text{var}}^*$, we also have stability of the first moment, and thus $\mathbb{E}[\boldsymbol{\theta}_t^\parallel] = \mathbb{E}[\boldsymbol{\theta}_0^\parallel]$ for any $t \geq 0$. Namely, SGD's dynamics in the null space is a random walk without drift. Note that moving in the null space does not increase the loss, however it might change the trained model. Furthermore, in the proof, we show that under a mild assumption $\limsup_{t\to\infty}\mathbb{E}[\|\boldsymbol{\theta}_t^\perp - \boldsymbol{\theta}^{*\perp}\|^2]$ is finite if and only if $0 \leq \eta < \eta_{\text{var}}^*$.

Next, we turn to compute the limit of the second moment of the dynamics (see proof in App. G).

**Theorem 5** (Covariance limit). *Assume that $\boldsymbol{\theta}^*$ is a twice differentiable regular minimum. Consider the linear dynamics of $\{\boldsymbol{\theta}_t\}$ from Def. 1. If $0 < \eta < \eta_{\text{var}}^*$ then*

$$\lim_{t\to\infty} \text{vec}\left(\boldsymbol{\Sigma}_t^\perp\right) = \eta p \left(2\boldsymbol{C} - \eta\boldsymbol{D}\right)^\dagger \text{vec}\left(\boldsymbol{\Sigma}_{\boldsymbol{g}}^\perp\right), \tag{17}$$

*where $\boldsymbol{\Sigma}_{\boldsymbol{g}}^\perp = \frac{1}{n}\sum_{i=1}^n \boldsymbol{g}_i^\perp \left(\boldsymbol{g}_i^\perp\right)^{\text{T}}$.*

Using this result we can obtain the mean squared distance to the minimum, the mean of the second-order Taylor expansion of the loss, and the mean of the squared norm of the expansion's gradient squared at large times (see proof in App. H).

**Corollary 1** (Limit values). *Assume that $\boldsymbol{\theta}^*$ is a twice differentiable regular minimum. Consider the linear dynamics of $\{\boldsymbol{\theta}_t\}$ from Def. 1 and the second-order Taylor expansion of the loss, $\tilde{\mathcal{L}}$ of (8). If $\eta < \eta_{\text{var}}^*$ then*

1. $\lim_{t\to\infty} \mathbb{E}\left[\|\boldsymbol{\theta}_t^\perp - \boldsymbol{\theta}^{*\perp}\|^2\right] = \eta p(\text{vec}\left(\boldsymbol{I}\right))^{\text{T}}\left(2\boldsymbol{C} - \eta\boldsymbol{D}\right)^\dagger \text{vec}\left(\boldsymbol{\Sigma}_{\boldsymbol{g}}^\perp\right)$;

2. $\lim_{t\to\infty} \mathbb{E}\left[\tilde{\mathcal{L}}(\boldsymbol{\theta}_t)\right] - \tilde{\mathcal{L}}(\boldsymbol{\theta}^*) = \frac{1}{2}\eta p(\text{vec}\left(\boldsymbol{H}\right))^{\text{T}}\left(2\boldsymbol{C} - \eta\boldsymbol{D}\right)^\dagger \text{vec}\left(\boldsymbol{\Sigma}_{\boldsymbol{g}}^\perp\right)$;

3. $\lim_{t\to\infty} \mathbb{E}\left[\|\nabla\tilde{\mathcal{L}}(\boldsymbol{\theta}_t)\|^2\right] = \eta p \left(\text{vec}\left(\boldsymbol{H}^2\right)\right)^{\text{T}}\left(2\boldsymbol{C} - \eta\boldsymbol{D}\right)^\dagger \text{vec}\left(\boldsymbol{\Sigma}_{\boldsymbol{g}}^\perp\right)$.

We see that these values depend linearly on the covariance matrix of the gradients. Specifically, if $\boldsymbol{\Sigma}_{\boldsymbol{g}} = \boldsymbol{0}$ then we recover the results of interpolating minima. Moreover, note that for $\eta \ll \eta_{\text{var}}^*$, we have that $2\boldsymbol{C} - \eta\boldsymbol{D} \approx 2\boldsymbol{C}$. Therefore, the main dependence on $\eta$ comes from the factor of $\eta$ preceding these expressions. We thus get that when decreasing the learning rate, the loss level drops, and the parameters $\boldsymbol{\theta}_t$ get closer to the minimum. This explains the empirical behavior observed when decreasing the learning rate in neural network training, which causes the loss level to drop.

## 3.3 Proof Outline for Theorem 3

Here we give an outline of the proof of Theorem 3. Ma & Ying (2021) showed that the second moment matrix $\boldsymbol{\Sigma}_t = \mathbb{E}\left[(\boldsymbol{\theta}_t - \boldsymbol{\theta}^*)(\boldsymbol{\theta}_t - \boldsymbol{\theta}^*)^{\text{T}}\right]$ evolves over time as

$$\text{vec}\left(\boldsymbol{\Sigma}_{t+1}\right) = \boldsymbol{Q}\,\text{vec}\left(\boldsymbol{\Sigma}_t\right), \tag{18}$$

where $\boldsymbol{Q}$ is given in (10). Since $\boldsymbol{\Sigma}_t$ is PSD by definition, we only care about the effect of $\boldsymbol{Q}$ on vectorizations of PSD matrices. Hence, $\{\boldsymbol{\Sigma}_t\}$ are bounded if and only if (proof in Ma & Ying (2021))

$$\max_{\boldsymbol{\Sigma}\in\mathcal{S}_+(\mathbb{R}^{d\times d})} \frac{\|\boldsymbol{Q}(\eta, B)\,\text{vec}\left(\boldsymbol{\Sigma}\right)\|}{\|\boldsymbol{\Sigma}\|_{\text{F}}} \leq 1. \tag{19}$$

To obtain the result of Thm. 3 we first rearrange the terms in $\boldsymbol{Q}$ as

$$\boldsymbol{Q}(\eta, B) = (1 - p) \times (\boldsymbol{I} - \eta\boldsymbol{H}) \otimes (\boldsymbol{I} - \eta\boldsymbol{H}) + p \times \frac{1}{n}\sum_{i=1}^n (\boldsymbol{I} - \eta\boldsymbol{H}_i) \otimes (\boldsymbol{I} - \eta\boldsymbol{H}_i). \tag{20}$$

Then, to relax the optimization problem we use the following theorem (see proof in App. C).

**Theorem 6.** *Assume $\{\boldsymbol{A}_i\}$ are symmetric matrices over $\mathbb{R}^{d \times d}$. Define*

$$\boldsymbol{Q} = \sum_{i=1}^{M} \boldsymbol{A}_i \otimes \boldsymbol{A}_i, \tag{21}$$

*and let $\boldsymbol{z}_{\max}$ be a top eigenvector of $\boldsymbol{Q}$. Then*

1. *there always exists a set of eigenvectors $\{\boldsymbol{z}_j\}$ for $\boldsymbol{Q}$ such that each $\boldsymbol{Z}_j = \mathrm{vec}^{-1}(\boldsymbol{z}_j)$ is either a symmetric or a skew-symmetric matrix;*

2. *the spectral radius $\rho(\boldsymbol{Q}) = \lambda_{\max}(\boldsymbol{Q})$, i.e., the dominant eigenvalue is positive;*

3. *$\mathrm{vec}^{-1}(\boldsymbol{z}_{\max}) \in \mathcal{S}_+(\mathbb{R}^{d \times d})$, i.e., the top eigenvector corresponds to a PSD matrix.*

Applying this theorem we get that the maximizer for the constrained optimization problem in (19) is, in fact, the top eigenvalue of $\boldsymbol{Q}$. Hence, the linear system is stable if and only if $\lambda_{\max}(\boldsymbol{Q}) \leq 1$. Since $\boldsymbol{Q}$ is symmetric, $\lambda_{\max}(\boldsymbol{Q}) \leq 1$ is equivalent to $\boldsymbol{u}^{\mathrm{T}} \boldsymbol{Q} \boldsymbol{u} \leq 1$ for all $\boldsymbol{u} \in \mathbb{S}^{d^2-1}$. It is easy to show that $\boldsymbol{Q} = \boldsymbol{I} - 2\eta \boldsymbol{C} + \eta^2 \boldsymbol{D}$. In App. B we show that

$$\boldsymbol{u}^{\mathrm{T}} \boldsymbol{Q} \boldsymbol{u} = 1 - 2\eta \boldsymbol{u}^{\mathrm{T}} \boldsymbol{C} \boldsymbol{u} + \eta^2 \boldsymbol{u}^{\mathrm{T}} \boldsymbol{D} \boldsymbol{u} \leq 1 \tag{22}$$

holds for all $\boldsymbol{u} \in \mathbb{S}^{d^2-1}$ if and only if

$$\eta \leq \frac{2}{\lambda_{\max}\left(\boldsymbol{C}^{\dagger} \boldsymbol{D}\right)} = \eta_{\mathrm{var}}^*. \tag{23}$$

The full proof is provided in App. B.

## 4 EXPERIMENTS

In this section, we experimentally validate our theoretical results. We trained single hidden-layer ReLU networks with varying step sizes and batch sizes on a subset of the MNIST dataset (see App. J for details). Since training with cross-entropy and softmax in overparametrized networks results in infima rather than minima, here we opted to use the quadratic loss. Specifically, each class was labeled with a one-hot vector, and the network was trained to predict the label without softmax. Our primary goal in this experiment is to test the stability threshold of SGD; hence, we initialized the training with large weights to ensure that the minimum closest to the starting point is unstable (large weights imply large Hessians, and are thus more likely to violate the stability criterion). We used the same initial point for all the training runs to eliminate initialization effects. To avoid divergence, we started with a very small learning rate and gradually increased it until it reached its designated value (*i.e.,* LR warm-up). Together, large initialization and warm-up force SGD out of the unstable region until it finds a stable minimum and converges as closely as possible at the stability threshold. Convergence was determined when the loss remained below $10^{-6}$ for 200 consecutive epochs.

Figure 1(a) visualizes the sharpness of the converged minima versus the learning rate for several values of $B$. It can be observed that for small batch sizes, $\lambda_{\max}(\boldsymbol{H})$ is far from $2/\eta$. Yet, for moderate batch sizes and above (*e.g.,* $B \geq 32$), these curves virtually coincide, indicating that, in the context of stability, SGD behaves like GD. Figure 1(b) shows the sharpness versus the batch size for three step sizes. Here the stability threshold of SGD rapidly converges to that of GD as the batch size increases.

Apart for the sharpness $\lambda_{\max}(\boldsymbol{H})$, we also want to compare the generalized sharpness $\lambda_{\max}(\boldsymbol{C}^{\dagger} \boldsymbol{D})$ to $2/\eta$. Since computing the generalized sharpness is impractical in this task, we underestimate it via a lower bound, which results in a tighter necessary condition than (15). The bound corresponds to restricting the optimization problem in (11) to rank one PSD matrices, and is given by (see App. F.1)

$$\frac{2}{\eta_{\mathrm{var}}^*} = \lambda_{\max}\left(\boldsymbol{C}^{\dagger} \boldsymbol{D}\right) \geq \max_{\boldsymbol{v}:\|\boldsymbol{v}\|=1} \left\{ \boldsymbol{v}^{\mathrm{T}} \boldsymbol{H} \boldsymbol{v} + p \frac{\frac{1}{n}\sum_{i=1}^{n}(\boldsymbol{v}^{\mathrm{T}} \boldsymbol{H}_i \boldsymbol{v} - \boldsymbol{v}^{\mathrm{T}} \boldsymbol{H} \boldsymbol{v})^2}{\boldsymbol{v}^{\mathrm{T}} \boldsymbol{H} \boldsymbol{v}} \right\}. \tag{24}$$

We solve this optimization problem numerically, by using GD on the unit sphere with predetermined scheduled geodesic step size. In the following, we present graphs of the sharpness $\lambda_{\max}(\boldsymbol{H})$ at the minima to which we converged, as well as the bounds (24) and (15) on the generalized sharpness

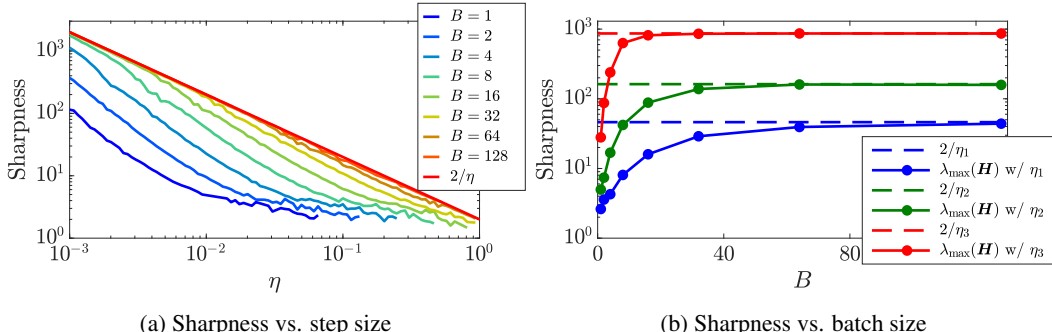

(a) Sharpness vs. step size

(b) Sharpness vs. batch size

Figure 1: **Sharpness vs. step size and batch size.** We trained single hidden-layer ReLU networks using varying step sizes and batch sizes on a subset of MNIST. Panel (a) visualizes the sharpness of the converged minima versus learning rate for different batch sizes. For small batch sizes, $\lambda_{\max}(\boldsymbol{H})$ deviates significantly from $2/\eta$. Yet, as the batch size increases to a moderate value, these curves coincide, indicating that in terms of stability, SGD behaves similarly to GD. Panel (b) plots the sharpness against the batch size for three different learning rates $\eta_1 = 0.043, \eta_2 = 0.012, \eta_3 = 0.002$. Here we see a similar trend where SGD with behaves like GD for $B \geq 32$.

$\lambda_{\max}(\boldsymbol{C}^\dagger\boldsymbol{D})$. Using the color coding of Fig. 2, these correspond to

$$
\begin{aligned}
\frac{2}{\eta} &\geq \lambda_{\max}\left(\boldsymbol{C}^\dagger\boldsymbol{D}\right) \quad \left(= \frac{2}{\eta^*_{\text{var}}}\right) \\
&\geq \max_{\boldsymbol{v}:\|\boldsymbol{v}\|=1}\left\{\boldsymbol{v}^{\mathrm{T}}\boldsymbol{H}\boldsymbol{v} + p\frac{\frac{1}{n}\sum_{i=1}^{n}(\boldsymbol{v}^{\mathrm{T}}\boldsymbol{H}_i\boldsymbol{v} - \boldsymbol{v}^{\mathrm{T}}\boldsymbol{H}\boldsymbol{v})^2}{\boldsymbol{v}^{\mathrm{T}}\boldsymbol{H}\boldsymbol{v}}\right\} \\
&\geq \lambda_{\max}(\boldsymbol{H}) + p\frac{\frac{1}{n}\sum_{i=1}^{n}(\boldsymbol{v}_{\max}^{\mathrm{T}}\boldsymbol{H}_i\boldsymbol{v}_{\max} - \lambda_{\max}(\boldsymbol{H}))^2}{\lambda_{\max}(\boldsymbol{H})} \\
&\geq \lambda_{\max}(\boldsymbol{H}),
\end{aligned}
\tag{25}
$$

where $\boldsymbol{v}_{\max}$ denotes the top eigenvector of $\boldsymbol{H}$.

Figure 2 depicts the expressions in (25) versus the step size for three batch sizes. We see that for $B = 1$ and $B = 2$, the gap between $2/\eta$ (red) and the optimized bound (24) (purple) upon convergence is small. Particularly, they coincide over a wide range of step sizes $\eta$. Since the generalized sharpness $\lambda_{\max}(\boldsymbol{C}^\dagger\boldsymbol{D})$ must reside between those two curves, we can deduce two things: (a) Our theory correctly predicts the stability threshold, while SGD converged at the edge of stability (as designed in our experiment); (b) For small batches, the second order moment matrix that maximizes (11) is rank one. As the batch size increases, the two curves draw apart, indicating that the rank of the dominant second-order moment matrix becomes larger. Furthermore, the gap between our simple necessary condition (15) (blue) and the trivial bound of $2/\lambda_{\max}(\boldsymbol{H})$ (yellow) is large for high learning rates and small for small step sizes. This gap represents the variance of the widths of the minima of the per-sample losses (corresponding to the widths of the quadratic functions $\{(\boldsymbol{\theta} - \boldsymbol{\theta}^*)^{\mathrm{T}}\boldsymbol{H}_i(\boldsymbol{\theta} - \boldsymbol{\theta}^*)\}$) in the direction of $\boldsymbol{v}_{\max}$, the top eigenvector of $\boldsymbol{H}$. Thus we find that for small learning rates, this variance is small and the model is aligned in this direction, and for large learning rates, this variance is high. For more details and experimental results, please see App. J.

## 5    RELATED WORK

The stability of SGD in the vicinity of minima has been previously studied in multiple works. On the theoretical side, Wu *et al.* (2018) examined stability in the mean square sense and gave an implicit sufficient condition. Granziol *et al.* (2022) used random matrix theory to find the maximal stable learning rate as a function of the batch size. Their work assumes some conditions on the Hessian's noise caused by batching, and the result holds in the limit of an infinite number of samples and batch size. Velikanov *et al.* (2023) examined SGD with momentum and derived a bound on the maximal learning rate. Their derivation uses "spectrally expressible" approximations and the result is

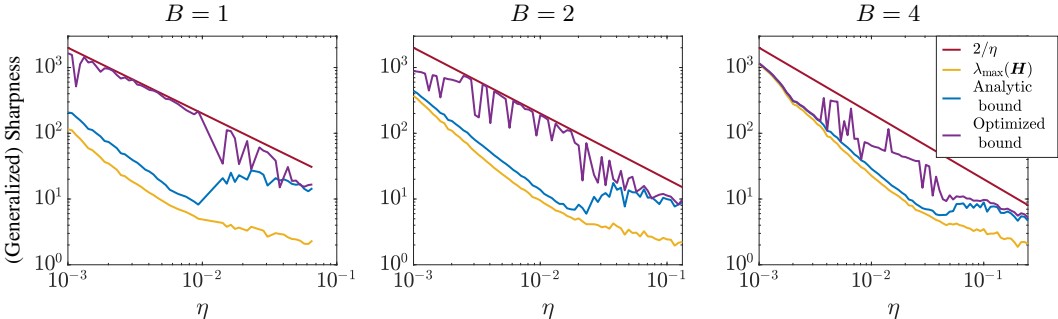

Figure 2: **(Generalized) Sharpness vs. step size.** We trained single hidden-layer ReLU networks using varying step sizes and batch sizes on MNIST dataset. For each pair of hyper-parameters $(\eta, B)$, we measured the sharpness of the minimum (yellow), our necessary condition for stability (blue), and the optimized bound (purple), which their relations are given in (25). We see that for small batch sizes $B = 1$ and $B = 2$, the optimized bound (24) coincides with $2/\eta$, confirming that SGD converged at the edge of stability ($\eta = \eta_{\mathrm{var}}^*$). For additional insights and detail, see discussion in Sec. 4.

given implicitly through a moment-generating function. Ma & Ying (2021) studied the dynamics of higher moments of SGD and gave an implicit necessary and sufficient condition for stability (see Thm. 2 and the discussion following it). Wu *et al.* (2022) gave a necessary condition for stability via alignment property. However, the result assumes and uses a lower bound on a property they coin "alignment" but an analytic bound for this alignment property is lacking for the general case. Ziyin *et al.* (2023) studied the stability of SGD in probability, rather than in mean square. Since convergence in probability is a weaker requirement, theoretically, SGD can converge with high probability to minima which are unstable in the mean square sense. Indeed, their theory predicts that SGD can converge far beyond GD's threshold. Yet, this did not happen in extensive experiments done in (Cohen *et al.*, 2021, App. G) and (Gilmer *et al.*, 2022). Finally, Mulayoff *et al.* (2021) analyzed the stability in non-differentiable minima, and gave a necessary condition for a minimum to be "strongly stable", *i.e.,* such that SGD does not escape a ball with a given radius from the minimum.

Liu *et al.* (2021) studied the covariance matrix of the stationary distribution of the iterates in the vicinity of minima. Ziyin *et al.* (2022) improved their results while deriving an implicit equation that relates this covariance to the covariance of the gradient noise. However, these papers do not discuss the conditions under which the dynamics converge to the stationary state. Lee & Jang (2023) studied the stability of SGD along its trajectory and gave an explicit exact condition. Yet, their result does not apply to minima, since the denominator in their condition vanishes at minima.

On the empirical side, Cohen *et al.* (2021) examined the behavior of GD, and showed that it typically converges at the edge of stability. Additionally, for SGD (see their App. G) they found that with large batches, the sharpness behaves similarly to full-batch gradient descent. Moreover, they found that the smaller the batch size, the lower the sharpness at the converged minimum. Gilmer *et al.* (2022) studied how the curvature of the loss affects the training dynamics in multiple settings. They observed that SGD *with momentum* is stable only when the optimization trajectory primarily resides in a region of parameter space where $\eta \lesssim 2/\lambda_{\max}(\boldsymbol{H})$. Further experimental results in Jastrzębski *et al.* (2020; 2019) show that the sharpness along the trajectory of SGD is implicitly regularized.

## 6 CONCLUSION

We presented an explicit threshold on SGD's step size, which is both necessary and sufficient for guaranteeing mean-square stability. We showed that this threshold is a monotonically non-decreasing function of the batch size, which implies that decreasing the batch size can only make the process less stable. Additionally, we interpreted the role of the batch size $B$ through an equivalent process that takes in each iteration either a full batch gradient step or a single sample gradient step. Our interpretation highlights that even with moderate batch sizes, SGD's stability threshold is very close to that of GD. We also proved simpler necessary conditions for stability, which depend on the batch size, and are easier to compute. Finally, we verified our theory through experiments on MNIST.

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

# A    NOTATIONS AND THE KRONECKER PRODUCT

Throughout our derivations, we use the following notations.

| | |
|---|---|
| $a$ | Lower case non-bold letters for scalars |
| $\boldsymbol{a}$ | Lower case bold letters for vectors |
| $\boldsymbol{A}$ | Upper case bold for matrices |
| $\boldsymbol{A}^\dagger$ | Moore–Penrose inverse of $\boldsymbol{A}$ |
| $\boldsymbol{P}_\mathcal{V}$ | Projection matrix onto the subspace $\mathcal{V}$ |
| $\mathcal{N}(\boldsymbol{A})$ | Null space of $\boldsymbol{A}$ |
| $\mathcal{N}^\perp(\boldsymbol{A})$ | Orthogonal complement of the null space of $\boldsymbol{A}$ |
| $\mathcal{R}(\boldsymbol{A})$ | Range of $\boldsymbol{A}$ |
| $\mathcal{R}^\perp(\boldsymbol{A})$ | Orthogonal complement of the range of $\boldsymbol{A}$ |
| $\otimes$ | Kronecker product |
| $\oplus$ | Kronecker sum |
| $\odot k$ | $k$'th Hadamard power |
| $\mathbb{E}$ | Expectation |
| $\mathbb{P}$ | Probability |
| $\|\boldsymbol{a}\|$ | Euclidean norm of $\boldsymbol{a}$ |
| $\|\boldsymbol{A}\|$ | Top singular value of $\boldsymbol{A}$ |
| $\|\boldsymbol{A}\|_\mathrm{F}$ | Frobenius norm of $\boldsymbol{A}$ |
| $\rho(\boldsymbol{A})$ | Spectral radius of $\boldsymbol{A}$ |
| $\mathrm{vec}(\boldsymbol{A})$ | Vectorization of $\boldsymbol{A}$ (column stack) |
| $\mathrm{vec}^{-1}(\boldsymbol{a})$ | Reshaping $\boldsymbol{a}$ back to $d \times d$ matrix |
| $\mathcal{L}$ | Loss function |
| $\boldsymbol{\theta}$ | Parameters vector of the loss |
| $\boldsymbol{\theta}^*$ | Minimum point of the loss |
| $d$ | Dimension of $\boldsymbol{\theta}$ |
| $n$ | Number of training samples |
| $\eta$ | Step size |
| $B$ | Batch size |
| $p$ | Defined to be $(n-B)/\big(B(n-1)\big)$ |
| $\boldsymbol{\Sigma}$ | Second moment matrix |
| $\boldsymbol{H}$ | Hessian of the full loss at $\boldsymbol{\theta}^*$ |
| $\boldsymbol{H}_i$ | Hessian of the loss of the sample $i$ at $\boldsymbol{\theta}^*$ |
| $\boldsymbol{g}_i$ | Gradient of the loss of the sample $i$ at $\boldsymbol{\theta}^*$ |
| $\boldsymbol{a}^\|$ | Projection of $\boldsymbol{a}$ onto the null space of $\boldsymbol{H}$ |
| $\boldsymbol{a}^\perp$ | Projection of $\boldsymbol{a}$ onto the orthogonal complement of the null space of $\boldsymbol{H}$ |
| $\mathcal{S}_+\big(\mathbb{R}^{d\times d}\big)$ | The set of all positive semi-definite (PSD) matrices over $\mathbb{R}^{d\times d}$ |
| $\mathbb{S}^{d-1}$ | Unit sphere in $\mathbb{R}^d$ |

Table 1: Table of notations

Further notations that we use are given below.

$$
\boldsymbol{\mu}_t \triangleq \mathbb{E}\left[\boldsymbol{\theta}_t - \boldsymbol{\theta}^*\right], \qquad \boldsymbol{\Sigma}_t \triangleq \mathbb{E}\left[(\boldsymbol{\theta}_t - \boldsymbol{\theta}^*)(\boldsymbol{\theta}_t - \boldsymbol{\theta}^*)^\mathrm{T}\right],
$$

$$
\boldsymbol{\mu}_t^\perp \triangleq \mathbb{E}\left[\boldsymbol{\theta}_t^\perp - \boldsymbol{\theta}^{*\perp}\right], \qquad \boldsymbol{\Sigma}_t^\perp \triangleq \mathbb{E}\left[(\boldsymbol{\theta}_t^\perp - \boldsymbol{\theta}^{*\perp})(\boldsymbol{\theta}_t^\perp - \boldsymbol{\theta}^{*\perp})^\mathrm{T}\right],
$$

$$
\boldsymbol{\mu}_t^\| \triangleq \mathbb{E}\left[\boldsymbol{\theta}_t^\| - \boldsymbol{\theta}^{*\|}\right], \qquad \boldsymbol{\Sigma}_t^\| \triangleq \mathbb{E}\left[(\boldsymbol{\theta}_t^\| - \boldsymbol{\theta}^{*\|})(\boldsymbol{\theta}_t^\| - \boldsymbol{\theta}^{*\|})^\mathrm{T}\right]. \tag{26}
$$

Additionally, we make extensive use of the following properties of the Kronecker product. For any matrices $\boldsymbol{M}_1, \boldsymbol{M}_2, \boldsymbol{M}_3, \boldsymbol{M}_4$,

$$\text{vec}\left(\boldsymbol{M}_1 \boldsymbol{M}_2 \boldsymbol{M}_3\right) = \left(\boldsymbol{M}_3^T \otimes \boldsymbol{M}_1\right)\text{vec}\left(\boldsymbol{M}_2\right), \tag{P1}$$

$$\left(\boldsymbol{M}_1 \otimes \boldsymbol{M}_2\right)^T = \left(\boldsymbol{M}_1^T \otimes \boldsymbol{M}_2^T\right), \tag{P2}$$

$$\left(\boldsymbol{M}_1 \otimes \boldsymbol{M}_2\right)\left(\boldsymbol{M}_3 \otimes \boldsymbol{M}_4\right) = \left(\boldsymbol{M}_1 \boldsymbol{M}_3\right) \otimes \left(\boldsymbol{M}_2 \boldsymbol{M}_4\right), \tag{P3}$$

$$\left[\text{vec}\left(\boldsymbol{M}_1\right)\right]^{\text{T}}\left(\boldsymbol{M}_2 \otimes \boldsymbol{M}_3\right)\text{vec}\left(\boldsymbol{M}_4\right) = \text{Tr}\left(\boldsymbol{M}_1^{\text{T}} \boldsymbol{M}_3 \boldsymbol{M}_4 \boldsymbol{M}_2^{\text{T}}\right). \tag{P4}$$

Finally, we give here the definition of Kronecker sum. If $\boldsymbol{M}_1$ is $d_1 \times d_1$, $\boldsymbol{M}_2$ is $d_2 \times d_2$ and $\boldsymbol{I}_d$ denotes the $d \times d$ identity matrix then

$$\boldsymbol{M}_1 \oplus \boldsymbol{M}_1 = \boldsymbol{M}_1 \otimes \boldsymbol{I}_{d_2} + \boldsymbol{I}_{d_1} \otimes \boldsymbol{M}_2. \tag{27}$$

# B    STABILITY OF THE FIRST AND SECOND MOMENTS

Using our notation (see App. A), for all $\boldsymbol{v} \in \mathbb{R}^d$ we have $\boldsymbol{v}^\perp = \boldsymbol{P}_{\mathcal{N}^\perp(\boldsymbol{H})}\boldsymbol{v}$ and $\boldsymbol{v}^\| = \boldsymbol{P}_{\mathcal{N}(\boldsymbol{H})}\boldsymbol{v}$ . Since $\boldsymbol{H}$ is symmetric,

$$\boldsymbol{P}_{\mathcal{N}(\boldsymbol{H})}\boldsymbol{H} = \boldsymbol{H}\boldsymbol{P}_{\mathcal{N}(\boldsymbol{H})} = \boldsymbol{0}, \quad\text{and}\quad \boldsymbol{P}_{\mathcal{N}^\perp(\boldsymbol{H})}\boldsymbol{H} = \boldsymbol{H}\boldsymbol{P}_{\mathcal{N}^\perp(\boldsymbol{H})} = \boldsymbol{H}. \tag{28}$$

If $\boldsymbol{H}_i \in \mathcal{S}_+(\mathbb{R}^{d \times d})$ for all $i \in [n]$, then the null space of $\boldsymbol{H}$ is contained in the null space of each $\boldsymbol{H}_i$, and therefore we also have that

$$\boldsymbol{P}_{\mathcal{N}(\boldsymbol{H})}\boldsymbol{H}_i = \boldsymbol{H}_i\boldsymbol{P}_{\mathcal{N}(\boldsymbol{H})} = \boldsymbol{0}, \quad\text{and}\quad \boldsymbol{P}_{\mathcal{N}^\perp(\boldsymbol{H})}\boldsymbol{H}_i = \boldsymbol{H}_i\boldsymbol{P}_{\mathcal{N}^\perp(\boldsymbol{H})} = \boldsymbol{H}_i. \tag{29}$$

## B.1    LINEARIZED DYNAMICS

The linearized dynamics near $\boldsymbol{\theta}^*$ is

$$\boldsymbol{\theta}_{t+1} = \boldsymbol{\theta}_t - \frac{\eta}{B}\sum_{i\in\mathfrak{B}_t}\boldsymbol{H}_i(\boldsymbol{\theta}_t - \boldsymbol{\theta}^*) - \frac{\eta}{B}\sum_{i\in\mathfrak{B}_t}\boldsymbol{g}_i. \tag{30}$$

Therefore,

$$\boldsymbol{\theta}_{t+1} - \boldsymbol{\theta}^* = \boldsymbol{\theta}_t - \boldsymbol{\theta}^* - \frac{\eta}{B}\sum_{i\in\mathfrak{B}_t}\boldsymbol{H}_i(\boldsymbol{\theta}_t - \boldsymbol{\theta}^*) - \frac{\eta}{B}\sum_{i\in\mathfrak{B}_t}\boldsymbol{g}_i$$

$$= \left(\boldsymbol{I} - \frac{\eta}{B}\sum_{i\in\mathfrak{B}_t}\boldsymbol{H}_i\right)(\boldsymbol{\theta}_t - \boldsymbol{\theta}^*) - \frac{\eta}{B}\sum_{i\in\mathfrak{B}_t}\boldsymbol{g}_i. \tag{31}$$

Here we assume that the batches are chosen uniformly at random, independently across iterations.

**Linearized dynamics in the orthogonal complement.** Under the assumption that $\boldsymbol{H}_i \in \mathcal{S}_+(\mathbb{R}^{d \times d})$ for all $i \in [n]$, the linearized dynamics in the orthogonal complement is given by

$$\boldsymbol{\theta}_{t+1}^\perp - \boldsymbol{\theta}^{*\perp} = \boldsymbol{P}_{\mathcal{N}^\perp(\boldsymbol{H})}\left(\boldsymbol{\theta}_{t+1} - \boldsymbol{\theta}^*\right)$$

$$= \boldsymbol{P}_{\mathcal{N}^\perp(\boldsymbol{H})}\left(\boldsymbol{I} - \frac{\eta}{B}\sum_{i\in\mathfrak{B}_t}\boldsymbol{H}_i\right)(\boldsymbol{\theta}_t - \boldsymbol{\theta}^*) - \frac{\eta}{B}\sum_{i\in\mathfrak{B}_t}\boldsymbol{P}_{\mathcal{N}^\perp(\boldsymbol{H})}\boldsymbol{g}_i$$

$$= \left(\boldsymbol{P}_{\mathcal{N}^\perp(\boldsymbol{H})} - \frac{\eta}{B}\sum_{i\in\mathfrak{B}_t}\boldsymbol{P}_{\mathcal{N}^\perp(\boldsymbol{H})}\boldsymbol{H}_i\right)(\boldsymbol{\theta}_t - \boldsymbol{\theta}^*) - \frac{\eta}{B}\sum_{i\in\mathfrak{B}_t}\boldsymbol{g}_i^\perp$$

$$= \left(\boldsymbol{P}_{\mathcal{N}^\perp(\boldsymbol{H})} - \frac{\eta}{B}\sum_{i\in\mathfrak{B}_t}\boldsymbol{H}_i\boldsymbol{P}_{\mathcal{N}^\perp(\boldsymbol{H})}\right)(\boldsymbol{\theta}_t - \boldsymbol{\theta}^*) - \frac{\eta}{B}\sum_{i\in\mathfrak{B}_t}\boldsymbol{g}_i^\perp$$

$$= \left(\boldsymbol{I} - \frac{\eta}{B}\sum_{i\in\mathfrak{B}_t}\boldsymbol{H}_i\right)\boldsymbol{P}_{\mathcal{N}^\perp(\boldsymbol{H})}(\boldsymbol{\theta}_t - \boldsymbol{\theta}^*) - \frac{\eta}{B}\sum_{i\in\mathfrak{B}_t}\boldsymbol{g}_i^\perp$$

$$= \left(\boldsymbol{I} - \frac{\eta}{B}\sum_{i\in\mathfrak{B}_t}\boldsymbol{H}_i\right)\left(\boldsymbol{\theta}_{t+1}^\perp - \boldsymbol{\theta}^{*\perp}\right) - \frac{\eta}{B}\sum_{i\in\mathfrak{B}_t}\boldsymbol{g}_i^\perp. \tag{32}$$

**Linearized dynamics in the null space.** Under the assumption that $\boldsymbol{H}_i \in \mathcal{S}_+(\mathbb{R}^{d\times d})$ for all $i \in [n]$, the linearized dynamics in the null space is given by

$$
\begin{aligned}
\boldsymbol{\theta}^{\|}_{t+1} - \boldsymbol{\theta}^{*\|} &= \boldsymbol{P}_{\mathcal{N}(\boldsymbol{H})}\left(\boldsymbol{\theta}_{t+1} - \boldsymbol{\theta}^*\right) \\
&= \boldsymbol{P}_{\mathcal{N}(\boldsymbol{H})}\left(\boldsymbol{I} - \frac{\eta}{B}\sum_{i\in\mathfrak{B}_t}\boldsymbol{H}_i\right)(\boldsymbol{\theta}_t - \boldsymbol{\theta}^*) - \frac{\eta}{B}\sum_{i\in\mathfrak{B}_t}\boldsymbol{P}_{\mathcal{N}(\boldsymbol{H})}\boldsymbol{g}_i \\
&= \left(\boldsymbol{P}_{\mathcal{N}(\boldsymbol{H})} - \frac{\eta}{B}\sum_{i\in\mathfrak{B}_t}\boldsymbol{P}_{\mathcal{N}(\boldsymbol{H})}\boldsymbol{H}_i\right)(\boldsymbol{\theta}_t - \boldsymbol{\theta}^*) - \frac{\eta}{B}\sum_{i\in\mathfrak{B}_t}\boldsymbol{g}^{\|}_i \\
&= \left(\boldsymbol{P}_{\mathcal{N}(\boldsymbol{H})} - \boldsymbol{0}\right)(\boldsymbol{\theta}_t - \boldsymbol{\theta}^*) - \frac{\eta}{B}\sum_{i\in\mathfrak{B}_t}\boldsymbol{g}^{\|}_i \\
&= \boldsymbol{P}_{\mathcal{N}(\boldsymbol{H})}(\boldsymbol{\theta}_t - \boldsymbol{\theta}^*) - \frac{\eta}{B}\sum_{i\in\mathfrak{B}_t}\boldsymbol{g}^{\|}_i \\
&= \boldsymbol{\theta}^{\|}_t - \boldsymbol{\theta}^{*\|} - \frac{\eta}{B}\sum_{i\in\mathfrak{B}_t}\boldsymbol{g}^{\|}_i.
\end{aligned}
\tag{33}
$$

Namely,

$$
\boldsymbol{\theta}^{\|}_{t+1} = \boldsymbol{\theta}^{\|}_t - \frac{\eta}{B}\sum_{i\in\mathfrak{B}_t}\boldsymbol{g}^{\|}_i.
\tag{34}
$$

Note that if $\boldsymbol{g}^{\|}_i = \boldsymbol{0}$ for all $i \in [n]$ then

$$
\boldsymbol{\theta}^{\|}_{t+1} = \boldsymbol{\theta}^{\|}_t.
\tag{35}
$$

### B.2 Mean dynamics (proof of Theorem 1)

First, we compute the mean of the linearized dynamics.

$$
\begin{aligned}
\boldsymbol{\mu}_{t+1} = \mathbb{E}\left[\boldsymbol{\theta}_{t+1} - \boldsymbol{\theta}^*\right] &= \mathbb{E}\left[\left(\boldsymbol{I} - \frac{\eta}{B}\sum_{i\in\mathfrak{B}_t}\boldsymbol{H}_i\right)(\boldsymbol{\theta}_t - \boldsymbol{\theta}^*)\right] - \mathbb{E}\left[\frac{\eta}{B}\sum_{i\in\mathfrak{B}_t}\boldsymbol{g}_i\right] \\
&= \mathbb{E}\left[\left(\boldsymbol{I} - \frac{\eta}{B}\sum_{i\in\mathfrak{B}_t}\boldsymbol{H}_i\right)\mathbb{E}\left[(\boldsymbol{\theta}_t - \boldsymbol{\theta}^*)|\mathfrak{B}_t\right]\right] \\
&= (\boldsymbol{I} - \eta\boldsymbol{H})\,\mathbb{E}\left[(\boldsymbol{\theta}_t - \boldsymbol{\theta}^*)\right] \\
&= (\boldsymbol{I} - \eta\boldsymbol{H})\,\boldsymbol{\mu}_t,
\end{aligned}
\tag{36}
$$

where in the second line we used the law of total expectation, as well as (6). This system is stable if and only if the spectral radius $\rho(\boldsymbol{I} - \eta\boldsymbol{H}) \leq 1$. This condition is equivalent to $\lambda_{\max}(\boldsymbol{H}) \leq 2/\eta$ (see proof in, *e.g.,* Cohen *et al.* (2021); Mulayoff *et al.* (2021)), thus proving point 2 of Theorem 1.

**Mean dynamics in the orthogonal complement.** In a similar manner, taking the expectation of both sides of (32) and using (6), we get

$$
\boldsymbol{\mu}^{\perp}_{t+1} = (\boldsymbol{I} - \eta\boldsymbol{H})\,\boldsymbol{\mu}^{\perp}_t,
\tag{37}
$$

Note that for all $t \geq 0$,

$$
\boldsymbol{\mu}^{\perp}_{t+1} = \boldsymbol{P}_{\mathcal{N}^{\perp}(\boldsymbol{H})}\boldsymbol{\mu}^{\perp}_{t+1} = \boldsymbol{P}_{\mathcal{N}^{\perp}(\boldsymbol{H})}(\boldsymbol{I} - \eta\boldsymbol{H})\,\boldsymbol{\mu}^{\perp}_t = \left(\boldsymbol{P}_{\mathcal{N}^{\perp}(\boldsymbol{H})} - \eta\boldsymbol{H}\right)\boldsymbol{\mu}^{\perp}_t.
\tag{38}
$$

Namely, $\boldsymbol{\mu}^{\perp}_t = \left(\boldsymbol{P}_{\mathcal{N}^{\perp}(\boldsymbol{H})} - \eta\boldsymbol{H}\right)^t\boldsymbol{\mu}^{\perp}_0$, and thus

$$
\|\boldsymbol{\mu}^{\perp}_t\| = \left\|\left(\boldsymbol{P}_{\mathcal{N}^{\perp}(\boldsymbol{H})} - \eta\boldsymbol{H}\right)^t\boldsymbol{\mu}^{\perp}_0\right\| \leq \left\|\boldsymbol{P}_{\mathcal{N}^{\perp}(\boldsymbol{H})} - \eta\boldsymbol{H}\right\|^t\|\boldsymbol{\mu}^{\perp}_0\|.
\tag{39}
$$

It is easy to show that

$$
\left\|\boldsymbol{P}_{\mathcal{N}^{\perp}(\boldsymbol{H})} - \eta\boldsymbol{H}\right\| = \max_{\lambda_i(\boldsymbol{H})\neq 0}\{|1 - \eta\lambda_i(\boldsymbol{H})|\}.
\tag{40}
$$

Therefore, if $0 < \eta < 2/\lambda_{\max}$, we have that $\left\|\boldsymbol{P}_{\mathcal{N}^{\perp}(\boldsymbol{H})} - \eta\boldsymbol{H}\right\| < 1$ and thus

$$
\lim_{t\to\infty}\|\boldsymbol{\mu}^{\perp}_t\| \leq \lim_{t\to\infty}\left\|\boldsymbol{P}_{\mathcal{N}^{\perp}(\boldsymbol{H})} - \eta\boldsymbol{H}\right\|^t\|\boldsymbol{\mu}^{\perp}_0\| = 0.
\tag{41}
$$

This demonstrates point 3 of Theorem 1.

**Mean dynamics in the null space.** Taking the expectation of both sides of (33) and using (6), we obtain

$$\boldsymbol{\mu}_{t+1}^{\parallel} = \boldsymbol{\mu}_t^{\parallel}. \tag{42}$$

This demonstrates that for all $t \geq 0$,

$$\mathbb{E}\left[\boldsymbol{\theta}_t^{\parallel} - \boldsymbol{\theta}^{*\parallel}\right] = \boldsymbol{\mu}_t^{\parallel} = \boldsymbol{\mu}_0^{\parallel} = \mathbb{E}\left[\boldsymbol{\theta}_0^{\parallel} - \boldsymbol{\theta}^{*\parallel}\right], \tag{43}$$

so that

$$\mathbb{E}\left[\boldsymbol{\theta}_t^{\parallel}\right] = \mathbb{E}\left[\boldsymbol{\theta}_0^{\parallel}\right]. \tag{44}$$

This proves Point 1 of Theorem 1.

### B.3 COVARIANCE DYNAMICS FOR THE ORTHOGONAL COMPLEMENT

Before providing a complete proof for Theorem 3 (see App. B.7), we next examine the evolution over time of the covariance of the parameter vector. We start by focusing on the orthogonal complement space. Define

$$\boldsymbol{A}_t = \boldsymbol{I} - \frac{\eta}{B} \sum_{i \in \mathfrak{B}_t} \boldsymbol{H}_i \qquad \text{and} \qquad \boldsymbol{v}_t = \frac{\eta}{B} \sum_{i \in \mathfrak{B}_t} \boldsymbol{g}_i, \tag{45}$$

so that (32) can be compactly written as

$$\boldsymbol{\theta}_{t+1}^{\perp} - \boldsymbol{\theta}^{*\perp} = \boldsymbol{A}_t\left(\boldsymbol{\theta}_t^{\perp} - \boldsymbol{\theta}^{*\perp}\right) - \boldsymbol{v}_t^{\perp}. \tag{46}$$

Recall that this holds under the assumption that $\boldsymbol{H}_i \in \mathcal{S}_+(\mathbb{R}^{d \times d})$ for all $i \in [n]$. Note that $\{\boldsymbol{A}_t\}$ are i.i.d. and that $\boldsymbol{\theta}_t^{\perp}$ is constructed from $\boldsymbol{A}_0, \ldots, \boldsymbol{A}_{t-1}$, so that $\boldsymbol{\theta}_t^{\perp}$ is independent of $\boldsymbol{A}_t$. We therefore have

$$
\begin{aligned}
\boldsymbol{\Sigma}_{t+1}^{\perp} &= \mathbb{E}\left[\left(\boldsymbol{\theta}_{t+1}^{\perp} - \boldsymbol{\theta}^{*\perp}\right)\left(\boldsymbol{\theta}_{t+1}^{\perp} - \boldsymbol{\theta}^{*\perp}\right)^{\mathrm{T}}\right] \\
&= \mathbb{E}\left[\left(\boldsymbol{A}_t\left(\boldsymbol{\theta}_t^{\perp} - \boldsymbol{\theta}^{*\perp}\right) - \boldsymbol{v}_t^{\perp}\right)\left(\boldsymbol{A}_t\left(\boldsymbol{\theta}_t^{\perp} - \boldsymbol{\theta}^{*\perp}\right) - \boldsymbol{v}_t^{\perp}\right)^{\mathrm{T}}\right] \\
&= \mathbb{E}\left[\boldsymbol{A}_t\left(\boldsymbol{\theta}_t^{\perp} - \boldsymbol{\theta}^{*\perp}\right)\left(\boldsymbol{\theta}_t^{\perp} - \boldsymbol{\theta}^{*\perp}\right)^{\mathrm{T}} \boldsymbol{A}_t^{\mathrm{T}}\right] - \mathbb{E}\left[\boldsymbol{A}_t\left(\boldsymbol{\theta}_t^{\perp} - \boldsymbol{\theta}^{*\perp}\right)\left(\boldsymbol{v}_t^{\perp}\right)^{\mathrm{T}}\right] \\
&\quad - \mathbb{E}\left[\boldsymbol{v}_t^{\perp}\left(\boldsymbol{\theta}_t^{\perp} - \boldsymbol{\theta}^{*\perp}\right)^{\mathrm{T}} \boldsymbol{A}_t^{\mathrm{T}}\right] + \mathbb{E}\left[\boldsymbol{v}_t^{\perp}\left(\boldsymbol{v}_t^{\perp}\right)^{\mathrm{T}}\right] \\
&= \mathbb{E}\left[\boldsymbol{A}_t \mathbb{E}\left[\left(\boldsymbol{\theta}_t^{\perp} - \boldsymbol{\theta}^{*\perp}\right)\left(\boldsymbol{\theta}_t^{\perp} - \boldsymbol{\theta}^{*\perp}\right)^{\mathrm{T}}\right] \boldsymbol{A}_t^{\mathrm{T}}\right] - \mathbb{E}\left[\boldsymbol{A}_t \mathbb{E}\left[\boldsymbol{\theta}_t^{\perp} - \boldsymbol{\theta}^{*\perp}\right]\left(\boldsymbol{v}_t^{\perp}\right)^{\mathrm{T}}\right] \\
&\quad - \mathbb{E}\left[\boldsymbol{v}_t^{\perp} \mathbb{E}\left[\boldsymbol{\theta}_t^{\perp} - \boldsymbol{\theta}^{*\perp}\right]^{\mathrm{T}} \boldsymbol{A}_t^{\mathrm{T}}\right] + \boldsymbol{\Sigma}_{\boldsymbol{v}}^{\perp} \\
&= \mathbb{E}\left[\boldsymbol{A}_t \boldsymbol{\Sigma}_t^{\perp} \boldsymbol{A}_t^{\mathrm{T}}\right] - \mathbb{E}\left[\boldsymbol{A}_t \boldsymbol{\mu}_t^{\perp}(\boldsymbol{v}_t^{\perp})^{\mathrm{T}}\right] - \mathbb{E}\left[\boldsymbol{v}_t^{\perp}(\boldsymbol{\mu}_t^{\perp})^{\mathrm{T}} \boldsymbol{A}_t^{\mathrm{T}}\right] + \boldsymbol{\Sigma}_{\boldsymbol{v}}^{\perp},
\end{aligned} \tag{47}
$$

where in the second equality we used the fact that $\boldsymbol{A}_t$ is independent of $\boldsymbol{\theta}_t^{\perp}$. Using vectorization, the above equation can be written as

$$
\begin{aligned}
\operatorname{vec}\left(\boldsymbol{\Sigma}_{t+1}^{\perp}\right) &= \mathbb{E}\left[\operatorname{vec}\left(\boldsymbol{A}_t \boldsymbol{\Sigma}_t^{\perp} \boldsymbol{A}_t^{\mathrm{T}}\right)\right] - \mathbb{E}\left[\operatorname{vec}\left(\boldsymbol{A}_t \boldsymbol{\mu}_t^{\perp}(\boldsymbol{v}_t^{\perp})^{\mathrm{T}}\right)\right] - \mathbb{E}\left[\operatorname{vec}\left(\boldsymbol{v}_t^{\perp}(\boldsymbol{\mu}_t^{\perp})^{\mathrm{T}} \boldsymbol{A}_t^{\mathrm{T}}\right)\right] + \operatorname{vec}\left(\boldsymbol{\Sigma}_{\boldsymbol{v}}^{\perp}\right) \\
&= \mathbb{E}\left[\boldsymbol{A}_t \otimes \boldsymbol{A}_t\right] \operatorname{vec}\left(\boldsymbol{\Sigma}_t^{\perp}\right) - \mathbb{E}\left[\boldsymbol{v}_t^{\perp} \otimes \boldsymbol{A}_t\right] \boldsymbol{\mu}_t^{\perp} - \mathbb{E}\left[\boldsymbol{A}_t \otimes \boldsymbol{v}_t^{\perp}\right] \boldsymbol{\mu}_t^{\perp} + \operatorname{vec}\left(\boldsymbol{\Sigma}_{\boldsymbol{v}}^{\perp}\right) \\
&= \boldsymbol{Q} \operatorname{vec}\left(\boldsymbol{\Sigma}_t^{\perp}\right) - \left(\mathbb{E}\left[\boldsymbol{v}_t^{\perp} \otimes \boldsymbol{A}_t\right] + \mathbb{E}\left[\boldsymbol{A}_t \otimes \boldsymbol{v}_t^{\perp}\right]\right) \boldsymbol{\mu}_t^{\perp} + \operatorname{vec}\left(\boldsymbol{\Sigma}_{\boldsymbol{v}}^{\perp}\right),
\end{aligned} \tag{48}
$$

where we denoted

$$\boldsymbol{Q} \triangleq \mathbb{E}\left[\boldsymbol{A}_t \otimes \boldsymbol{A}_t\right]. \tag{49}$$

Overall, the joint dynamics of $\boldsymbol{\Sigma}_t^{\perp}$ and $\boldsymbol{\mu}_t^{\perp}$ is given by

$$
\begin{pmatrix} \boldsymbol{\mu}_{t+1}^{\perp} \\ \operatorname{vec}\left(\boldsymbol{\Sigma}_{t+1}^{\perp}\right) \end{pmatrix} = \begin{pmatrix} \boldsymbol{I} - \eta \boldsymbol{H} & \boldsymbol{0} \\ -\mathbb{E}\left[\boldsymbol{v}_t^{\perp} \otimes \boldsymbol{A}_t\right] - \mathbb{E}\left[\boldsymbol{A}_t \otimes \boldsymbol{v}_t^{\perp}\right] & \boldsymbol{Q} \end{pmatrix} \begin{pmatrix} \boldsymbol{\mu}_t^{\perp} \\ \operatorname{vec}\left(\boldsymbol{\Sigma}_t^{\perp}\right) \end{pmatrix} + \begin{pmatrix} \boldsymbol{0} \\ \operatorname{vec}\left(\boldsymbol{\Sigma}_{\boldsymbol{v}}^{\perp}\right) \end{pmatrix}. \tag{50}
$$

In some cases, it is easier to look at a projected version of the transition matrix. In (38) we showed that

$$\boldsymbol{\mu}_{t+1}^{\perp} = \left(\boldsymbol{P}_{\mathcal{N}^{\perp}(\boldsymbol{H})} - \eta \boldsymbol{H}\right) \boldsymbol{\mu}_t^{\perp}. \tag{51}$$

Moreover, from (48),

$$
\begin{aligned}
\operatorname{vec}\left(\boldsymbol{\Sigma}_{t+1}^{\perp}\right) &= \operatorname{vec}\left(\boldsymbol{P}_{\mathcal{N}^{\perp}(\boldsymbol{H})}\boldsymbol{\Sigma}_{t+1}^{\perp}\boldsymbol{P}_{\mathcal{N}^{\perp}(\boldsymbol{H})}\right) \\
&= \left(\boldsymbol{P}_{\mathcal{N}^{\perp}(\boldsymbol{H})}\otimes\boldsymbol{P}_{\mathcal{N}^{\perp}(\boldsymbol{H})}\right)\operatorname{vec}\left(\boldsymbol{\Sigma}_{t+1}^{\perp}\right) \\
&= \left(\boldsymbol{P}_{\mathcal{N}^{\perp}(\boldsymbol{H})}\otimes\boldsymbol{P}_{\mathcal{N}^{\perp}(\boldsymbol{H})}\right)\left(\boldsymbol{Q}\operatorname{vec}\left(\boldsymbol{\Sigma}_{t}^{\perp}\right) - \left(\mathbb{E}\left[\boldsymbol{v}_{t}^{\perp}\otimes\boldsymbol{A}_{t}\right] + \mathbb{E}\left[\boldsymbol{A}_{t}\otimes\boldsymbol{v}_{t}^{\perp}\right]\right)\boldsymbol{\mu}_{t}^{\perp} + \operatorname{vec}\left(\boldsymbol{\Sigma}_{\boldsymbol{v}}^{\perp}\right)\right) \\
&= \left(\boldsymbol{P}_{\mathcal{N}^{\perp}(\boldsymbol{H})}\otimes\boldsymbol{P}_{\mathcal{N}^{\perp}(\boldsymbol{H})}\right)\boldsymbol{Q}\operatorname{vec}\left(\boldsymbol{\Sigma}_{t}^{\perp}\right) \\
&\quad - \left(\boldsymbol{P}_{\mathcal{N}^{\perp}(\boldsymbol{H})}\otimes\boldsymbol{P}_{\mathcal{N}^{\perp}(\boldsymbol{H})}\right)\left(\mathbb{E}\left[\boldsymbol{v}_{t}^{\perp}\otimes\boldsymbol{A}_{t}\right] + \mathbb{E}\left[\boldsymbol{A}_{t}\otimes\boldsymbol{v}_{t}^{\perp}\right]\right)\boldsymbol{\mu}_{t}^{\perp} + \operatorname{vec}\left(\boldsymbol{\Sigma}_{\boldsymbol{v}}^{\perp}\right). \quad (52)
\end{aligned}
$$

Therefore, the linear system in (50) can be written as

$$
\begin{aligned}
&\begin{pmatrix}\boldsymbol{\mu}_{t+1}^{\perp} \\ \operatorname{vec}\left(\boldsymbol{\Sigma}_{t+1}^{\perp}\right)\end{pmatrix} = \\
&\begin{pmatrix}\boldsymbol{P}_{\mathcal{N}^{\perp}(\boldsymbol{H})} - \eta\boldsymbol{H} & \boldsymbol{0} \\ -\left(\boldsymbol{P}_{\mathcal{N}^{\perp}(\boldsymbol{H})}\otimes\boldsymbol{P}_{\mathcal{N}^{\perp}(\boldsymbol{H})}\right)\left(\mathbb{E}\left[\boldsymbol{v}_{t}^{\perp}\otimes\boldsymbol{A}_{t}\right] + \mathbb{E}\left[\boldsymbol{A}_{t}\otimes\boldsymbol{v}_{t}^{\perp}\right]\right) & \left(\boldsymbol{P}_{\mathcal{N}^{\perp}(\boldsymbol{H})}\otimes\boldsymbol{P}_{\mathcal{N}^{\perp}(\boldsymbol{H})}\right)\boldsymbol{Q}\end{pmatrix}\begin{pmatrix}\boldsymbol{\mu}_{t}^{\perp} \\ \operatorname{vec}\left(\boldsymbol{\Sigma}_{t}^{\perp}\right)\end{pmatrix} \\
&\quad + \begin{pmatrix}\boldsymbol{0} \\ \operatorname{vec}\left(\boldsymbol{\Sigma}_{\boldsymbol{v}}^{\perp}\right)\end{pmatrix}. \quad (53)
\end{aligned}
$$

### B.4 THE TRANSITION MATRIX OF THE COVARIANCE DYNAMICS

We now proceed to develop an explicit expression for the covariance transition matrix $\boldsymbol{Q}$ of (49). We have

$$
\begin{aligned}
\mathbb{E}\left[\boldsymbol{A}_{t}\otimes\boldsymbol{A}_{t}\right] &= \mathbb{E}\left[\left(\boldsymbol{I} - \frac{\eta}{B}\sum_{i\in\mathfrak{B}_{t}}\boldsymbol{H}_{i}\right)\otimes\left(\boldsymbol{I} - \frac{\eta}{B}\sum_{i\in\mathfrak{B}_{t}}\boldsymbol{H}_{i}\right)\right] \\
&= \mathbb{E}\left[\boldsymbol{I} - \frac{\eta}{B}\sum_{i\in\mathfrak{B}_{t}}\left(\boldsymbol{I}\otimes\boldsymbol{H}_{i} + \boldsymbol{H}_{i}\otimes\boldsymbol{I}\right) + \frac{\eta^{2}}{B^{2}}\sum_{i,j\in\mathfrak{B}_{t}}\boldsymbol{H}_{i}\otimes\boldsymbol{H}_{j}\right] \\
&= \boldsymbol{I} - \eta\left(\boldsymbol{I}\otimes\boldsymbol{H} + \boldsymbol{H}\otimes\boldsymbol{I}\right) + \eta^{2}\mathbb{E}\left[\frac{1}{B^{2}}\sum_{i,j\in\mathfrak{B}_{t}}\boldsymbol{H}_{i}\otimes\boldsymbol{H}_{j}\right]. \quad (54)
\end{aligned}
$$

Note that

$$
\begin{aligned}
\mathbb{E}\left[\frac{1}{B^{2}}\sum_{i,j\in\mathfrak{B}_{t}}\boldsymbol{H}_{i}\otimes\boldsymbol{H}_{j}\right] &= \mathbb{E}\left[\frac{1}{B^{2}}\sum_{i\neq j\in\mathfrak{B}_{t}}\boldsymbol{H}_{i}\otimes\boldsymbol{H}_{j} + \frac{1}{B^{2}}\sum_{i\in\mathfrak{B}_{t}}\boldsymbol{H}_{i}\otimes\boldsymbol{H}_{i}\right] \\
&= \frac{1}{B^{2}}\times B(B-1)\mathbb{E}\left[\boldsymbol{H}_{i}\otimes\boldsymbol{H}_{j}|i\neq j\in\mathfrak{B}_{t}\right] + \frac{1}{B^{2}}\mathbb{E}\left[\sum_{i\in\mathfrak{B}_{t}}\boldsymbol{H}_{i}\otimes\boldsymbol{H}_{i}\right] \\
&= \frac{B-1}{B}\mathbb{E}\left[\boldsymbol{H}_{i}\otimes\boldsymbol{H}_{j}|i\neq j\in\mathfrak{B}_{t}\right] + \frac{1}{nB}\sum_{i=1}^{n}\boldsymbol{H}_{i}\otimes\boldsymbol{H}_{i}. \quad (55)
\end{aligned}
$$

Specifically using symmetry and (7),

$$
\begin{aligned}
\mathbb{E}\left[\boldsymbol{H}_{i}\otimes\boldsymbol{H}_{j}|i\neq j\in\mathfrak{B}_{t}\right] &= \sum_{i\neq j=1}^{n}\frac{1}{n(n-1)}\boldsymbol{H}_{i}\otimes\boldsymbol{H}_{j} \\
&= \frac{n}{(n-1)}\frac{1}{n^{2}}\sum_{i\neq j=1}^{n}\boldsymbol{H}_{i}\otimes\boldsymbol{H}_{j} \\
&= \frac{n}{(n-1)}\left(\boldsymbol{H}\otimes\boldsymbol{H} - \frac{1}{n^{2}}\sum_{i=1}^{n}\boldsymbol{H}_{i}\otimes\boldsymbol{H}_{i}\right). \quad (56)
\end{aligned}
$$

Hence,

$$
\begin{aligned}
\frac{B-1}{B}\mathbb{E}\left[\boldsymbol{H}_i \otimes \boldsymbol{H}_j | i \neq j \in \mathfrak{B}_t\right] &= \frac{n(B-1)}{B(n-1)}\left(\boldsymbol{H} \otimes \boldsymbol{H} - \frac{1}{n^2}\sum_{i=1}^n \boldsymbol{H}_i \otimes \boldsymbol{H}_i\right) \\
&= \frac{n(B-1)}{B(n-1)}\boldsymbol{H} \otimes \boldsymbol{H} - \frac{B-1}{Bn(n-1)}\sum_{i=1}^n \boldsymbol{H}_i \otimes \boldsymbol{H}_i \\
&= \boldsymbol{H} \otimes \boldsymbol{H} - \frac{n-B}{B(n-1)}\boldsymbol{H} \otimes \boldsymbol{H} - \frac{B-1}{Bn(n-1)}\sum_{i=1}^n \boldsymbol{H}_i \otimes \boldsymbol{H}_i.
\end{aligned}
\tag{57}
$$

Overall,

$$
\begin{aligned}
\mathbb{E}\left[\frac{1}{B^2}\sum_{i,j\in\mathfrak{B}_t}\boldsymbol{H}_i \otimes \boldsymbol{H}_j\right] &= \boldsymbol{H} \otimes \boldsymbol{H} - \frac{n-B}{B(n-1)}\boldsymbol{H} \otimes \boldsymbol{H} - \frac{B-1}{Bn(n-1)}\sum_{i=1}^n \boldsymbol{H}_i \otimes \boldsymbol{H}_i \\
&\quad + \frac{1}{nB}\sum_{i=1}^n \boldsymbol{H}_i \otimes \boldsymbol{H}_i \\
&= \boldsymbol{H} \otimes \boldsymbol{H} - \frac{n-B}{B(n-1)}\boldsymbol{H} \otimes \boldsymbol{H} + \frac{n-B}{B(n-1)} \times \frac{1}{n}\sum_{i=1}^n \boldsymbol{H}_i \otimes \boldsymbol{H}_i \\
&= \boldsymbol{H} \otimes \boldsymbol{H} + \frac{n-B}{B(n-1)}\left(\frac{1}{n}\sum_{i=1}^n \boldsymbol{H}_i \otimes \boldsymbol{H}_i - \boldsymbol{H} \otimes \boldsymbol{H}\right).
\end{aligned}
\tag{58}
$$

Therefore, we have that $\boldsymbol{Q}$ can be written as

$$
\begin{aligned}
\boldsymbol{Q}(B,\eta) &= \boldsymbol{I} - \eta\left(\boldsymbol{I} \otimes \boldsymbol{H} + \boldsymbol{H} \otimes \boldsymbol{I}\right) + \eta^2 \boldsymbol{H} \otimes \boldsymbol{H} + \eta^2 \frac{n-B}{B(n-1)}\left(\frac{1}{n}\sum_{i=1}^n \boldsymbol{H}_i \otimes \boldsymbol{H}_i - \boldsymbol{H} \otimes \boldsymbol{H}\right) \\
&= \boldsymbol{I} - \eta\left(\boldsymbol{I} \otimes \boldsymbol{H} + \boldsymbol{H} \otimes \boldsymbol{I}\right) + \eta^2 \left[(1-p)\boldsymbol{H} \otimes \boldsymbol{H} + p \times \frac{1}{n}\sum_{i=1}^n \boldsymbol{H}_i \otimes \boldsymbol{H}_i\right] \\
&= (\boldsymbol{I} - \eta\boldsymbol{H}) \otimes (\boldsymbol{I} - \eta\boldsymbol{H}) + \eta^2 p \times \left(\frac{1}{n}\sum_{i=1}^n (\boldsymbol{H}_i - \boldsymbol{H}) \otimes (\boldsymbol{H}_i - \boldsymbol{H})\right) \\
&= (1-p) \times (\boldsymbol{I} - \eta\boldsymbol{H}) \otimes (\boldsymbol{I} - \eta\boldsymbol{H}) + p \times \frac{1}{n}\sum_{i=1}^n (\boldsymbol{I} - \eta\boldsymbol{H}_i) \otimes (\boldsymbol{I} - \eta\boldsymbol{H}_i),
\end{aligned}
\tag{59}
$$

where

$$
p = \frac{n-B}{B(n-1)}.
\tag{60}
$$

## B.5 Covariance matrix of the gradient noise

We now develop an explicit expression for the covariance matrix of the gradient noise $\boldsymbol{v}$ of (45). We have

$$
\begin{aligned}
\boldsymbol{\Sigma_v} = \mathbb{E}\left[\boldsymbol{v}_t \boldsymbol{v}_t^{\mathrm{T}}\right] &= \left(\frac{\eta}{B}\right)^2 \mathbb{E}\left[\sum_{i,j\in\mathfrak{B}_t}\boldsymbol{g}_i \boldsymbol{g}_j^{\mathrm{T}}\right] \\
&= \left(\frac{\eta}{B}\right)^2 \mathbb{E}\left[\sum_{i\neq j\in\mathfrak{B}_t}\boldsymbol{g}_i \boldsymbol{g}_j^{\mathrm{T}} + \sum_{i\in\mathfrak{B}_t}\boldsymbol{g}_i \boldsymbol{g}_i^{\mathrm{T}}\right] \\
&= \left(\frac{\eta}{B}\right)^2 \left(B(B-1)\mathbb{E}\left[\boldsymbol{g}_i \boldsymbol{g}_j^{\mathrm{T}} | i \neq j \in \mathfrak{B}_t\right] + \frac{B}{n}\sum_{i=1}^n \boldsymbol{g}_i \boldsymbol{g}_i^{\mathrm{T}}\right).
\end{aligned}
\tag{61}
$$

Observe that

$$
\begin{aligned}
\mathbb{E}\left[\boldsymbol{g}_i \boldsymbol{g}_j^{\mathrm{T}} \big| i \neq j \in \mathfrak{B}_t\right] &= \sum_{i \neq j=1}^n \frac{1}{n(n-1)} \boldsymbol{g}_i \boldsymbol{g}_j^{\mathrm{T}} \\
&= \frac{1}{n(n-1)} \left( \sum_{i,j=1}^n \boldsymbol{g}_i \boldsymbol{g}_j^{\mathrm{T}} - \sum_{i=1}^n \boldsymbol{g}_i \boldsymbol{g}_i^{\mathrm{T}} \right) \\
&= \frac{1}{n(n-1)} \left( \left(\sum_{i=1}^n \boldsymbol{g}_i\right) \left(\sum_{i=1}^n \boldsymbol{g}_i\right)^{\mathrm{T}} - \sum_{i=1}^n \boldsymbol{g}_i \boldsymbol{g}_i^{\mathrm{T}} \right) \\
&= -\frac{1}{n(n-1)} \sum_{i=1}^n \boldsymbol{g}_i \boldsymbol{g}_i^{\mathrm{T}},
\end{aligned}
\tag{62}
$$

where in the last step we used (6). Thus,

$$
\begin{aligned}
\boldsymbol{\Sigma_v} &= \left(\frac{\eta}{B}\right)^2 \left( \frac{B}{n} - \frac{B(B-1)}{n(n-1)} \right) \sum_{i=1}^n \boldsymbol{g}_i \boldsymbol{g}_i^{\mathrm{T}} \\
&= \left(\frac{\eta}{B}\right)^2 \times \frac{B(n-B)}{n(n-1)} \sum_{i=1}^n \boldsymbol{g}_i \boldsymbol{g}_i^{\mathrm{T}} \\
&= \eta^2 \frac{n-B}{B(n-1)} \times \frac{1}{n} \sum_{i=1}^n \boldsymbol{g}_i \boldsymbol{g}_i^{\mathrm{T}} \\
&= \eta^2 p \boldsymbol{\Sigma_g},
\end{aligned}
\tag{63}
$$

where we denoted

$$
\boldsymbol{\Sigma_g} = \frac{1}{n} \sum_{i=1}^n \boldsymbol{g}_i \boldsymbol{g}_i^{\mathrm{T}}.
\tag{64}
$$

### B.6 THE NULL SPACES OF $\boldsymbol{C}$, $\boldsymbol{D}$ AND $\boldsymbol{E}$

Let us now analyze the relation between the null spaces of $\boldsymbol{C}$, $\boldsymbol{D}$ and $\boldsymbol{E}$. First, it is easy to see that under the assumption that $\boldsymbol{H}_i \in \mathcal{S}_+(\mathbb{R}^{d \times d})$ for all $i \in [n]$,

$$
\mathcal{N}(\boldsymbol{H}) = \bigcap_{i=1}^n \mathcal{N}(\boldsymbol{H}_i),
\tag{65}
$$

where $\mathcal{N}(\cdot)$ denotes null space of a matrix. Here we show the following.

**Lemma 1.** *Assume that $\boldsymbol{H}_i \in \mathcal{S}_+(\mathbb{R}^{d \times d})$ for all $i \in [n]$ and let*

$$
\boldsymbol{C} = \frac{1}{2} \boldsymbol{H} \oplus \boldsymbol{H},
$$

$$
\boldsymbol{D} = (1-p)\, \boldsymbol{H} \otimes \boldsymbol{H} + p\, \frac{1}{n} \sum_{i=1}^n \boldsymbol{H}_i \otimes \boldsymbol{H}_i,
$$

$$
\boldsymbol{E} = \frac{1}{n} \sum_{i=1}^n (\boldsymbol{H}_i - \boldsymbol{H}) \otimes (\boldsymbol{H}_i - \boldsymbol{H}).
\tag{66}
$$

*Then $\mathcal{N}(\boldsymbol{C}) \subseteq \mathcal{N}(\boldsymbol{D})$ and $\mathcal{N}(\boldsymbol{C}) \subseteq \mathcal{N}(\boldsymbol{E})$.*

*Proof.* Let $\boldsymbol{u} \in \mathcal{N}(\boldsymbol{C})$ and denote $\boldsymbol{U} = \mathrm{vec}^{-1}(\boldsymbol{u})$, then

$$
\boldsymbol{0} = 2\boldsymbol{C}\boldsymbol{u} = \boldsymbol{H} \oplus \boldsymbol{H}\boldsymbol{u} = (\boldsymbol{H} \otimes \boldsymbol{I} + \boldsymbol{I} \otimes \boldsymbol{H})\boldsymbol{u}.
\tag{67}
$$

In matrix form we get

$$
\boldsymbol{U}\boldsymbol{H} + \boldsymbol{H}\boldsymbol{U} = \boldsymbol{0}.
\tag{68}
$$

Let us take the Frobenius norm, then

$$\|UH + HU\|_{\mathrm{F}}^2 = \|UH\|_{\mathrm{F}}^2 + \|HU\|_{\mathrm{F}}^2 + 2\mathrm{Tr}\left((UH)^{\mathrm{T}}HU\right) = 0, \tag{69}$$

where

$$\mathrm{Tr}\left((UH)^{\mathrm{T}}HU\right) = \mathrm{Tr}\left(HU^{\mathrm{T}}HU\right) = \mathrm{Tr}\left(H^{\frac{1}{2}}U^{\mathrm{T}}HUH^{\frac{1}{2}}\right) \geq 0 \tag{70}$$

because $H^{\frac{1}{2}}U^{\mathrm{T}}HUH^{\frac{1}{2}} = (H^{\frac{1}{2}}UH^{\frac{1}{2}})^{\mathrm{T}}(H^{\frac{1}{2}}UH^{\frac{1}{2}})$ is PSD. This implies that

$$\|UH\|_{\mathrm{F}}^2 = \|HU\|_{\mathrm{F}}^2 = 0. \tag{71}$$

Thus, $u \in \mathcal{N}(C)$ if and only if $UH = 0$ and $HU = 0$. Since the null space of $H$ is the intersection of $\{H_i\}$ we have that $U$ also satisfies $H_i U = U H_i = 0$ for all $i \in [n]$. Now,

$$Du = (1 - p)\, H \otimes Hu + p\, \frac{1}{n}\sum_{i=1}^{n} H_i \otimes H_i u, \tag{72}$$

and in matrix form,

$$\mathrm{vec}^{-1}(Du) = (1 - p)\, HUH + p\, \frac{1}{n}\sum_{i=1}^{n} H_i U H_i = 0. \tag{73}$$

Namely, $u \in \mathcal{N}(D)$. Similarly,

$$Eu = \frac{1}{n}\sum_{i=1}^{n}(H_i - H) \otimes (H_i - H)u, \tag{74}$$

and in matrix form,

$$\mathrm{vec}^{-1}(Eu) = \frac{1}{n}\sum_{i=1}^{n}(H_i - H)U(H_i - H) = 0. \tag{75}$$

Namely, $u \in \mathcal{N}(E)$. ∎

### B.7 PROOF OF THEOREM 3

We are now ready to prove Theorem 3.

**First statement.** In (35) we showed that for interpolating minima $\theta_{t+1}^{\|} = \theta_t^{\|}$, which completes the proof for the first statement of the theorem.

**Second statement.** Ma & Ying (2021) showed that the second moment $\Sigma_t = [(\theta_t - \theta^*)(\theta_t - \theta^*)^{\mathrm{T}}]$ for interpolating minima evolves over time as

$$\mathrm{vec}\left(\Sigma_{t+1}\right) = Q\,\mathrm{vec}\left(\Sigma_t\right), \tag{76}$$

where $Q$ is given in (10). Since $\Sigma_t$ is PSD by definition, we only care about the effect of $Q$ on vectorizations of PSD matrices. Therefore, we have that $\{\Sigma_t\}$ are bounded if and only if (see proof in (Ma & Ying, 2021))

$$\max_{\Sigma \in \mathcal{S}_+(\mathbb{R}^{d \times d})} \frac{\|Q(\eta, B)\,\mathrm{vec}\left(\Sigma\right)\|}{\|\Sigma\|_{\mathrm{F}}} \leq 1. \tag{77}$$

To obtain the second result of Thm. 3 we first rearrange the terms in $Q$ as

$$Q(\eta, B) = (1 - p) \times (I - \eta H) \otimes (I - \eta H) + p \times \frac{1}{n}\sum_{i=1}^{n}(I - \eta H_i) \otimes (I - \eta H_i). \tag{78}$$

By applying Theorem 6, we have that the spectral radius of $Q$ equals its top eigenvalue, and the corresponding eigenvector is a vectorization of a PSD matrix (see proof of Thm. 6 in App. C). Thus, we have that the maximizer for the constrained optimization problem in (77) is, in fact, the top

eigenvalue of $Q$. Hence, the linear system is stable if and only if $\lambda_{\max}(Q) \leq 1$. Regathering the terms of $Q$ gives

$$
\begin{aligned}
Q &= I - \eta(H \oplus H) + \eta^2 \left[ (1-p)H \otimes H + p \times \frac{1}{n} \sum_{i=1}^{n} H_i \otimes H_i \right] \\
&= I - 2\eta C + \eta^2 D.
\end{aligned}
\tag{79}
$$

Because $Q$ is symmetric, the condition $\lambda_{\max}(Q) \leq 1$ is equivalent to the requirement that $u^{\mathrm{T}} Q u \leq 1$ for all $u \in \mathbb{S}^{d^2-1}$. Recall from App. B.6 that $\mathcal{N}(C) \subseteq \mathcal{N}(D)$. Therefore, if $u \in \mathcal{N}(C)$ then also $u \in \mathcal{N}(D)$ and we get

$$
1 \geq u^{\mathrm{T}} Q u = 1 - 2\eta u^{\mathrm{T}} C u + \eta^2 u^{\mathrm{T}} D u = 1.
\tag{80}
$$

Namely, directions in the null space of $C$ do not impose any constraint on the learning rate, and thus can be ignored. Additionally, if $u \in \mathcal{N}(D)$ but $u \notin \mathcal{N}(C)$, then

$$
1 \geq u^{\mathrm{T}} Q u = 1 - 2\eta u^{\mathrm{T}} C u + \eta^2 u^{\mathrm{T}} D u = 1 - 2\eta u^{\mathrm{T}} C u,
\tag{81}
$$

which is true for all $\eta \geq 0$, because $C$ is PSD. Now,

$$
u^{\mathrm{T}} Q u = 1 - 2\eta u^{\mathrm{T}} C u + \eta^2 u^{\mathrm{T}} D u \leq 1
\tag{82}
$$

happens for all $u \notin \mathcal{N}(D)$ (which also results in $u \notin \mathcal{N}(C)$), if and only if

$$
0 \leq \eta \leq 2 \inf_{u \in \mathbb{S}^{d^2-1} : u \notin \mathcal{N}(D)} \left\{ \frac{u^{\mathrm{T}} C u}{u^{\mathrm{T}} D u} \right\}.
\tag{83}
$$

Therefore, the stability threshold $\eta_{\mathrm{var}}^*$ is given by

$$
\begin{aligned}
\eta_{\mathrm{var}}^* &= 2 \inf_{u \in \mathbb{S}^{d^2-1} : u \notin \mathcal{N}(D)} \left\{ \frac{u^{\mathrm{T}} C u}{u^{\mathrm{T}} D u} \right\} \\
&= 2 \left( \sup_{u \in \mathbb{S}^{d^2-1} : u \notin \mathcal{N}(D)} \left\{ \frac{u^{\mathrm{T}} D u}{u^{\mathrm{T}} C u} \right\} \right)^{-1}.
\end{aligned}
\tag{84}
$$

Note that $u$'s norm has no effect, and therefore we can relax the constraint $u \in \mathbb{S}^{d^2-1}$. Additionally, we can also reduce the constraint $u \notin \mathcal{N}(D)$ to $u \notin \mathcal{N}(C)$, because the supremum in (84) is over a non-negative function, and will not be affected by adding to the domain points at which the function vanishes. Since $\mathcal{N}(C) \subseteq \mathcal{N}(D)$ we have that $\mathcal{N}^{\perp}(D) \subseteq \mathcal{N}^{\perp}(C)$ and therefore

$$
P_{\mathcal{N}^{\perp}(C)} D = D P_{\mathcal{N}^{\perp}(C)} = D,
\tag{85}
$$

where $P_{\mathcal{N}^{\perp}(C)}$ is the projection matrix onto the orthogonal complement of the null space of $C$. Additionally, $C$ is PSD, and therefore $C^{\frac{1}{2}}$ exists and is also PSD, so that

$$
P_{\mathcal{N}^{\perp}(C)} = \left( C^{\frac{1}{2}} \right)^{\dagger} C^{\frac{1}{2}} = C^{\frac{1}{2}} \left( C^{\frac{1}{2}} \right)^{\dagger}.
\tag{86}
$$

Therefore,

$$
\begin{aligned}
\sup_{u \notin \mathcal{N}(C)} \left\{ \frac{u^{\mathrm{T}} D u}{u^{\mathrm{T}} C u} \right\} &= \sup_{u \notin \mathcal{N}(C)} \left\{ \frac{u^{\mathrm{T}} P_{\mathcal{N}^{\perp}(C)} D P_{\mathcal{N}^{\perp}(C)} u}{u^{\mathrm{T}} C u} \right\} \\
&= \sup_{u \notin \mathcal{N}(C)} \left\{ \frac{u^{\mathrm{T}} C^{\frac{1}{2}} \left( C^{\frac{1}{2}} \right)^{\dagger} D \left( C^{\frac{1}{2}} \right)^{\dagger} C^{\frac{1}{2}} u}{u^{\mathrm{T}} C^{\frac{1}{2}} C^{\frac{1}{2}} u} \right\} \\
&= \sup_{u \notin \mathcal{N}(C)} \left\{ \frac{\left( C^{\frac{1}{2}} u \right)^{\mathrm{T}} \left( C^{\frac{1}{2}} \right)^{\dagger} D \left( C^{\frac{1}{2}} \right)^{\dagger} \left( C^{\frac{1}{2}} u \right)}{\left( C^{\frac{1}{2}} u \right)^{\mathrm{T}} \left( C^{\frac{1}{2}} u \right)} \right\},
\end{aligned}
\tag{87}
$$

where in the first step we used (85), and in the second step we used (86). By a simple change of variables $\boldsymbol{y} = \boldsymbol{C}^{\frac{1}{2}} \boldsymbol{u} \in \mathcal{N}^{\perp}(\boldsymbol{C})$ we get

$$
\max_{\boldsymbol{y} \in \mathcal{N}^{\perp}(\boldsymbol{C})} \left\{ \frac{\boldsymbol{y}^{\mathrm{T}} \left(\boldsymbol{C}^{\frac{1}{2}}\right)^{\dagger} \boldsymbol{D} \left(\boldsymbol{C}^{\frac{1}{2}}\right)^{\dagger} \boldsymbol{y}}{\boldsymbol{y}^{\mathrm{T}} \boldsymbol{y}} \right\} = \max_{\boldsymbol{y} \in \mathbb{R}^{d^2}} \left\{ \frac{\boldsymbol{y}^{\mathrm{T}} \left(\boldsymbol{C}^{\frac{1}{2}}\right)^{\dagger} \boldsymbol{D} \left(\boldsymbol{C}^{\frac{1}{2}}\right)^{\dagger} \boldsymbol{y}}{\boldsymbol{y}^{\mathrm{T}} \boldsymbol{y}} \right\}
$$
$$
= \lambda_{\max} \left( \left(\boldsymbol{C}^{\frac{1}{2}}\right)^{\dagger} \boldsymbol{D} \left(\boldsymbol{C}^{\frac{1}{2}}\right)^{\dagger} \right), \tag{88}
$$

where in the first step we used the fact that adding to $\boldsymbol{y}$ a component in $\mathcal{N}(\boldsymbol{C})$ will increase the denominator by $\|\boldsymbol{P}_{\mathcal{N}(\boldsymbol{C})}\boldsymbol{y}\|^2$ but will not affect the numerator. Namely, the optimum cannot be attained by $\boldsymbol{y} \notin \mathcal{N}^{\perp}(\boldsymbol{C})$. Now, let $(\lambda_i, \boldsymbol{y}_i)$ be an eigenpair of $(\boldsymbol{C}^{\frac{1}{2}})^{\dagger} \boldsymbol{D} (\boldsymbol{C}^{\frac{1}{2}})^{\dagger}$, then we have

$$
\lambda_i \boldsymbol{y}_i = \left(\boldsymbol{C}^{\frac{1}{2}}\right)^{\dagger} \boldsymbol{D} \left(\boldsymbol{C}^{\frac{1}{2}}\right)^{\dagger} \boldsymbol{y}_i. \tag{89}
$$

Since we only care about nonzero eigenvalues, we can assume that $\lambda_i \neq 0$, *i.e.*, $\boldsymbol{y}_i \notin \mathcal{N}(\boldsymbol{C})$. Multiplying by $(\boldsymbol{C}^{\frac{1}{2}})^{\dagger}$ from the left we get

$$
\lambda_i \left(\boldsymbol{C}^{\frac{1}{2}}\right)^{\dagger} \boldsymbol{y}_i = \left(\boldsymbol{C}^{\frac{1}{2}}\right)^{\dagger} \left(\boldsymbol{C}^{\frac{1}{2}}\right)^{\dagger} \boldsymbol{D} \left(\boldsymbol{C}^{\frac{1}{2}}\right)^{\dagger} \boldsymbol{y}_i
$$
$$
= \boldsymbol{C}^{\dagger} \boldsymbol{D} \left(\boldsymbol{C}^{\frac{1}{2}}\right)^{\dagger} \boldsymbol{y}_i. \tag{90}
$$

Namely, $(\boldsymbol{C}^{\frac{1}{2}})^{\dagger} \boldsymbol{y}_i \neq \boldsymbol{0}$ is an eigenvector of $\boldsymbol{C}^{\dagger} \boldsymbol{D}$ with eigenvalue $\lambda_i$, which means that $(\boldsymbol{C}^{\frac{1}{2}})^{\dagger} \boldsymbol{D} (\boldsymbol{C}^{\frac{1}{2}})^{\dagger}$ and $\boldsymbol{C}^{\dagger} \boldsymbol{D}$ have the same eigenvalues[3], and in particular have the same top eigenvalue. Therefore,

$$
\lambda_{\max} \left( \left(\boldsymbol{C}^{\frac{1}{2}}\right)^{\dagger} \boldsymbol{D} \left(\boldsymbol{C}^{\frac{1}{2}}\right)^{\dagger} \right) = \lambda_{\max} \left( \boldsymbol{C}^{\dagger} \boldsymbol{D} \right). \tag{91}
$$

Overall, we showed that the condition in (11) is equivalent to

$$
\eta \leq \frac{2}{\lambda_{\max} \left( \boldsymbol{C}^{\dagger} \boldsymbol{D} \right)}. \tag{92}
$$

This completes the proof for the second statement of the theorem.

**Third statement.** For the third statement of the theorem, note that for all $t \geq 0$

$$
\mathrm{vec} \left( \boldsymbol{\Sigma}_{t+1}^{\perp} \right) = \mathrm{vec} \left( \boldsymbol{P}_{\mathcal{N}^{\perp}(\boldsymbol{H})} \boldsymbol{\Sigma}_{t+1}^{\perp} \boldsymbol{P}_{\mathcal{N}^{\perp}(\boldsymbol{H})} \right)
$$
$$
= \left( \boldsymbol{P}_{\mathcal{N}^{\perp}(\boldsymbol{H})} \otimes \boldsymbol{P}_{\mathcal{N}^{\perp}(\boldsymbol{H})} \right) \mathrm{vec} \left( \boldsymbol{\Sigma}_{t+1}^{\perp} \right)
$$
$$
= \left( \boldsymbol{P}_{\mathcal{N}^{\perp}(\boldsymbol{H})} \otimes \boldsymbol{P}_{\mathcal{N}^{\perp}(\boldsymbol{H})} \right) \boldsymbol{Q} \, \mathrm{vec} \left( \boldsymbol{\Sigma}_{t}^{\perp} \right). \tag{93}
$$

Namely, $\mathrm{vec} \left( \boldsymbol{\Sigma}_{t}^{\perp} \right) = [(\boldsymbol{P}_{\mathcal{N}^{\perp}(\boldsymbol{H})} \otimes \boldsymbol{P}_{\mathcal{N}^{\perp}(\boldsymbol{H})}) \boldsymbol{Q}]^{t} \mathrm{vec} \left( \boldsymbol{\Sigma}_{0}^{\perp} \right)$, and thus

$$
\left\| \mathrm{vec} \left( \boldsymbol{\Sigma}_{t}^{\perp} \right) \right\| = \left\| [(\boldsymbol{P}_{\mathcal{N}^{\perp}(\boldsymbol{H})} \otimes \boldsymbol{P}_{\mathcal{N}^{\perp}(\boldsymbol{H})}) \boldsymbol{Q}]^{t} \mathrm{vec} \left( \boldsymbol{\Sigma}_{0}^{\perp} \right) \right\| \leq \left\| (\boldsymbol{P}_{\mathcal{N}^{\perp}(\boldsymbol{H})} \otimes \boldsymbol{P}_{\mathcal{N}^{\perp}(\boldsymbol{H})}) \boldsymbol{Q} \right\|^{t} \left\| \mathrm{vec} \left( \boldsymbol{\Sigma}_{0}^{\perp} \right) \right\|. \tag{94}
$$

Here

$$
(\boldsymbol{P}_{\mathcal{N}^{\perp}(\boldsymbol{H})} \otimes \boldsymbol{P}_{\mathcal{N}^{\perp}(\boldsymbol{H})}) \boldsymbol{Q} = (\boldsymbol{P}_{\mathcal{N}^{\perp}(\boldsymbol{H})} \otimes \boldsymbol{P}_{\mathcal{N}^{\perp}(\boldsymbol{H})})
$$
$$
\left[ (1-p)(\boldsymbol{I} - \eta \boldsymbol{H}) \otimes (\boldsymbol{I} - \eta \boldsymbol{H}) + p \times \frac{1}{n} \sum_{i=1}^{n} (\boldsymbol{I} - \eta \boldsymbol{H}_i) \otimes (\boldsymbol{I} - \eta \boldsymbol{H}_i) \right]
$$
$$
= (1-p)(\boldsymbol{P}_{\mathcal{N}^{\perp}(\boldsymbol{H})} - \eta \boldsymbol{H}) \otimes (\boldsymbol{P}_{\mathcal{N}^{\perp}(\boldsymbol{H})} - \eta \boldsymbol{H})
$$
$$
+ p \times \frac{1}{n} \sum_{i=1}^{n} (\boldsymbol{P}_{\mathcal{N}^{\perp}(\boldsymbol{H})} - \eta \boldsymbol{H}_i) \otimes (\boldsymbol{P}_{\mathcal{N}^{\perp}(\boldsymbol{H})} - \eta \boldsymbol{H}_i). \tag{95}
$$

---

[3]Since from Lemma 1 $\mathcal{N}(\boldsymbol{C}) \subseteq \mathcal{N}(\boldsymbol{D})$ and thus $\dim(\mathcal{N}((\boldsymbol{C}^{\frac{1}{2}})^{\dagger} \boldsymbol{D} (\boldsymbol{C}^{\frac{1}{2}})^{\dagger})) = \dim(\mathcal{N}(\boldsymbol{C}^{\dagger} \boldsymbol{D}))$.

We see that $(\boldsymbol{P}_{\mathcal{N}^\perp(\boldsymbol{H})} \otimes \boldsymbol{P}_{\mathcal{N}^\perp(\boldsymbol{H})})\boldsymbol{Q}$ is a sum of Kronecker products, where each product is a symmetric matrix multiplied by itself. This means that Theorem 6 applies to $(\boldsymbol{P}_{\mathcal{N}^\perp(\boldsymbol{H})} \otimes \boldsymbol{P}_{\mathcal{N}^\perp(\boldsymbol{H})})\boldsymbol{Q}$, and thus we have that $\left\|(\boldsymbol{P}_{\mathcal{N}^\perp(\boldsymbol{H})} \otimes \boldsymbol{P}_{\mathcal{N}^\perp(\boldsymbol{H})})\boldsymbol{Q}\right\| = \lambda_{\max}((\boldsymbol{P}_{\mathcal{N}^\perp(\boldsymbol{H})} \otimes \boldsymbol{P}_{\mathcal{N}^\perp(\boldsymbol{H})})\boldsymbol{Q})$. Moreover, it is easy to show that $\boldsymbol{P}_{\mathcal{N}^\perp(\boldsymbol{H})} \otimes \boldsymbol{P}_{\mathcal{N}^\perp(\boldsymbol{H})} = \boldsymbol{P}_{\mathcal{N}^\perp(\boldsymbol{D})}$, and $\boldsymbol{P}_{\mathcal{N}^\perp(\boldsymbol{D})}\boldsymbol{C} = \boldsymbol{C}\boldsymbol{P}_{\mathcal{N}^\perp(\boldsymbol{D})}$. Combining this with (79), we have

$$(\boldsymbol{P}_{\mathcal{N}^\perp(\boldsymbol{H})} \otimes \boldsymbol{P}_{\mathcal{N}^\perp(\boldsymbol{H})})\boldsymbol{Q} = \boldsymbol{P}_{\mathcal{N}^\perp(\boldsymbol{D})} - 2\eta\boldsymbol{C}\boldsymbol{P}_{\mathcal{N}^\perp(\boldsymbol{D})} + \eta^2\boldsymbol{D}. \tag{96}$$

Thus, for all $\boldsymbol{u} \in \mathcal{N}(\boldsymbol{D})$ we have

$$(\boldsymbol{P}_{\mathcal{N}^\perp(\boldsymbol{H})} \otimes \boldsymbol{P}_{\mathcal{N}^\perp(\boldsymbol{H})})\boldsymbol{Q}\boldsymbol{u} = \boldsymbol{P}_{\mathcal{N}^\perp(\boldsymbol{D})}\boldsymbol{u} - 2\eta\boldsymbol{C}\boldsymbol{P}_{\mathcal{N}^\perp(\boldsymbol{D})}\boldsymbol{u} + \eta^2\boldsymbol{D}\boldsymbol{u} = \boldsymbol{0}. \tag{97}$$

Since the eigenvectors of symmetric matrices are orthogonal, and $\mathcal{N}(\boldsymbol{D})$ is an eigenspace, we get that the top eigenvector of $(\boldsymbol{P}_{\mathcal{N}^\perp(\boldsymbol{H})} \otimes \boldsymbol{P}_{\mathcal{N}^\perp(\boldsymbol{H})})\boldsymbol{Q}$ should be in $\mathcal{N}^\perp(\boldsymbol{D})$. Now, for $\boldsymbol{u} \in \mathcal{N}^\perp(\boldsymbol{D}) \cap \mathbb{S}^{d^2-1}$

$$\boldsymbol{u}^\mathrm{T}(\boldsymbol{P}_{\mathcal{N}^\perp(\boldsymbol{H})} \otimes \boldsymbol{P}_{\mathcal{N}^\perp(\boldsymbol{H})})\boldsymbol{Q}\boldsymbol{u} = \boldsymbol{u}^\mathrm{T}\boldsymbol{P}_{\mathcal{N}^\perp(\boldsymbol{D})}\boldsymbol{u} - 2\eta\boldsymbol{u}^\mathrm{T}\boldsymbol{C}\boldsymbol{P}_{\mathcal{N}^\perp(\boldsymbol{D})}\boldsymbol{u} + \eta^2\boldsymbol{u}^\mathrm{T}\boldsymbol{D}\boldsymbol{u}$$
$$= 1 - 2\eta\boldsymbol{u}^\mathrm{T}\boldsymbol{C}\boldsymbol{u} + \eta^2\boldsymbol{u}^\mathrm{T}\boldsymbol{D}\boldsymbol{u}, \tag{98}$$

where in the second step we used the fact that $\boldsymbol{u} \in \mathcal{N}^\perp(\boldsymbol{D})$, and therefore $\boldsymbol{P}_{\mathcal{N}^\perp(\boldsymbol{D})}\boldsymbol{u} = \boldsymbol{u}$. Additionally, note that

$$\inf_{\boldsymbol{u} \in \mathbb{S}^{d^2-1} : \boldsymbol{u} \notin \mathcal{N}(\boldsymbol{D})} \left\{ \frac{\boldsymbol{u}^\mathrm{T}\boldsymbol{C}\boldsymbol{u}}{\boldsymbol{u}^\mathrm{T}\boldsymbol{D}\boldsymbol{u}} \right\} = \inf_{\boldsymbol{u} \in \mathbb{S}^{d^2-1} \cap \mathcal{N}^\perp(\boldsymbol{D})} \left\{ \frac{\boldsymbol{u}^\mathrm{T}\boldsymbol{C}\boldsymbol{u}}{\boldsymbol{u}^\mathrm{T}\boldsymbol{D}\boldsymbol{u}} \right\}. \tag{99}$$

Namely, having a component of $\boldsymbol{u}$ in $\mathcal{N}(\boldsymbol{D})$ can only be non-optimal, since the denominator is invariant to vectors in $\mathcal{N}(\boldsymbol{D})$, while the numerator can only increase ($\boldsymbol{C}$ is PSD). Now, assuming $\eta > 0$ we have from the derivation of $\eta^*_{\mathrm{var}}$ in the second statement (see (84))

$$\eta < \eta^*_{\mathrm{var}}$$
$$\Leftrightarrow \quad \eta < 2 \inf_{\boldsymbol{u} \in \mathbb{S}^{d^2-1} : \boldsymbol{u} \notin \mathcal{N}(\boldsymbol{D})} \left\{ \frac{\boldsymbol{u}^\mathrm{T}\boldsymbol{C}\boldsymbol{u}}{\boldsymbol{u}^\mathrm{T}\boldsymbol{D}\boldsymbol{u}} \right\}$$
$$\Leftrightarrow \quad \eta < 2 \inf_{\boldsymbol{u} \in \mathbb{S}^{d^2-1} \cap \mathcal{N}^\perp(\boldsymbol{D})} \left\{ \frac{\boldsymbol{u}^\mathrm{T}\boldsymbol{C}\boldsymbol{u}}{\boldsymbol{u}^\mathrm{T}\boldsymbol{D}\boldsymbol{u}} \right\}$$
$$\Leftrightarrow \quad \eta < 2\frac{\boldsymbol{u}^\mathrm{T}\boldsymbol{C}\boldsymbol{u}}{\boldsymbol{u}^\mathrm{T}\boldsymbol{D}\boldsymbol{u}} \quad \forall \boldsymbol{u} \in \mathbb{S}^{d^2-1} \cap \mathcal{N}^\perp(\boldsymbol{D})$$
$$\Leftrightarrow \quad \eta < 2\frac{\boldsymbol{u}^\mathrm{T}\boldsymbol{C}\boldsymbol{u}}{\boldsymbol{u}^\mathrm{T}\boldsymbol{D}\boldsymbol{u}} \quad \forall \boldsymbol{u} \in \mathbb{S}^{d^2-1} \cap \mathcal{N}^\perp(\boldsymbol{D})$$
$$\Leftrightarrow \quad \eta\boldsymbol{u}^\mathrm{T}\boldsymbol{D}\boldsymbol{u} < 2\boldsymbol{u}^\mathrm{T}\boldsymbol{C}\boldsymbol{u} \quad \forall \boldsymbol{u} \in \mathbb{S}^{d^2-1} \cap \mathcal{N}^\perp(\boldsymbol{D})$$
$$\Leftrightarrow \quad \eta^2\boldsymbol{u}^\mathrm{T}\boldsymbol{D}\boldsymbol{u} < 2\eta\boldsymbol{u}^\mathrm{T}\boldsymbol{C}\boldsymbol{u} \quad \forall \boldsymbol{u} \in \mathbb{S}^{d^2-1} \cap \mathcal{N}^\perp(\boldsymbol{D}) \quad (\eta > 0)$$
$$\Leftrightarrow \quad -2\eta\boldsymbol{u}^\mathrm{T}\boldsymbol{C}\boldsymbol{u} + \eta^2\boldsymbol{u}^\mathrm{T}\boldsymbol{D}\boldsymbol{u} < 0 \quad \forall \boldsymbol{u} \in \mathbb{S}^{d^2-1} \cap \mathcal{N}^\perp(\boldsymbol{D})$$
$$\Leftrightarrow \quad 1 - 2\eta\boldsymbol{u}^\mathrm{T}\boldsymbol{C}\boldsymbol{u} + \eta^2\boldsymbol{u}^\mathrm{T}\boldsymbol{D}\boldsymbol{u} < 1 \quad \forall \boldsymbol{u} \in \mathbb{S}^{d^2-1} \cap \mathcal{N}^\perp(\boldsymbol{D})$$
$$\Leftrightarrow \quad \boldsymbol{u}^\mathrm{T}\left(\boldsymbol{P}_{\mathcal{N}^\perp(\boldsymbol{H})} \otimes \boldsymbol{P}_{\mathcal{N}^\perp(\boldsymbol{H})}\right)\boldsymbol{Q}\boldsymbol{u} < 1 \quad \forall \boldsymbol{u} \in \mathbb{S}^{d^2-1} \cap \mathcal{N}^\perp(\boldsymbol{D})$$
$$\Leftrightarrow \quad \lambda_{\max}\left(\left(\boldsymbol{P}_{\mathcal{N}^\perp(\boldsymbol{H})} \otimes \boldsymbol{P}_{\mathcal{N}^\perp(\boldsymbol{H})}\right)\boldsymbol{Q}\right) < 1 \tag{100}$$

where in the penultimate step we used (98). Overall we have that $0 < \eta < \eta^*_{\mathrm{var}}$ if and only if $\lambda_{\max}\left((\boldsymbol{P}_{\mathcal{N}^\perp(\boldsymbol{H})} \otimes \boldsymbol{P}_{\mathcal{N}^\perp(\boldsymbol{H})})\boldsymbol{Q}\right) < 1$ (we will use this fact later on). Therefore, $\lambda_{\max}\left((\boldsymbol{P}_{\mathcal{N}^\perp(\boldsymbol{H})} \otimes \boldsymbol{P}_{\mathcal{N}^\perp(\boldsymbol{H})})\boldsymbol{Q}\right) = \left\|(\boldsymbol{P}_{\mathcal{N}^\perp(\boldsymbol{H})} \otimes \boldsymbol{P}_{\mathcal{N}^\perp(\boldsymbol{H})})\boldsymbol{Q}\right\| < 1$. Hence, from (94) we get

$$\lim_{t\to\infty} \|\mathrm{vec}\,(\boldsymbol{\Sigma}^\perp_t)\| \leq \left\|(\boldsymbol{P}_{\mathcal{N}^\perp(\boldsymbol{H})} \otimes \boldsymbol{P}_{\mathcal{N}^\perp(\boldsymbol{H})})\boldsymbol{Q}\right\|^t \|\mathrm{vec}\,(\boldsymbol{\Sigma}^\perp_0)\| = 0, \tag{101}$$

which proves the statement.

## B.8 Proof of Theorem 4

**First statement.** Let us start by proving the first statement. In (34) we showed that if the minimum is regular then

$$\boldsymbol{\theta}^\|_{t+1} - \boldsymbol{\theta}^{*\|} = \boldsymbol{\theta}^\|_t - \boldsymbol{\theta}^{*\|} - \frac{\eta}{B}\sum_{i\in\mathfrak{B}_t} \boldsymbol{g}^\|_i. \tag{102}$$

Let us compute the expected squared norm. We have

$$
\mathbb{E}\left[\left\|\boldsymbol{\theta}_{t+1}^{\|} - \boldsymbol{\theta}^{*\|}\right\|^2\right] = \mathbb{E}\left[\left\|\boldsymbol{\theta}_t^{\|} - \boldsymbol{\theta}^{*\|} - \frac{\eta}{B}\sum_{i\in\mathfrak{B}_t}\boldsymbol{g}_i^{\|}\right\|^2\right]
$$

$$
= \mathbb{E}\left[\left\|\boldsymbol{\theta}_t^{\|} - \boldsymbol{\theta}^{*\|}\right\|^2\right] + \mathbb{E}\left[\left\|\frac{\eta}{B}\sum_{i\in\mathfrak{B}_t}\boldsymbol{g}_i^{\|}\right\|^2\right] - 2\mathbb{E}\left[\left(\boldsymbol{\theta}_t^{\|} - \boldsymbol{\theta}^{*\|}\right)^{\mathrm{T}}\left(\frac{\eta}{B}\sum_{i\in\mathfrak{B}_t}\boldsymbol{g}_i^{\|}\right)\right]
$$

$$
= \mathbb{E}\left[\left\|\boldsymbol{\theta}_t^{\|} - \boldsymbol{\theta}^{*\|}\right\|^2\right] + \mathbb{E}\left[\left\|\frac{\eta}{B}\sum_{i\in\mathfrak{B}_t}\boldsymbol{g}_i^{\|}\right\|^2\right] - 2\mathbb{E}\left[\boldsymbol{\theta}_t^{\|} - \boldsymbol{\theta}^{*\|}\right]^{\mathrm{T}}\mathbb{E}\left[\frac{\eta}{B}\sum_{i\in\mathfrak{B}_t}\boldsymbol{g}_i^{\|}\right]
$$

$$
= \mathbb{E}\left[\left\|\boldsymbol{\theta}_t^{\|} - \boldsymbol{\theta}^{*\|}\right\|^2\right] + \mathbb{E}\left[\left\|\frac{\eta}{B}\sum_{i\in\mathfrak{B}_t}\boldsymbol{g}_i^{\|}\right\|^2\right], \tag{103}
$$

where in the second line we used the fact that $\boldsymbol{\theta}_t^{\|}$ is independent of $\mathfrak{B}_t$ and in the last line we used the fact that

$$
\mathbb{E}\left[\frac{\eta}{B}\sum_{i\in\mathfrak{B}_t}\boldsymbol{g}_i^{\|}\right] = \frac{\eta}{n}\sum_{i=1}^n \boldsymbol{g}_i^{\|} = \boldsymbol{P}_{\mathcal{N}(\boldsymbol{H})}\frac{\eta}{n}\sum_{i=1}^n \boldsymbol{g}_i = \boldsymbol{0}. \tag{104}
$$

Using (63) we have that (see definition in (45))

$$
\mathbb{E}\left[\left\|\frac{\eta}{B}\sum_{i\in\mathfrak{B}_t}\boldsymbol{g}_i^{\|}\right\|^2\right] = \mathbb{E}\left[\left\|\boldsymbol{P}_{\mathcal{N}(\boldsymbol{H})}\boldsymbol{v}_t\right\|^2\right]
$$

$$
= \mathrm{Tr}\left(\boldsymbol{P}_{\mathcal{N}(\boldsymbol{H})}\mathbb{E}\left[\boldsymbol{v}_t\boldsymbol{v}_t^{\mathrm{T}}\right]\boldsymbol{P}_{\mathcal{N}(\boldsymbol{H})}\right)
$$

$$
= \eta^2\frac{n-B}{B(n-1)}\frac{1}{n}\sum_{i=1}^n\mathrm{Tr}\left(\boldsymbol{P}_{\mathcal{N}(\boldsymbol{H})}\boldsymbol{g}_i\boldsymbol{g}_i^{\mathrm{T}}\boldsymbol{P}_{\mathcal{N}(\boldsymbol{H})}\right)
$$

$$
= \eta^2 p\frac{1}{n}\sum_{i=1}^n\left\|\boldsymbol{P}_{\mathcal{N}(\boldsymbol{H})}\boldsymbol{g}_i\right\|^2
$$

$$
= \eta^2 p\frac{1}{n}\sum_{i=1}^n\left\|\boldsymbol{g}_i^{\|}\right\|^2. \tag{105}
$$

Unrolling (103) we have that

$$
\mathbb{E}\left[\left\|\boldsymbol{\theta}_t^{\|} - \boldsymbol{\theta}^{*\|}\right\|^2\right] = \mathbb{E}\left[\left\|\boldsymbol{\theta}_0^{\|} - \boldsymbol{\theta}^{*\|}\right\|^2\right] + t\times\eta^2 p\frac{1}{n}\sum_{i=1}^n\left\|\boldsymbol{g}_i^{\|}\right\|^2. \tag{106}
$$

Thus, $\lim_{t\to\infty}\mathbb{E}[\|\boldsymbol{\theta}_t^{\|} - \boldsymbol{\theta}^{*\|}\|^2] = \infty$ if and only if $\sum_{i=1}^n\|\boldsymbol{g}_i^{\|}\|^2 > 0$.

**Second and third statements.** Next, we turn to prove the second and third statements of the theorem. In App. B.9 we show the following.

**Lemma 2.** *Assume that $\boldsymbol{\theta}^*$ is a twice differentiable regular minimum. Consider the linear dynamics of $\{\boldsymbol{\theta}_t\}$ from Def. 1.*

1. *If $\lambda_{\max}\left((\boldsymbol{P}_{\mathcal{N}^{\perp}(\boldsymbol{H})}\otimes\boldsymbol{P}_{\mathcal{N}^{\perp}(\boldsymbol{H})})\boldsymbol{Q}\right) < 1$ then $\limsup_{t\to\infty}\mathbb{E}[\|\boldsymbol{\theta}_t^{\perp} - \boldsymbol{\theta}^{*\perp}\|^2]$ is finite.*

2. *If $\limsup_{t\to\infty}\mathbb{E}[\|\boldsymbol{\theta}_t^{\perp} - \boldsymbol{\theta}^{*\perp}\|^2]$ is finite then $\lambda_{\max}\left((\boldsymbol{P}_{\mathcal{N}^{\perp}(\boldsymbol{H})}\otimes\boldsymbol{P}_{\mathcal{N}^{\perp}(\boldsymbol{H})})\boldsymbol{Q}\right) \leq 1$.*

3. *Let $\boldsymbol{z}_{\max}$ denote the top eigenvector of $(\boldsymbol{P}_{\mathcal{N}^{\perp}(\boldsymbol{H})}\otimes\boldsymbol{P}_{\mathcal{N}^{\perp}(\boldsymbol{H})})\boldsymbol{Q}$, and assume that $\boldsymbol{z}_{\max}^{\mathrm{T}}\mathrm{vec}(\boldsymbol{\Sigma}_{\boldsymbol{g}}^{\perp}) \neq 0$. If $\limsup_{t\to\infty}\mathbb{E}[\|\boldsymbol{\theta}_t^{\perp} - \boldsymbol{\theta}^{*\perp}\|^2]$ is finite then $\lambda_{\max}\left((\boldsymbol{P}_{\mathcal{N}^{\perp}(\boldsymbol{H})}\otimes\boldsymbol{P}_{\mathcal{N}^{\perp}(\boldsymbol{H})})\boldsymbol{Q}\right) < 1$.*

Moreover, in (100) we showed that $\lambda_{\max}\big((\boldsymbol{P}_{\mathcal{N}^\perp(\boldsymbol{H})} \otimes \boldsymbol{P}_{\mathcal{N}^\perp(\boldsymbol{H})})\boldsymbol{Q}\big) < 1$ if and only if $0 < \eta < \eta^*_{\text{var}}$, which proves the second and third statements. Note that under the mild assumption that $\boldsymbol{z}_{\max}^{\mathrm{T}}\text{vec}(\boldsymbol{\Sigma}_{\boldsymbol{g}}^\perp) \neq 0$ we get that $\limsup\limits_{t\to\infty}\mathbb{E}[\|\boldsymbol{\theta}_t^\perp - \boldsymbol{\theta}^{*\perp}\|^2]$ is finite if and only if $0 \leq \eta < \eta^*_{\text{var}}$.

## B.9 Proof of Lemma 2

**First statement.** Here we assume $\lambda_{\max}\big((\boldsymbol{P}_{\mathcal{N}^\perp(\boldsymbol{H})} \otimes \boldsymbol{P}_{\mathcal{N}^\perp(\boldsymbol{H})})\boldsymbol{Q}\big) < 1$, and show this implies that $\limsup\limits_{t\to\infty}\mathbb{E}[\|\boldsymbol{\theta}_t^\perp - \boldsymbol{\theta}^{*\perp}\|^2]$ is finite. The (projected) transition matrix that governs the dynamics of $\boldsymbol{\Sigma}_t^\perp$ and $\boldsymbol{\mu}_t^\perp$ in (53) is given by

$$\boldsymbol{\Xi} = \begin{pmatrix} \boldsymbol{P}_{\mathcal{N}^\perp(\boldsymbol{H})} - \eta\boldsymbol{H} & \boldsymbol{0} \\ -\big(\boldsymbol{P}_{\mathcal{N}^\perp(\boldsymbol{H})} \otimes \boldsymbol{P}_{\mathcal{N}^\perp(\boldsymbol{H})}\big)\left(\mathbb{E}\left[\boldsymbol{v}_t^\perp \otimes \boldsymbol{A}_t\right] + \mathbb{E}\left[\boldsymbol{A}_t \otimes \boldsymbol{v}_t^\perp\right]\right) & \big(\boldsymbol{P}_{\mathcal{N}^\perp(\boldsymbol{H})} \otimes \boldsymbol{P}_{\mathcal{N}^\perp(\boldsymbol{H})}\big)\boldsymbol{Q} \end{pmatrix}. \tag{107}$$

Since this matrix is a block lower triangular matrix, its eigenvalues are

$$\{\lambda_j(\boldsymbol{\Xi})\} = \{\lambda_j(\boldsymbol{P}_{\mathcal{N}^\perp(\boldsymbol{H})} - \eta\boldsymbol{H})\} \bigcup \{\lambda_j\big((\boldsymbol{P}_{\mathcal{N}^\perp(\boldsymbol{H})} \otimes \boldsymbol{P}_{\mathcal{N}^\perp(\boldsymbol{H})})\boldsymbol{Q}\big)\}. \tag{108}$$

Note that if $\rho\big((\boldsymbol{P}_{\mathcal{N}^\perp(\boldsymbol{H})} \otimes \boldsymbol{P}_{\mathcal{N}^\perp(\boldsymbol{H})})\boldsymbol{Q}\big) < 1$ then $\rho(\boldsymbol{P}_{\mathcal{N}^\perp(\boldsymbol{H})} - \eta\boldsymbol{H}) < 1$ (see App. B.9.1). Therefore, all the eigenvalues of $\boldsymbol{\Xi}$ are less than 1 in absolute value. Therefore, $\|\text{vec}(\boldsymbol{\Sigma}_t^\perp)\|_2 = \|\boldsymbol{\Sigma}_t^\perp\|_{\text{F}}$ is bounded. Since $\boldsymbol{\Sigma}_t^\perp$ is PSD we have

$$\|\boldsymbol{\Sigma}_t^\perp\|_{\text{F}} = \sqrt{\sum_{j=1}^d \lambda_j^2(\boldsymbol{\Sigma}_t^\perp)} \geq \frac{1}{\sqrt{d}}\sum_{j=1}^d \lambda_j(\boldsymbol{\Sigma}_t^\perp) = \frac{1}{\sqrt{d}}\text{Tr}(\boldsymbol{\Sigma}_t^\perp) = \frac{1}{\sqrt{d}}\mathbb{E}[\|\boldsymbol{\theta}_t^\perp - \boldsymbol{\theta}^{*\perp}\|^2]. \tag{109}$$

Therefore, $\mathbb{E}[\|\boldsymbol{\theta}_t^\perp - \boldsymbol{\theta}^{*\perp}\|^2]$ is bounded.

**Second statement.** Here we assume that $\limsup\limits_{t\to\infty}\mathbb{E}[\|\boldsymbol{\theta}_t^\perp - \boldsymbol{\theta}^{*\perp}\|^2]$ is finite, then we show $\lambda_{\max}\big((\boldsymbol{P}_{\mathcal{N}^\perp(\boldsymbol{H})} \otimes \boldsymbol{P}_{\mathcal{N}^\perp(\boldsymbol{H})})\boldsymbol{Q}\big) \leq 1$. The matrix $\big(\boldsymbol{P}_{\mathcal{N}^\perp(\boldsymbol{H})} \otimes \boldsymbol{P}_{\mathcal{N}^\perp(\boldsymbol{H})}\big)\boldsymbol{Q}$ can be written as

$$(\boldsymbol{P}_{\mathcal{N}^\perp(\boldsymbol{H})} \otimes \boldsymbol{P}_{\mathcal{N}^\perp(\boldsymbol{H})})\boldsymbol{Q} = \big(\boldsymbol{P}_{\mathcal{N}^\perp(\boldsymbol{H})} - \eta\boldsymbol{H}\big) \otimes \big(\boldsymbol{P}_{\mathcal{N}^\perp(\boldsymbol{H})} - \eta\boldsymbol{H}\big)$$
$$+ \eta^2 p\left(\frac{1}{n}\sum_{i=1}^n (\boldsymbol{H}_i - \boldsymbol{H}) \otimes (\boldsymbol{H}_i - \boldsymbol{H})\right). \tag{110}$$

This expression is a sum of Kronecker products, where each product is a symmetric matrix with itself. Therefore, according to Theorem 6, we get $\lambda_{\max}\big((\boldsymbol{P}_{\mathcal{N}^\perp(\boldsymbol{H})} \otimes \boldsymbol{P}_{\mathcal{N}^\perp(\boldsymbol{H})})\boldsymbol{Q}\big) = \rho\big((\boldsymbol{P}_{\mathcal{N}^\perp(\boldsymbol{H})} \otimes \boldsymbol{P}_{\mathcal{N}^\perp(\boldsymbol{H})})\boldsymbol{Q}\big)$ and $\boldsymbol{Z}_{\max} = \text{vec}^{-1}(\boldsymbol{z}_{\max})$ is a PSD matrix, where $\boldsymbol{z}_{\max}$ is a normalized top eigenvector of $\big(\boldsymbol{P}_{\mathcal{N}^\perp(\boldsymbol{H})} \otimes \boldsymbol{P}_{\mathcal{N}^\perp(\boldsymbol{H})}\big)\boldsymbol{Q}$. Now, set[4] $\boldsymbol{\Sigma}_0^\perp = \boldsymbol{Z}_{\max}$ and $\boldsymbol{\mu}_0 = \boldsymbol{0}$, then in this case $\boldsymbol{\mu}_t^\perp = (\boldsymbol{P}_{\mathcal{N}^\perp(\boldsymbol{H})} - \eta\boldsymbol{H})^t\boldsymbol{\mu}_0^\perp = \boldsymbol{0}$ for all $t > 0$. Therefore, from (53)

$$\text{vec}(\boldsymbol{\Sigma}_{t+1}) = (\boldsymbol{P}_{\mathcal{N}^\perp(\boldsymbol{H})} \otimes \boldsymbol{P}_{\mathcal{N}^\perp(\boldsymbol{H})})\boldsymbol{Q}\text{vec}(\boldsymbol{\Sigma}_t^\perp) + \text{vec}(\boldsymbol{\Sigma}_{\boldsymbol{v}}^\perp). \tag{111}$$

Thus,

$$\boldsymbol{z}_{\max}^{\mathrm{T}}\text{vec}(\boldsymbol{\Sigma}_{t+1}^\perp) = \boldsymbol{z}_{\max}^{\mathrm{T}}(\boldsymbol{P}_{\mathcal{N}^\perp(\boldsymbol{H})} \otimes \boldsymbol{P}_{\mathcal{N}^\perp(\boldsymbol{H})})\boldsymbol{Q}\text{vec}(\boldsymbol{\Sigma}_t^\perp) + \boldsymbol{z}_{\max}^{\mathrm{T}}\text{vec}(\boldsymbol{\Sigma}_{\boldsymbol{v}}^\perp)$$
$$= \lambda_{\max}\big((\boldsymbol{P}_{\mathcal{N}^\perp(\boldsymbol{H})} \otimes \boldsymbol{P}_{\mathcal{N}^\perp(\boldsymbol{H})})\boldsymbol{Q}\big)\boldsymbol{z}_{\max}^{\mathrm{T}}\text{vec}(\boldsymbol{\Sigma}_t^\perp) + \text{Tr}(\boldsymbol{Z}_{\max}\boldsymbol{\Sigma}_{\boldsymbol{v}}^\perp)$$
$$\geq \lambda_{\max}\big((\boldsymbol{P}_{\mathcal{N}^\perp(\boldsymbol{H})} \otimes \boldsymbol{P}_{\mathcal{N}^\perp(\boldsymbol{H})})\boldsymbol{Q}\big)\boldsymbol{z}_{\max}^{\mathrm{T}}\text{vec}(\boldsymbol{\Sigma}_t^\perp), \tag{112}$$

where we used the fact that $\boldsymbol{Z}_{\max}^{\frac{1}{2}}\boldsymbol{\Sigma}_{\boldsymbol{v}}^\perp\boldsymbol{Z}_{\max}^{\frac{1}{2}}$ is a PSD matrix, and thus

$$\text{Tr}(\boldsymbol{Z}_{\max}\boldsymbol{\Sigma}_{\boldsymbol{v}}^\perp) = \text{Tr}\left(\boldsymbol{Z}_{\max}^{\frac{1}{2}}\boldsymbol{\Sigma}_{\boldsymbol{v}}^\perp\boldsymbol{Z}_{\max}^{\frac{1}{2}}\right) \geq 0. \tag{113}$$

---

[4] $\boldsymbol{Z}_{\max} \in \mathcal{N}^\perp(\boldsymbol{D})$, *i.e.*, $\boldsymbol{P}_{\mathcal{N}^\perp(\boldsymbol{H})}\boldsymbol{Z}_{\max}\boldsymbol{P}_{\mathcal{N}^\perp(\boldsymbol{H})} = \boldsymbol{Z}_{\max}$, and therefore this initialization is possible (see discussion below (97)).

Therefore,

$$\boldsymbol{z}_{\max}^{\mathrm{T}}\mathrm{vec}\left(\boldsymbol{\Sigma}_t^{\perp}\right) \geq \lambda_{\max}^t\left((\boldsymbol{P}_{\mathcal{N}^{\perp}(\boldsymbol{H})} \otimes \boldsymbol{P}_{\mathcal{N}^{\perp}(\boldsymbol{H})})\boldsymbol{Q}\right)\boldsymbol{z}_{\max}^{\mathrm{T}}\mathrm{vec}\left(\boldsymbol{\Sigma}_0\right)$$
$$= \lambda_{\max}^t\left((\boldsymbol{P}_{\mathcal{N}^{\perp}(\boldsymbol{H})} \otimes \boldsymbol{P}_{\mathcal{N}^{\perp}(\boldsymbol{H})})\boldsymbol{Q}\right), \tag{114}$$

where in the last step we used $\boldsymbol{\Sigma}_0 = \boldsymbol{Z}_{\max}$ and $\|\boldsymbol{Z}_{\max}\|_{\mathrm{F}} = 1$. Additionally, for all $t > 0$

$$\boldsymbol{z}_{\max}^{\mathrm{T}}\mathrm{vec}\left(\boldsymbol{\Sigma}_t^{\perp}\right) \leq \|\boldsymbol{z}_{\max}\|_2 \, \|\mathrm{vec}\left(\boldsymbol{\Sigma}_t^{\perp}\right)\|_2$$
$$= \|\boldsymbol{\Sigma}_t^{\perp}\|_{\mathrm{F}}$$
$$= \sqrt{\sum_{j=1}^d \lambda_j^2(\boldsymbol{\Sigma}_t^{\perp})}$$
$$\leq \sum_{j=1}^d \lambda_j(\boldsymbol{\Sigma}_t^{\perp})$$
$$= \mathrm{Tr}(\boldsymbol{\Sigma}_t^{\perp})$$
$$= \mathbb{E}[\|\boldsymbol{\theta}_t^{\perp} - \boldsymbol{\theta}^{*\perp}\|^2]. \tag{115}$$

Overall, combining (114) and (115) results with

$$\lambda_{\max}^t\left((\boldsymbol{P}_{\mathcal{N}^{\perp}(\boldsymbol{H})} \otimes \boldsymbol{P}_{\mathcal{N}^{\perp}(\boldsymbol{H})})\boldsymbol{Q}\right) \leq \mathbb{E}[\|\boldsymbol{\theta}_t^{\perp} - \boldsymbol{\theta}^{*\perp}\|^2]. \tag{116}$$

Since $\mathbb{E}[\|\boldsymbol{\theta}_t^{\perp} - \boldsymbol{\theta}^{*\perp}\|^2]$ is bounded then $\lambda_{\max}\left((\boldsymbol{P}_{\mathcal{N}^{\perp}(\boldsymbol{H})} \otimes \boldsymbol{P}_{\mathcal{N}^{\perp}(\boldsymbol{H})})\boldsymbol{Q}\right) \leq 1$.

**Third statement.** Furthermore, if $\boldsymbol{z}_{\max}^{\mathrm{T}}\mathrm{vec}\left(\boldsymbol{\Sigma}_{\boldsymbol{v}}^{\perp}\right) \neq 0$ we get from (113)

$$\boldsymbol{z}_{\max}^{\mathrm{T}}\mathrm{vec}\left(\boldsymbol{\Sigma}_{\boldsymbol{v}}^{\perp}\right) > 0. \tag{117}$$

Assume by contradiction that $\lambda_{\max}\left((\boldsymbol{P}_{\mathcal{N}^{\perp}(\boldsymbol{H})} \otimes \boldsymbol{P}_{\mathcal{N}^{\perp}(\boldsymbol{H})})\boldsymbol{Q}\right) = 1$, then (112) gives

$$\boldsymbol{z}_{\max}^{\mathrm{T}}\mathrm{vec}\left(\boldsymbol{\Sigma}_{t+1}^{\perp}\right) = \boldsymbol{z}_{\max}^{\mathrm{T}}\mathrm{vec}\left(\boldsymbol{\Sigma}_t^{\perp}\right) + \boldsymbol{z}_{\max}^{\mathrm{T}}\mathrm{vec}\left(\boldsymbol{\Sigma}_{\boldsymbol{v}}^{\perp}\right). \tag{118}$$

Unrolling this equation gives $\boldsymbol{z}_{\max}^{\mathrm{T}}\mathrm{vec}\left(\boldsymbol{\Sigma}_t^{\perp}\right) = t\boldsymbol{z}_{\max}^{\mathrm{T}}\mathrm{vec}\left(\boldsymbol{\Sigma}_{\boldsymbol{v}}^{\perp}\right)$. Then, by (115) we get

$$\mathbb{E}[\|\boldsymbol{\theta}_t - \boldsymbol{\theta}^*\|^2] \geq \boldsymbol{z}_{\max}^{\mathrm{T}}\mathrm{vec}\left(\boldsymbol{\Sigma}_t^{\perp}\right) = t\boldsymbol{z}_{\max}^{\mathrm{T}}\mathrm{vec}\left(\boldsymbol{\Sigma}_{\boldsymbol{v}}^{\perp}\right). \tag{119}$$

Since $\boldsymbol{z}_{\max}^{\mathrm{T}}\mathrm{vec}\left(\boldsymbol{\Sigma}_{\boldsymbol{v}}^{\perp}\right) > 0$, then $\mathbb{E}[\|\boldsymbol{\theta}_t - \boldsymbol{\theta}^*\|^2] \to \infty$ and we have a contradiction. Therefore $\lambda_{\max}\left((\boldsymbol{P}_{\mathcal{N}^{\perp}(\boldsymbol{H})} \otimes \boldsymbol{P}_{\mathcal{N}^{\perp}(\boldsymbol{H})})\boldsymbol{Q}\right) < 1$.

### B.9.1 $\rho(\boldsymbol{Q}) < 1$ IMPLIES $\rho(\boldsymbol{I} - \eta\boldsymbol{H}) < 1$

The matrix $(\boldsymbol{P}_{\mathcal{N}^{\perp}(\boldsymbol{H})} \otimes \boldsymbol{P}_{\mathcal{N}^{\perp}(\boldsymbol{H})})\boldsymbol{Q}$ can be written as

$$(\boldsymbol{P}_{\mathcal{N}^{\perp}(\boldsymbol{H})} \otimes \boldsymbol{P}_{\mathcal{N}^{\perp}(\boldsymbol{H})})\boldsymbol{Q} = \left(\boldsymbol{P}_{\mathcal{N}^{\perp}(\boldsymbol{H})} - \eta\boldsymbol{H}\right) \otimes \left(\boldsymbol{P}_{\mathcal{N}^{\perp}(\boldsymbol{H})} - \eta\boldsymbol{H}\right)$$
$$+ \eta^2 p\left(\frac{1}{n}\sum_{i=1}^n (\boldsymbol{H}_i - \boldsymbol{H}) \otimes (\boldsymbol{H}_i - \boldsymbol{H})\right). \tag{120}$$

Note that $\boldsymbol{P}_{\mathcal{N}^{\perp}(\boldsymbol{H})} - \eta\boldsymbol{H}$ is a symmetric matrix, and thus its principal eigenvector $\tilde{\boldsymbol{v}} \in \mathbb{S}^{d-1}$ satisfies $\rho(\boldsymbol{P}_{\mathcal{N}^{\perp}(\boldsymbol{H})} - \eta\boldsymbol{H}) = |\tilde{\boldsymbol{v}}^{\mathrm{T}}(\boldsymbol{P}_{\mathcal{N}^{\perp}(\boldsymbol{H})} - \eta\boldsymbol{H})\tilde{\boldsymbol{v}}|$. Additionally, $\|\tilde{\boldsymbol{v}} \otimes \tilde{\boldsymbol{v}}\| = \|\tilde{\boldsymbol{v}}\tilde{\boldsymbol{v}}^{\mathrm{T}}\|_{\mathrm{F}} = 1$, *i.e.*, $\tilde{\boldsymbol{v}} \otimes \tilde{\boldsymbol{v}} \in \mathbb{S}^{d^2-1}$. Now, assume that the spectral radius $\rho\left((\boldsymbol{P}_{\mathcal{N}^{\perp}(\boldsymbol{H})} \otimes \boldsymbol{P}_{\mathcal{N}^{\perp}(\boldsymbol{H})})\boldsymbol{Q}\right) < 1$ then

$$1 > \rho\left((\boldsymbol{P}_{\mathcal{N}^{\perp}(\boldsymbol{H})} \otimes \boldsymbol{P}_{\mathcal{N}^{\perp}(\boldsymbol{H})})\boldsymbol{Q}\right)$$
$$\geq \left|[\tilde{\boldsymbol{v}} \otimes \tilde{\boldsymbol{v}}]^{\mathrm{T}}\left(\boldsymbol{P}_{\mathcal{N}^{\perp}(\boldsymbol{H})} \otimes \boldsymbol{P}_{\mathcal{N}^{\perp}(\boldsymbol{H})}\right)\boldsymbol{Q}[\tilde{\boldsymbol{v}} \otimes \tilde{\boldsymbol{v}}]\right|$$
$$= \left(\tilde{\boldsymbol{v}}^{\mathrm{T}}\left(\boldsymbol{P}_{\mathcal{N}^{\perp}(\boldsymbol{H})} - \eta\boldsymbol{H}\right)\tilde{\boldsymbol{v}}\right)^2 + \eta^2 p\frac{1}{n}\sum_{i=1}^n \left(\tilde{\boldsymbol{v}}^{\mathrm{T}}\left(\boldsymbol{H}_i - \boldsymbol{H}\right)\tilde{\boldsymbol{v}}\right)^2$$
$$= \rho^2(\boldsymbol{P}_{\mathcal{N}^{\perp}(\boldsymbol{H})} - \eta\boldsymbol{H}) + \eta^2 p\frac{1}{n}\sum_{i=1}^n \left(\tilde{\boldsymbol{v}}^{\mathrm{T}}\left(\boldsymbol{H}_i - \boldsymbol{H}\right)\tilde{\boldsymbol{v}}\right)^2$$
$$\geq \rho^2(\boldsymbol{P}_{\mathcal{N}^{\perp}(\boldsymbol{H})} - \eta\boldsymbol{H}). \tag{121}$$

## C  PROOF OF THEOREM 6

**First statement.**  Let $z \in \mathbb{R}^{d^2}$ be an eigenvector of $Q$, then we can look at its matrix form $Z = \mathrm{vec}^{-1}(z)$, where $Z \in \mathbb{R}^{d \times d}$. First, we show that $Q$ always has a set of eigenvectors that correspond to either symmetric or skew-symmetric matrices $\{Z_j\}$ (see proof in App. C.1).

**Proposition 4.** *Let $\{A_i\}$ be symmetric matrices over $\mathbb{R}^{d \times d}$, and define*

$$Q = \sum_{i=1}^{M} A_i \otimes A_i. \tag{122}$$

*Then there exists a set of eigenvectors $\{z_j\}$ such that each $Z_j = \mathrm{vec}^{-1}(z_j)$ is either a symmetric or a skew-symmetric matrix.*

**Second and third statement.**  Our next step is to bring the matrices $\{Z_j\}$ to a normal (canonical) form. Here we assume without loss of generality that the eigenvectors are normalized, *i.e.,* $\|Z_j\|_\mathrm{F} = 1$. For symmetric matrix $Z$, we have the spectral decomposition theorem, and thus $Z = V S V^\mathrm{T}$, where $V$ is an orthogonal matrix and $S$ is diagonal. We can also bring a skew-symmetric matrix to a similar form, but with $S$ a block diagonal matrix, with $\lfloor d/2 \rfloor$ blocks of size $2 \times 2$. Specifically, these blocks are in the form of (Zumino, 1962)

$$\begin{bmatrix} 0 & s_i \\ -s_i & 0 \end{bmatrix}. \tag{123}$$

If the dimension $d$ is odd, then $S$ has a row and column at the end filed with zeros. For numerical purposes, this normal (canonical) form can be computed by using the real Schur decomposition. Now, for symmetric matrices, we define the vector $s_\mathrm{sym} \in \mathbb{R}^d$ to be the diagonal of $S$, and for skew-symmetric matrices we define $s_\mathrm{skew} \in \mathbb{R}^{\lfloor d/2 \rfloor}$ to be $[s_1, s_2, \cdots, s_{\lfloor d/2 \rfloor}]^\mathrm{T}$. In App. C.2 we show that for symmetric matrix $Z$, the corresponding vector form $z$ satisfies

$$z^\mathrm{T} Q z = s_\mathrm{sym}^\mathrm{T} \sum_{i=1}^{M} M_i^{\odot 2} s_\mathrm{sym}, \tag{124}$$

where $M_i = V^\mathrm{T} A_i V$, the superscript $^{\odot k}$ denotes the Hadamard power and $\|s_\mathrm{skew}\| = 1$. For skew-symmetric matrices, we define for $1 \le \ell, p \le \lfloor d/2 \rfloor$

$$T_{i\,[\ell,p]} = M_{i\,[2\ell-1,2p-1]} M_{i\,[2\ell,2p]} - M_{i\,[2\ell-1,2p]} M_{i\,[2\ell-1,2p]}. \tag{125}$$

Namely, $T_i \in \mathbb{R}^{\lfloor d/2 \rfloor \times \lfloor d/2 \rfloor}$ is the determinant of each $2 \times 2$ block of $M_i$ without overlap. We show in App. C.2 that for skew-symmetric matrix $Z$, the corresponding vector form $z$ satisfies

$$z^\mathrm{T} Q z = 2 s_\mathrm{skew}^\mathrm{T} \sum_{i=1}^{M} T_i s_\mathrm{skew}, \tag{126}$$

where $\|s_\mathrm{skew}\| = 1/\sqrt{2}$. Let us define the projection matrix $P \in \mathbb{R}^{\lfloor d/2 \rfloor \times d}$ as

$$P = \frac{1}{\sqrt{2}} \begin{bmatrix} 1 & 1 & 0 & 0 & 0 & \cdots & 0 & 0 \\ 0 & 0 & 1 & 1 & 0 & \cdots & 0 & 0 \\ \vdots & & & \ddots & & & & \vdots \\ 0 & 0 & 0 & 0 & 0 & \cdots & 1 & 1 \end{bmatrix}. \tag{127}$$

This matrix is semi-orthogonal, *i.e.,* it satisfies $P P^\mathrm{T} = I$. Note that

$$\left[ P M_i^{\odot 2} P^\mathrm{T} \right]_{[\ell,p]} = \frac{1}{2} \left( M_{i\,[2\ell-1,2p-1]}^2 + M_{i\,[2\ell-1,2p]}^2 + M_{i\,[2\ell-1,2p]}^2 + M_{i\,[2\ell,2p]}^2 \right). \tag{128}$$

Therefore, for all $\ell, p$ and $i$ we have

$$\left| T_{i\,[\ell,p]} \right| \le \left[ P M_i^{\odot 2} P^\mathrm{T} \right]_{[\ell,p]}. \tag{129}$$

Now, given orthogonal matrix $\boldsymbol{V}$, for all $\boldsymbol{s}_{\mathrm{skew}} \in \mathbb{R}^{\lfloor d/2 \rfloor}$ such that $\|\boldsymbol{s}_{\mathrm{skew}}\| = 1/\sqrt{2}$, we set $\boldsymbol{u} = \sqrt{2}\boldsymbol{s}_{\mathrm{skew}}$ and then

$$
\left| 2\boldsymbol{s}_{\mathrm{skew}}^{\mathrm{T}} \sum_{i=1}^{M} \boldsymbol{T}_i \boldsymbol{s}_{\mathrm{skew}} \right| = \left| \boldsymbol{u}^{\mathrm{T}} \sum_{i=1}^{M} \boldsymbol{T}_i \boldsymbol{u} \right|
$$

$$
\leq \sum_{\ell=1}^{d} \sum_{p=1}^{d} \sum_{i=1}^{M} |\boldsymbol{u}_\ell| \left| \boldsymbol{T}_{i\,[\ell,p]} \right| |\boldsymbol{u}_p|
$$

$$
\leq \sum_{\ell=1}^{d} \sum_{p=1}^{d} \sum_{i=1}^{M} |\boldsymbol{u}_\ell| \left[ \boldsymbol{P}\boldsymbol{M}_i^{\odot 2}\boldsymbol{P}^{\mathrm{T}} \right]_{[\ell,p]} |\boldsymbol{u}_p|
$$

$$
\leq \lambda_{\max} \left( \sum_{i=1}^{M} \boldsymbol{P}\boldsymbol{M}_i^{\odot 2}\boldsymbol{P}^{\mathrm{T}} \right)
$$

$$
= \lambda_{\max} \left( \boldsymbol{P} \left( \sum_{i=1}^{M} \boldsymbol{M}_i^{\odot 2} \right) \boldsymbol{P}^{\mathrm{T}} \right)
$$

$$
\leq \lambda_{\max} \left( \sum_{i=1}^{M} \boldsymbol{M}_i^{\odot 2} \right), \tag{130}
$$

where in the last step we used the Cauchy interlacing theorem (a.k.a. Poincaré separation theorem). Thus, for every $\boldsymbol{s}_{\mathrm{skew}} \in \mathbb{R}^{\lfloor d/2 \rfloor}$ such that $\|\boldsymbol{s}_{\mathrm{skew}}\| = 1/\sqrt{2}$, which corresponds to a skew-symmetric matrix $\boldsymbol{z}_{\mathrm{skew}}$, we can always find $\boldsymbol{s}_{\mathrm{sym}} \in \mathbb{S}^{d-1}$ (the top eigenvector of $\sum_{i=1}^{M} \boldsymbol{M}_i^{\odot 2}$) which corresponds to a symmetric matrix $\boldsymbol{z}_{\mathrm{sym}}$ with the same basis $\boldsymbol{V}$ such that

$$
\left| \boldsymbol{z}_{\mathrm{skew}}^{\mathrm{T}} \boldsymbol{Q} \boldsymbol{z}_{\mathrm{skew}} \right| \leq \boldsymbol{z}_{\mathrm{sym}}^{\mathrm{T}} \boldsymbol{Q} \boldsymbol{z}_{\mathrm{sym}}. \tag{131}
$$

Therefore, the principal eigenvector of $\boldsymbol{Q}$ corresponds to a symmetric matrix rather than a skew-symmetric one. Denoted the principal eigenvector of $\boldsymbol{Q}$ by $\tilde{\boldsymbol{z}}$ and let $\tilde{\boldsymbol{Z}} = \tilde{\boldsymbol{V}}\tilde{\boldsymbol{S}}\tilde{\boldsymbol{V}}^{\mathrm{T}}$ be its spectral decomposition. Set

$$
\boldsymbol{\Psi} = \sum_{i=1}^{M} \tilde{\boldsymbol{M}}_i^{\odot 2}, \qquad \text{s.t.} \qquad \boldsymbol{M}_i = \tilde{\boldsymbol{V}}^{\mathrm{T}} \boldsymbol{A}_i \tilde{\boldsymbol{V}}. \tag{132}
$$

Since $\boldsymbol{Q}$ is symmetric, then by the spectral theorem we have that all its eigenvectors and eigenvalues are real, and they are given by the quadratic form using the corresponding eigenvectors. Thus,

$$
\rho(\boldsymbol{Q}) = \left| \tilde{\boldsymbol{z}}^{\mathrm{T}} \boldsymbol{Q} \tilde{\boldsymbol{z}} \right|
$$

$$
= \left| \tilde{\boldsymbol{s}}_{\mathrm{sym}}^{\mathrm{T}} \sum_{i=1}^{M} \tilde{\boldsymbol{M}}_i^{\odot 2} \tilde{\boldsymbol{s}}_{\mathrm{sym}} \right|
$$

$$
= \left| \tilde{\boldsymbol{s}}_{\mathrm{sym}}^{\mathrm{T}} \boldsymbol{\Psi} \tilde{\boldsymbol{s}}_{\mathrm{sym}} \right|
$$

$$
= \left| \sum_{i=1}^{d} \sum_{j=1}^{d} \tilde{\boldsymbol{s}}_{\mathrm{sym},i} \boldsymbol{\Psi}_{[i,j]} \tilde{\boldsymbol{s}}_{\mathrm{sym},j} \right|
$$

$$
\leq \sum_{i=1}^{d} \sum_{j=1}^{d} |\tilde{\boldsymbol{s}}_{\mathrm{sym},i}| \, \boldsymbol{\Psi}_{[i,j]} \, |\tilde{\boldsymbol{s}}_{\mathrm{sym},j}|
$$

$$
= |\tilde{\boldsymbol{s}}_{\mathrm{sym}}|^{\mathrm{T}} \boldsymbol{\Psi} \, |\tilde{\boldsymbol{s}}_{\mathrm{sym}}|
$$

$$
= \hat{\boldsymbol{z}}^{\mathrm{T}} \boldsymbol{Q} \hat{\boldsymbol{z}}, \tag{133}
$$

where $\hat{\boldsymbol{z}} = \mathrm{vec}(\tilde{\boldsymbol{V}}|\tilde{\boldsymbol{S}}|\tilde{\boldsymbol{V}}^{\mathrm{T}})$. Namely, the vector $\hat{\boldsymbol{z}}$ that corresponds to the matrix built by the element-wise absolute value of the spectrum of $\tilde{\boldsymbol{Z}}$ yields a greater or equal result than the spectral radius of $\boldsymbol{Q}$, while still having a unit Euclidean norm. Thus, if $\tilde{\boldsymbol{s}}_{\mathrm{sym}} \neq |\tilde{\boldsymbol{s}}_{\mathrm{sym}}|$ then both $\tilde{\boldsymbol{z}}$ and $\hat{\boldsymbol{z}}$ are principal eigenvectors (or else we get a contradiction). Note that $\mathrm{vec}^{-1}(\hat{\boldsymbol{z}})$ is in fact a PSD matrix. Therefore,

there is always a principal eigenvector for $\boldsymbol{Q}$ which corresponds to a PSD matrix. Additionally, since $\max_j |\lambda_j(\boldsymbol{Q})| = \rho(\boldsymbol{Q}) = \hat{z}^{\mathrm{T}} \boldsymbol{Q} \hat{z}$, then $\hat{z}$ is also a top eigenvector which corresponds to $\lambda_{\max}(\boldsymbol{Q})$, *i.e.*, $\rho(\boldsymbol{Q}) = \lambda_{\max}(\boldsymbol{Q})$.

### C.1  PROOF OF PROPOSITION 4

Let $(\lambda, z)$ be an eigenpair of $\boldsymbol{Q}$, *i.e.*, $\boldsymbol{Q}z = \lambda z$. Set $\boldsymbol{Z} = \mathrm{vec}^{-1}(z)$, then $\boldsymbol{Z}$ satisfies

$$
\begin{aligned}
\lambda \boldsymbol{Z} &= \mathrm{vec}^{-1}(\lambda z) \\
&= \mathrm{vec}^{-1}(\boldsymbol{Q}z) \\
&= \mathrm{vec}^{-1}\left(\sum_{i=1}^{M} \boldsymbol{A}_i \otimes \boldsymbol{A}_i z\right) \\
&= \sum_{i=1}^{M} \mathrm{vec}^{-1}\left(\boldsymbol{A}_i \otimes \boldsymbol{A}_i z\right) \\
&= \sum_{i=1}^{M} \boldsymbol{A}_i \boldsymbol{Z} \boldsymbol{A}_i^{\mathrm{T}} \\
&= \sum_{i=1}^{M} \boldsymbol{A}_i \boldsymbol{Z} \boldsymbol{A}_i.
\end{aligned}
\tag{134}
$$

Now, by taking a transpose on both ends of this equation we have

$$
\begin{aligned}
\lambda \boldsymbol{Z}^{\mathrm{T}} &= \sum_{i=1}^{M} \boldsymbol{A}^{\mathrm{T}} \boldsymbol{Z}^{\mathrm{T}} \boldsymbol{A}_i^{\mathrm{T}} \\
&= \sum_{i=1}^{M} \boldsymbol{A}_i \boldsymbol{Z}^{\mathrm{T}} \boldsymbol{A},
\end{aligned}
\tag{135}
$$

where in the last step we used the fact that $\{\boldsymbol{A}_i\}$ are symmetric. Thus, we have that $\mathrm{vec}(\boldsymbol{Z}^{\mathrm{T}})$ is also an eigenvector of $\boldsymbol{Q}$. If $\lambda$ has multiplicity one, then it must be that $\boldsymbol{Z}^{\mathrm{T}} = \pm \boldsymbol{Z}$, *i.e.*, symmetric or skew-symmetric matrix. If the multiplicity is greater than one and $\boldsymbol{Z}^{\mathrm{T}} \neq \pm \boldsymbol{Z}$, then any linear combination between $\boldsymbol{Z}$ and $\boldsymbol{Z}^{\mathrm{T}}$ will also be an eigenvector corresponding to $\lambda$. Particularly,

$$
\hat{\boldsymbol{Z}}_1 = \frac{1}{2}\left(\boldsymbol{Z} + \boldsymbol{Z}^{\mathrm{T}}\right) \qquad \hat{\boldsymbol{Z}}_2 = \frac{1}{2}\left(\boldsymbol{Z} - \boldsymbol{Z}^{\mathrm{T}}\right).
\tag{136}
$$

Now, $\hat{\boldsymbol{Z}}_1$ and $\hat{\boldsymbol{Z}}_2$ are symmetric and skew-symmetric eigenvectors, which correspond to $\lambda$. This procedure can be repeated while projecting the next eigenvectors of $\lambda$ onto the orthogonal complement of the already found vectors, until we find all the eigenvectors of $\lambda$. In this way, we can always find a set of eigenvectors that is comprised solely of symmetric and skew-symmetric vectors.

### C.2  QUADRATIC FORM CALCULATION FOR SYMMETRIC AND SKEW-SYMMETRIC MATRICES

Let $z = \mathrm{vec}(\boldsymbol{Z})$, and assume that $\boldsymbol{Z} = \boldsymbol{V} \boldsymbol{S} \boldsymbol{V}^{\mathrm{T}}$ where $\boldsymbol{V}$ is orthogonal matrix. Then

$$
\begin{aligned}
z^{\mathrm{T}} \boldsymbol{Q} z &= \left[\mathrm{vec}\left(\boldsymbol{V} \boldsymbol{S} \boldsymbol{V}^{\mathrm{T}}\right)\right]^{\mathrm{T}} \boldsymbol{Q}\, \mathrm{vec}\left(\boldsymbol{V} \boldsymbol{S} \boldsymbol{V}^{\mathrm{T}}\right) \\
&= \left[(\boldsymbol{V} \otimes \boldsymbol{V})\, \mathrm{vec}\left(\boldsymbol{S}\right)\right]^{\mathrm{T}} \boldsymbol{Q}\,(\boldsymbol{V} \otimes \boldsymbol{V})\, \mathrm{vec}\left(\boldsymbol{S}\right) \\
&= \left[\mathrm{vec}\left(\boldsymbol{S}\right)\right]^{\mathrm{T}}\left(\boldsymbol{V}^{\mathrm{T}} \otimes \boldsymbol{V}^{\mathrm{T}}\right) \sum_{i=1}^{M} \boldsymbol{A}_i \otimes \boldsymbol{A}_i\left(\boldsymbol{V} \otimes \boldsymbol{V}\right) \mathrm{vec}\left(\boldsymbol{S}\right) \\
&= \left[\mathrm{vec}\left(\boldsymbol{S}\right)\right]^{\mathrm{T}} \sum_{i=1}^{M}\left(\boldsymbol{V}^{\mathrm{T}} \boldsymbol{A}_i \boldsymbol{V}\right) \otimes \left(\boldsymbol{V}^{\mathrm{T}} \boldsymbol{A}_i \boldsymbol{V}\right) \mathrm{vec}\left(\boldsymbol{S}\right) \\
&= \sum_{i=1}^{M}\left[\mathrm{vec}\left(\boldsymbol{S}\right)\right]^{\mathrm{T}} \boldsymbol{M}_i \otimes \boldsymbol{M}_i\, \mathrm{vec}\left(\boldsymbol{S}\right).
\end{aligned}
\tag{137}
$$

Now, for each $i \in [n]$ we have

$$\left[\operatorname{vec}(\boldsymbol{S})\right]^{\mathrm{T}} \boldsymbol{M}_i \otimes \boldsymbol{M}_i \operatorname{vec}(\boldsymbol{S}) = \sum_{m=1}^{d^2} \sum_{k=1}^{d^2} [\boldsymbol{M}_i \otimes \boldsymbol{M}_i]_{[m,k]} \operatorname{vec}(\boldsymbol{S})_m \operatorname{vec}(\boldsymbol{S})_k. \tag{138}$$

Set $m = d(m_2 - 1) + m_1$ and $k = d(k_2 - 1) + k_1$ where $m_1, m_2, k_1, k_2 \in [d]$, then

$$\left[\operatorname{vec}(\boldsymbol{S})\right]^{\mathrm{T}} \boldsymbol{M}_i \otimes \boldsymbol{M}_i \operatorname{vec}(\boldsymbol{S}) = \sum_{m_2=1}^{d} \sum_{m_1=1}^{d} \sum_{k_2=1}^{d} \sum_{k_1=1}^{d} \boldsymbol{M}_{i\,[m_2,k_2]} \boldsymbol{M}_{i\,[m_1,k_1]} \boldsymbol{S}_{[m_2,m_1]} \boldsymbol{S}_{[k_2,k_1]}. \tag{139}$$

### C.2.1 SYMMETRIC EIGENVECTORS

Assume that $\boldsymbol{Z}$ is symmetric, then $\boldsymbol{S}$ is a diagonal matrix. Therefore, we only need to consider the terms in the series above for which $m_1 = m_2 = p$ and $k_1 = k_2 = \ell$.

$$\sum_{p=1}^{d} \sum_{\ell=1}^{d} \boldsymbol{M}_{i\,[p,\ell]} \boldsymbol{M}_{i\,[p,\ell]} \boldsymbol{S}_{[p,p]} \boldsymbol{S}_{[\ell,\ell]} = \sum_{p=1}^{d} \sum_{\ell=1}^{d} \boldsymbol{M}_{i\,[p,\ell]}^2 \boldsymbol{s}_{\mathrm{sym},p} \boldsymbol{s}_{\mathrm{sym},\ell} = \boldsymbol{s}_{\mathrm{sym}}^{\mathrm{T}} \boldsymbol{M}_i^{\odot 2} \boldsymbol{s}_{\mathrm{sym}}. \tag{140}$$

Overall,

$$\boldsymbol{z}^{\mathrm{T}} \boldsymbol{Q} \boldsymbol{z} = \boldsymbol{s}_{\mathrm{sym}}^{\mathrm{T}} \sum_{i=1}^{M} \boldsymbol{M}_i^{\odot 2} \boldsymbol{s}_{\mathrm{sym}}. \tag{141}$$

### C.2.2 SKEW-SYMMETRIC EIGENVECTORS

Assume that $\boldsymbol{Z}$ is skew-symmetric, then $\boldsymbol{S}$ is a block diagonal matrix, where each block is $2 \times 2$ in the form of

$$\begin{bmatrix} 0 & s_j \\ -s_j & 0 \end{bmatrix}. \tag{142}$$

If the dimension $d$ is odd, then $\boldsymbol{S}$ has a row and column at the end filed with zeros.

In this scenario, we have four different cases to consider.

**Case I:** $m_1 = 2p - 1, \; m_2 = 2p, \; k_1 = 2\ell - 1, \; k_2 = 2\ell.$

$$\sum_{p=1}^{\lfloor d/2 \rfloor} \sum_{\ell=1}^{\lfloor d/2 \rfloor} \boldsymbol{M}_{i\,[2p,2\ell]} \boldsymbol{M}_{i\,[2p-1,2\ell-1]} \boldsymbol{S}_{[2p,2p-1]} \boldsymbol{S}_{[2\ell,2\ell-1]} = \sum_{p=1}^{\lfloor d/2 \rfloor} \sum_{\ell=1}^{\lfloor d/2 \rfloor} \boldsymbol{M}_{i\,[2p,2\ell]} \boldsymbol{M}_{i\,[2p-1,2\ell-1]} \boldsymbol{s}_{\mathrm{skew},p} \boldsymbol{s}_{\mathrm{skew},\ell}. \tag{143}$$

**Case II:** $m_1 = 2p, \; m_2 = 2p - 1, \; k_1 = 2\ell - 1, \; k_2 = 2\ell.$

$$\sum_{p=1}^{\lfloor d/2 \rfloor} \sum_{\ell=1}^{\lfloor d/2 \rfloor} \boldsymbol{M}_{i\,[2p-1,2\ell]} \boldsymbol{M}_{i\,[2p,2\ell-1]} \boldsymbol{S}_{[2p-1,2p]} \boldsymbol{S}_{[2\ell,2\ell-1]} = -\sum_{p=1}^{\lfloor d/2 \rfloor} \sum_{\ell=1}^{\lfloor d/2 \rfloor} \boldsymbol{M}_{i\,[2p-1,2\ell]} \boldsymbol{M}_{i\,[2p,2\ell-1]} \boldsymbol{s}_{\mathrm{skew},p} \boldsymbol{s}_{\mathrm{skew},\ell}. \tag{144}$$

**Case III:** $m_1 = 2p - 1, \; m_2 = 2p, \; k_1 = 2\ell, \; k_2 = 2\ell - 1.$

$$\sum_{p=1}^{\lfloor d/2 \rfloor} \sum_{\ell=1}^{\lfloor d/2 \rfloor} \boldsymbol{M}_{i\,[2p,2\ell-1]} \boldsymbol{M}_{i\,[2p-1,2\ell]} \boldsymbol{S}_{[2p,2p-1]} \boldsymbol{S}_{[2\ell-1,2\ell]} = -\sum_{p=1}^{\lfloor d/2 \rfloor} \sum_{\ell=1}^{\lfloor d/2 \rfloor} \boldsymbol{M}_{i\,[2p,2\ell-1]} \boldsymbol{M}_{i\,[2p-1,2\ell]} \boldsymbol{s}_{\mathrm{skew},p} \boldsymbol{s}_{\mathrm{skew},\ell}. \tag{145}$$

**Case IV:** $m_1 = 2p, \; m_2 = 2p - 1, \; k_1 = 2\ell, \; k_2 = 2\ell - 1.$

$$\sum_{p=1}^{\lfloor d/2 \rfloor} \sum_{\ell=1}^{\lfloor d/2 \rfloor} \boldsymbol{M}_{i\,[2p-1,2\ell-1]} \boldsymbol{M}_{i\,[2p,2\ell]} \boldsymbol{S}_{[2p-1,2p]} \boldsymbol{S}_{[2\ell-1,2\ell]} = \sum_{p=1}^{\lfloor d/2 \rfloor} \sum_{\ell=1}^{\lfloor d/2 \rfloor} \boldsymbol{M}_{i\,[2p-1,2\ell-1]} \boldsymbol{M}_{i\,[2p,2\ell]} \boldsymbol{s}_{\mathrm{skew},p} \boldsymbol{s}_{\mathrm{skew},\ell}. \tag{146}$$

Summing over all these cases we get

$$
\begin{aligned}
\boldsymbol{z}^{\mathrm{T}} \boldsymbol{Q} \boldsymbol{z} &= \sum_{i=1}^{M} \sum_{p=1}^{\lfloor d/2 \rfloor} \sum_{\ell=1}^{\lfloor d/2 \rfloor} 2 \left( \boldsymbol{M}_{i\,[2\ell-1,2p-1]} \boldsymbol{M}_{i\,[2\ell,2p]} - \boldsymbol{M}_{i\,[2\ell-1,2p]} \boldsymbol{M}_{i\,[2\ell-1,2p]} \right) \boldsymbol{s}_{\mathrm{skew},p} \boldsymbol{s}_{\mathrm{skew},\ell} \\
&= 2 \sum_{i=1}^{M} \sum_{p=1}^{\lfloor d/2 \rfloor} \sum_{\ell=1}^{\lfloor d/2 \rfloor} \boldsymbol{T}_{i\,[p,\ell]} \boldsymbol{s}_{\mathrm{skew},p} \boldsymbol{s}_{\mathrm{skew},\ell} \\
&= 2 \sum_{i=1}^{M} \boldsymbol{s}_{\mathrm{skew}}^{\mathrm{T}} \boldsymbol{T}_i \boldsymbol{s}_{\mathrm{skew}}.
\end{aligned}
\tag{147}
$$

## D   PROOF OF PROPOSITION 1

Here we focus on interpolating minima for simplicity. A similar proof can be derived for regular minima. To begin with, note that (see (91))

$$
\lambda_{\max}\left( \boldsymbol{C}^{\dagger} \boldsymbol{D} \right) = \lambda_{\max}\left( \left( \boldsymbol{C}^{\frac{1}{2}} \right)^{\dagger} \boldsymbol{D} \left( \boldsymbol{C}^{\frac{1}{2}} \right)^{\dagger} \right).
\tag{148}
$$

Additionally,

$$
\begin{aligned}
\boldsymbol{D} &= (1-p)\, \boldsymbol{H} \otimes \boldsymbol{H} + p\, \frac{1}{n} \sum_{i=1}^{n} \boldsymbol{H}_i \otimes \boldsymbol{H}_i \\
&= \boldsymbol{H} \otimes \boldsymbol{H} + p \frac{1}{n} \sum_{i=1}^{n} (\boldsymbol{H}_i - \boldsymbol{H}) \otimes (\boldsymbol{H}_i - \boldsymbol{H}) \\
&= \boldsymbol{H} \otimes \boldsymbol{H} + p \boldsymbol{E},
\end{aligned}
\tag{149}
$$

where $\boldsymbol{E} \triangleq \frac{1}{n} \sum_{i=1}^{n} (\boldsymbol{H}_i - \boldsymbol{H}) \otimes (\boldsymbol{H}_i - \boldsymbol{H})$. Let $\boldsymbol{y} \in \mathbb{S}^{d^2 - 1}$ be the top eigenvector of $(\boldsymbol{C}^{\frac{1}{2}})^{\dagger} \boldsymbol{D} (\boldsymbol{C}^{\frac{1}{2}})^{\dagger}$, then since $(\boldsymbol{C}^{\frac{1}{2}})^{\dagger} \boldsymbol{D} (\boldsymbol{C}^{\frac{1}{2}})^{\dagger}$ is symmetric we have

$$
\begin{aligned}
\frac{\partial}{\partial p} \lambda_{\max}\left( \left( \boldsymbol{C}^{\frac{1}{2}} \right)^{\dagger} \boldsymbol{D} \left( \boldsymbol{C}^{\frac{1}{2}} \right)^{\dagger} \right) &= \boldsymbol{y}^{\mathrm{T}} \left[ \left( \boldsymbol{C}^{\frac{1}{2}} \right)^{\dagger} \left( \frac{\partial}{\partial p} \boldsymbol{D} \right) \left( \boldsymbol{C}^{\frac{1}{2}} \right)^{\dagger} \right] \boldsymbol{y} \\
&= \boldsymbol{y}^{\mathrm{T}} \left[ \left( \boldsymbol{C}^{\frac{1}{2}} \right)^{\dagger} \boldsymbol{E} \left( \boldsymbol{C}^{\frac{1}{2}} \right)^{\dagger} \right] \boldsymbol{y}.
\end{aligned}
\tag{150}
$$

In App. D.1 we show that $\boldsymbol{y}$ has the form of $\boldsymbol{y} = \boldsymbol{C}^{\frac{1}{2}} \boldsymbol{u}$ such that $\mathrm{vec}^{-1}(\boldsymbol{u}) \in \mathcal{S}_{+}(\mathbb{R}^{d \times d})$. Plugging this into the equation above we get

$$
\begin{aligned}
\boldsymbol{y}^{\mathrm{T}} \left[ \left( \boldsymbol{C}^{\frac{1}{2}} \right)^{\dagger} \boldsymbol{E} \left( \boldsymbol{C}^{\frac{1}{2}} \right)^{\dagger} \right] \boldsymbol{y} &= \boldsymbol{u}^{\mathrm{T}} \boldsymbol{C}^{\frac{1}{2}} \left( \boldsymbol{C}^{\frac{1}{2}} \right)^{\dagger} \boldsymbol{E} \left( \boldsymbol{C}^{\frac{1}{2}} \right)^{\dagger} \boldsymbol{C}^{\frac{1}{2}} \boldsymbol{u} \\
&= \boldsymbol{u}^{\mathrm{T}} \boldsymbol{P}_{\mathcal{N}^{\perp}(\boldsymbol{C})} \boldsymbol{E} \boldsymbol{P}_{\mathcal{N}^{\perp}(\boldsymbol{C})} \boldsymbol{u} \\
&= \boldsymbol{u}^{\mathrm{T}} \boldsymbol{E} \boldsymbol{u},
\end{aligned}
\tag{151}
$$

where in the first step we used the fact that $\boldsymbol{C}$ is symmetric. Additionally, in the second step, we used the fact that $\boldsymbol{C}^{\frac{1}{2}} (\boldsymbol{C}^{\frac{1}{2}})^{\dagger}$ and $(\boldsymbol{C}^{\frac{1}{2}})^{\dagger} \boldsymbol{C}^{\frac{1}{2}}$ are projection matrices onto the column space of $\boldsymbol{C}$. Since the null space of $\boldsymbol{E}$ contains the null space of $\boldsymbol{C}$, we have that these projections can be removed (see App. B.6). Note that $\mathrm{vec}^{-1}(\boldsymbol{u})$ is PSD, and let $\boldsymbol{V} \boldsymbol{S} \boldsymbol{V}^{\mathrm{T}}$ be its spectral decomposition, then in App. C.2 we show that in this case

$$
\boldsymbol{u}^{\mathrm{T}} \boldsymbol{E} \boldsymbol{u} = \boldsymbol{s}^{\mathrm{T}} \sum_{i=1}^{n} \boldsymbol{M}_i^{\odot 2} \boldsymbol{s},
\tag{152}
$$

where $\boldsymbol{M}_i = \boldsymbol{V}^{\mathrm{T}} (\boldsymbol{H}_i - \boldsymbol{H}) \boldsymbol{V}$ and $\boldsymbol{s}$ is a vector containing the eigenvalues of $\mathrm{vec}^{-1}(\boldsymbol{u})$. Since $\mathrm{vec}^{-1}(\boldsymbol{u})$ is PSD, we have the right-hand side of (152) is a sum over non-negative terms. Namely,

$$
\frac{\partial}{\partial p} \lambda_{\max}\left( \left( \boldsymbol{C}^{\frac{1}{2}} \right)^{\dagger} \boldsymbol{D} \left( \boldsymbol{C}^{\frac{1}{2}} \right)^{\dagger} \right) = \boldsymbol{u}^{\mathrm{T}} \boldsymbol{E} \boldsymbol{u} = \boldsymbol{s}^{\mathrm{T}} \sum_{i=1}^{n} \boldsymbol{M}_i^{\odot 2} \boldsymbol{s} \geq 0.
\tag{153}
$$

Therefore, $\lambda_{\max}\left(C^\dagger D\right)$ is monotonically non-decreasing in $p$, which means that $\eta^*_{\text{var}} = 2/\lambda_{\max}\left(C^\dagger D\right)$ is monotonically non-decreasing with $B$.

## D.1 TOP EIGENVECTOR OF $(C^{\frac{1}{2}})^\dagger D(C^{\frac{1}{2}})^\dagger$

Using the stability condition of Ma & Ying (2021), we have that $\{\mathbb{E}[\|\boldsymbol{\theta}_t - \boldsymbol{\theta}\|^2]\}$ is bounded if and only if (see proof in (Ma & Ying, 2021))

$$\max_{\boldsymbol{\Sigma} \in \mathcal{S}_+(\mathbb{R}^{d \times d})} \frac{\|\boldsymbol{Q}(\eta, B)\,\text{vec}\,(\boldsymbol{\Sigma})\|}{\|\boldsymbol{\Sigma}\|_{\text{F}}} \leq 1. \tag{154}$$

Let us repeat the same steps from Sec. 3.3 but *without* relaxing the constraint of PSD matrices. Applying Thm. 6 we get that the maximizer for (154) is the top eigenvalue of $\boldsymbol{Q}$ which corresponds to a PSD matrix. Then,

$$\boldsymbol{u}^{\text{T}}\boldsymbol{Q}\boldsymbol{u} = 1 - 2\eta\boldsymbol{u}^{\text{T}}\boldsymbol{C}\boldsymbol{u} + \eta^2\boldsymbol{u}^{\text{T}}\boldsymbol{D}\boldsymbol{u} \leq 1 \tag{155}$$

holds for any $\boldsymbol{u} \in \mathbb{S}^{d^2-1}$ such that $\text{vec}^{-1}(\boldsymbol{u}) \in \mathcal{S}_+(\mathbb{R}^{d \times d})$ and $\boldsymbol{u} \notin \mathcal{N}(\boldsymbol{D})$, if and only if

$$\eta \leq \frac{2}{\lambda^*} = \eta^*_{\text{var}}, \tag{156}$$

where

$$\lambda^* = \sup_{\boldsymbol{u} \in \mathbb{S}^{d^2-1}} \frac{\boldsymbol{u}^{\text{T}}\boldsymbol{D}\boldsymbol{u}}{\boldsymbol{u}^{\text{T}}\boldsymbol{C}\boldsymbol{u}} \quad \text{s.t.} \quad \text{vec}^{-1}(\boldsymbol{u}) \in \mathcal{S}_+(\mathbb{R}^{d \times d}) \text{ and } \boldsymbol{u} \notin \mathcal{N}(\boldsymbol{D}). \tag{157}$$

(Note that $\boldsymbol{u} \in \mathcal{N}(\boldsymbol{D})$ do not contribute any conditions on the learning rate, and therefore can be ignored - see App. B.7). With some algebra (see (87) and (88))

$$\lambda^* = \max_{\boldsymbol{y} \in \mathbb{S}^{d^2-1}} \boldsymbol{y}^{\text{T}}\left(\boldsymbol{C}^{\frac{1}{2}}\right)^\dagger \boldsymbol{D}\left(\boldsymbol{C}^{\frac{1}{2}}\right)^\dagger \boldsymbol{y} \quad \text{s.t.} \quad \boldsymbol{y} = (\boldsymbol{C}^{\frac{1}{2}}\boldsymbol{u}) \text{ and } \text{vec}^{-1}(\boldsymbol{u}) \in \mathcal{S}_+(\mathbb{R}^{d \times d}). \tag{158}$$

Since the alternative form of $\eta^*_{\text{var}}$ in (156) has to be equal to the definition in (13) (or else we will get a contradiction), we get

$$\lambda_{\max}\left(\left(\boldsymbol{C}^{\frac{1}{2}}\right)^\dagger \boldsymbol{D}\left(\boldsymbol{C}^{\frac{1}{2}}\right)^\dagger\right) = \lambda^*. \tag{159}$$

Namely, the top eigenvector $\boldsymbol{y}$ of $(\boldsymbol{C}^{\frac{1}{2}})^\dagger \boldsymbol{D}(\boldsymbol{C}^{\frac{1}{2}})^\dagger$ has the form of $\boldsymbol{y} = \boldsymbol{C}^{\frac{1}{2}}\boldsymbol{u}$ such that $\text{vec}^{-1}(\boldsymbol{u}) \in \mathcal{S}_+(\mathbb{R}^{d \times d})$.

## E PROOF OF PROPOSITION 2

Here we focus on interpolating minima for simplicity. A similar proof can be derived for regular minima. Let $\{\beta_t\}$ and $\{\kappa_t\}$ be i.i.d. random variables such that $\beta_t \sim \text{Bernoulli}(p)$ and $\kappa_t \sim \mathcal{U}(\{1, ..., n\})$, then

$$\mathfrak{B}_t = \begin{cases} \kappa_t & \text{if } \beta_t = 1, \\ \{1, ..., n\} & \text{otherwise.} \end{cases} \tag{160}$$

Let us consider the following stochastic loss function

$$\hat{\mathcal{L}}_t(\boldsymbol{\theta}) = \frac{1}{|\mathfrak{B}_t|} \sum_{j \in \mathfrak{B}_t} \ell_j(\boldsymbol{\theta}), \tag{161}$$

and define the following notation

$$\boldsymbol{A}_t = \boldsymbol{I} - \frac{\eta}{|\mathfrak{B}_t|} \sum_{i \in \mathfrak{B}_t} \boldsymbol{H}_i. \tag{162}$$

First, for interpolating minima we have

$$\boldsymbol{\theta}_{t+1} - \boldsymbol{\theta}^* = \left(\boldsymbol{I} - \frac{\eta}{|\mathfrak{B}_t|} \sum_{i \in \mathfrak{B}_t} \boldsymbol{H}_i\right)(\boldsymbol{\theta}_t - \boldsymbol{\theta}^*) = \boldsymbol{A}_t(\boldsymbol{\theta}_t - \boldsymbol{\theta}^*). \tag{163}$$

Thus,

$$
\begin{aligned}
\boldsymbol{\Sigma}_{t+1} &= \mathbb{E}\left[\left(\boldsymbol{\theta}_{t+1}-\boldsymbol{\theta}^{*}\right)\left(\boldsymbol{\theta}_{t+1}-\boldsymbol{\theta}^{*}\right)^{\mathrm{T}}\right] \\
&= \mathbb{E}\left[\boldsymbol{A}_{t}\left(\boldsymbol{\theta}_{t}-\boldsymbol{\theta}^{*}\right)\left(\boldsymbol{\theta}_{t}-\boldsymbol{\theta}^{*}\right)^{\mathrm{T}}\boldsymbol{A}_{t}\right] \\
&= \mathbb{E}\left[\boldsymbol{A}_{t}\mathbb{E}\left[\left(\boldsymbol{\theta}_{t}-\boldsymbol{\theta}^{*}\right)\left(\boldsymbol{\theta}_{t}-\boldsymbol{\theta}^{*}\right)^{\mathrm{T}}\Big|\mathfrak{B}_{t}\right]\boldsymbol{A}_{t}\right] \\
&= \mathbb{E}\left[\boldsymbol{A}_{t}\mathbb{E}\left[\left(\boldsymbol{\theta}_{t}-\boldsymbol{\theta}^{*}\right)\left(\boldsymbol{\theta}_{t}-\boldsymbol{\theta}^{*}\right)^{\mathrm{T}}\right]\boldsymbol{A}_{t}\right] \\
&= \mathbb{E}\left[\boldsymbol{A}_{t}\boldsymbol{\Sigma}_{t}\boldsymbol{A}_{t}\right].
\end{aligned}
\tag{164}
$$

Using vectorization we get

$$
\operatorname{vec}\left(\boldsymbol{\Sigma}_{t+1}\right) = \mathbb{E}\left[\boldsymbol{A}_{t}\otimes\boldsymbol{A}_{t}\right]\operatorname{vec}\left(\boldsymbol{\Sigma}_{t}\right).
\tag{165}
$$

Let us compute this term.

$$
\begin{aligned}
\mathbb{E}\left[\boldsymbol{A}_{t}\otimes\boldsymbol{A}_{t}\right] &= \mathbb{P}\left(\beta_{t}=0\right)\mathbb{E}\left[\boldsymbol{A}_{t}\otimes\boldsymbol{A}_{t}|\beta_{t}=0\right]+\mathbb{P}\left(\beta_{t}=1\right)\mathbb{E}\left[\boldsymbol{A}_{t}\otimes\boldsymbol{A}_{t}|\beta_{t}=1\right] \\
&= (1-p)\boldsymbol{Q}(\eta, B=n)+p\boldsymbol{Q}(\eta, B=1) \\
&= (1-p)\times(\boldsymbol{I}-\eta\boldsymbol{H})\otimes(\boldsymbol{I}-\eta\boldsymbol{H})+p\times\frac{1}{n}\sum_{i=1}^{n}(\boldsymbol{I}-\eta\boldsymbol{H}_{i})\otimes(\boldsymbol{I}-\eta\boldsymbol{H}_{i}).
\end{aligned}
\tag{166}
$$

This is the same matrix that we had for mini-batch SGD with batch size $B$ such that $p = \frac{n-B}{B(n-1)}$. Therefore, the stability threshold is the same.

## F  PROOF OF PROPOSITION 3

The stability threshold given by Theorem 3 and Theorem 4 is

$$
\eta_{\mathrm{var}}^{*} = \frac{2}{\lambda_{\max}\left(\boldsymbol{C}^{\dagger}\boldsymbol{D}\right)}
\tag{167}
$$

where

$$
\boldsymbol{C} = \frac{1}{2}\boldsymbol{H}\oplus\boldsymbol{H}, \qquad \boldsymbol{D} = (1-p)\,\boldsymbol{H}\otimes\boldsymbol{H}+p\,\frac{1}{n}\sum_{i=1}^{n}\boldsymbol{H}_{i}\otimes\boldsymbol{H}_{i}.
\tag{168}
$$

This threshold corresponds to a necessary and sufficient condition for stability. Here we derive simplified necessary conditions for stability. In App. B.7 we show that

$$
\frac{2}{\lambda_{\max}\left(\boldsymbol{C}^{\dagger}\boldsymbol{D}\right)} = 2\inf_{\boldsymbol{u}\in\mathbb{S}^{d^{2}-1}:\boldsymbol{u}\notin\mathcal{N}(\boldsymbol{D})}\left\{\frac{\boldsymbol{u}^{\mathrm{T}}\boldsymbol{C}\boldsymbol{u}}{\boldsymbol{u}^{\mathrm{T}}\boldsymbol{D}\boldsymbol{u}}\right\}.
\tag{169}
$$

We shall upper bound the stability threshold by considering non-optimal yet interesting vectors $\boldsymbol{u}$. Specifically, in the following we look at $\boldsymbol{u} = \boldsymbol{v}_{\max}\otimes\boldsymbol{v}_{\max}$, where $\boldsymbol{v}_{\max}$ is the top eigenvector of $\boldsymbol{H}$, and $\boldsymbol{u} = \operatorname{vec}(\boldsymbol{I})$ to obtain the results of Proposition 3.

### F.1  SETTING $\boldsymbol{u} = \boldsymbol{v}_{\max}\otimes\boldsymbol{v}_{\max}$

Let $\boldsymbol{u} = \boldsymbol{v}\otimes\boldsymbol{v}\notin\mathcal{N}(\boldsymbol{D})$ where $\|\boldsymbol{v}\|=1$, then

$$
\begin{aligned}
\boldsymbol{u}^{\mathrm{T}}\boldsymbol{C}\boldsymbol{u} &= \frac{1}{2}\boldsymbol{u}^{\mathrm{T}}\left(\boldsymbol{H}\otimes\boldsymbol{I}+\boldsymbol{I}\otimes\boldsymbol{H}\right)\boldsymbol{u} \\
&= \frac{1}{2}\left[\left(\boldsymbol{v}^{\mathrm{T}}\otimes\boldsymbol{v}^{\mathrm{T}}\right)\left(\boldsymbol{H}\otimes\boldsymbol{I}\right)\left(\boldsymbol{v}\otimes\boldsymbol{v}\right)+\left(\boldsymbol{v}^{\mathrm{T}}\otimes\boldsymbol{v}^{\mathrm{T}}\right)\left(\boldsymbol{I}\otimes\boldsymbol{H}\right)\left(\boldsymbol{v}\otimes\boldsymbol{v}\right)\right] \\
&= \frac{1}{2}\left[\left(\boldsymbol{v}^{\mathrm{T}}\boldsymbol{H}\boldsymbol{v}\right)\otimes\left(\boldsymbol{v}^{\mathrm{T}}\boldsymbol{v}\right)+\left(\boldsymbol{v}^{\mathrm{T}}\boldsymbol{v}\right)\otimes\left(\boldsymbol{v}^{\mathrm{T}}\boldsymbol{H}\boldsymbol{v}\right)\right] \\
&= \frac{1}{2}\left[\left(\boldsymbol{v}^{\mathrm{T}}\boldsymbol{H}\boldsymbol{v}\right)\otimes 1+1\otimes\left(\boldsymbol{v}^{\mathrm{T}}\boldsymbol{H}\boldsymbol{v}\right)\right] \\
&= \boldsymbol{v}^{\mathrm{T}}\boldsymbol{H}\boldsymbol{v}.
\end{aligned}
\tag{170}
$$

Similarly,

$$
\begin{aligned}
\boldsymbol{u}^{\mathrm{T}} \left( \boldsymbol{H} \otimes \boldsymbol{H} \right) \boldsymbol{u} &= \left( \boldsymbol{v}^{\mathrm{T}} \otimes \boldsymbol{v}^{\mathrm{T}} \right) \left( \boldsymbol{H} \otimes \boldsymbol{H} \right) \left( \boldsymbol{v} \otimes \boldsymbol{v} \right) \\
&= \left( \boldsymbol{v}^{\mathrm{T}} \boldsymbol{H} \boldsymbol{v} \right) \otimes \left( \boldsymbol{v}^{\mathrm{T}} \boldsymbol{H} \boldsymbol{v} \right) \\
&= \left( \boldsymbol{v}^{\mathrm{T}} \boldsymbol{H} \boldsymbol{v} \right) \otimes \left( \boldsymbol{v}^{\mathrm{T}} \boldsymbol{H} \boldsymbol{v} \right) \\
&= \left( \boldsymbol{v}^{\mathrm{T}} \boldsymbol{H} \boldsymbol{v} \right)^2 .
\end{aligned} \tag{171}
$$

And again

$$
\begin{aligned}
\boldsymbol{u}^{\mathrm{T}} \left( \frac{1}{n} \sum_{i=1}^{n} \boldsymbol{H}_i \otimes \boldsymbol{H}_i \right) \boldsymbol{u} &= \frac{1}{n} \sum_{i=1}^{n} \left( \boldsymbol{v}^{\mathrm{T}} \otimes \boldsymbol{v}^{\mathrm{T}} \right) \left( \boldsymbol{H}_i \otimes \boldsymbol{H}_i \right) \left( \boldsymbol{v} \otimes \boldsymbol{v} \right) \\
&= \frac{1}{n} \sum_{i=1}^{n} \left( \boldsymbol{v}^{\mathrm{T}} \boldsymbol{H}_i \boldsymbol{v} \right) \otimes \left( \boldsymbol{v}^{\mathrm{T}} \boldsymbol{H}_i \boldsymbol{v} \right) \\
&= \frac{1}{n} \sum_{i=1}^{n} \left( \boldsymbol{v}^{\mathrm{T}} \boldsymbol{H}_i \boldsymbol{v} \right)^2 .
\end{aligned} \tag{172}
$$

Thus,

$$
\begin{aligned}
\boldsymbol{u}^{\mathrm{T}} \boldsymbol{D} \boldsymbol{u} &= (1-p) \boldsymbol{u}^{\mathrm{T}} \left( \boldsymbol{H} \otimes \boldsymbol{H} \right) \boldsymbol{u} + p \boldsymbol{u}^{\mathrm{T}} \left( \frac{1}{n} \sum_{i=1}^{n} \boldsymbol{H}_i \otimes \boldsymbol{H}_i \right) \boldsymbol{u} \\
&= (1-p) \left( \boldsymbol{v}^{\mathrm{T}} \boldsymbol{H} \boldsymbol{v} \right)^2 + p \frac{1}{n} \sum_{i=1}^{n} \left( \boldsymbol{v}^{\mathrm{T}} \boldsymbol{H}_i \boldsymbol{v} \right)^2 \\
&= \left( \boldsymbol{v}^{\mathrm{T}} \boldsymbol{H} \boldsymbol{v} \right)^2 + p \left[ \frac{1}{n} \sum_{i=1}^{n} \left( \boldsymbol{v}^{\mathrm{T}} \boldsymbol{H}_i \boldsymbol{v} \right)^2 - \left( \boldsymbol{v}^{\mathrm{T}} \boldsymbol{H} \boldsymbol{v} \right)^2 \right] \\
&= \left( \boldsymbol{v}^{\mathrm{T}} \boldsymbol{H} \boldsymbol{v} \right)^2 + p \left[ \frac{1}{n} \sum_{i=1}^{n} \left( \boldsymbol{v}^{\mathrm{T}} \boldsymbol{H}_i \boldsymbol{v} \right)^2 - 2 \left( \boldsymbol{v}^{\mathrm{T}} \boldsymbol{H} \boldsymbol{v} \right) \left( \boldsymbol{v}^{\mathrm{T}} \boldsymbol{H} \boldsymbol{v} \right) + \left( \boldsymbol{v}^{\mathrm{T}} \boldsymbol{H} \boldsymbol{v} \right)^2 \right] \\
&= \left( \boldsymbol{v}^{\mathrm{T}} \boldsymbol{H} \boldsymbol{v} \right)^2 + p \frac{1}{n} \sum_{i=1}^{n} \left[ \left( \boldsymbol{v}^{\mathrm{T}} \boldsymbol{H}_i \boldsymbol{v} \right)^2 - 2 \left( \boldsymbol{v}^{\mathrm{T}} \boldsymbol{H}_i \boldsymbol{v} \right) \left( \boldsymbol{v}^{\mathrm{T}} \boldsymbol{H} \boldsymbol{v} \right) + \left( \boldsymbol{v}^{\mathrm{T}} \boldsymbol{H} \boldsymbol{v} \right)^2 \right] \\
&= \left( \boldsymbol{v}^{\mathrm{T}} \boldsymbol{H} \boldsymbol{v} \right)^2 + p \frac{1}{n} \sum_{i=1}^{n} \left( \boldsymbol{v}^{\mathrm{T}} \boldsymbol{H}_i \boldsymbol{v} - \left( \boldsymbol{v}^{\mathrm{T}} \boldsymbol{H} \boldsymbol{v} \right) \right)^2 .
\end{aligned} \tag{173}
$$

Therefore, for general $\boldsymbol{u} = \boldsymbol{v} \otimes \boldsymbol{v}$ we get

$$
\eta_{\mathrm{var}}^* \le 2 \frac{\boldsymbol{u}^{\mathrm{T}} \boldsymbol{C} \boldsymbol{u}}{\boldsymbol{u}^{\mathrm{T}} \boldsymbol{D} \boldsymbol{u}} = \frac{2 \boldsymbol{v}^{\mathrm{T}} \boldsymbol{H} \boldsymbol{v}}{(\boldsymbol{v}^{\mathrm{T}} \boldsymbol{H} \boldsymbol{v})^2 + \frac{p}{n} \sum_{i=1}^{n} (\boldsymbol{v}^{\mathrm{T}} \boldsymbol{H}_i \boldsymbol{v} - \boldsymbol{v}^{\mathrm{T}} \boldsymbol{H} \boldsymbol{v})^2} . \tag{174}
$$

Specifically, for $\boldsymbol{u} = \boldsymbol{v}_{\mathrm{max}} \otimes \boldsymbol{v}_{\mathrm{max}}$ we get

$$
\eta_{\mathrm{var}}^* \le \frac{2 \lambda_{\mathrm{max}}(\boldsymbol{H})}{\lambda_{\mathrm{max}}^2(\boldsymbol{H}) + p \frac{1}{n} \sum_{i=1}^{n} \left( \boldsymbol{v}_{\mathrm{max}}^{\mathrm{T}} \boldsymbol{H}_i \boldsymbol{v}_{\mathrm{max}} - \lambda_{\mathrm{max}}(\boldsymbol{H}) \right)^2} . \tag{175}
$$

Finally, from (174) we get the following result which we used in Sec. 4.

$$
\lambda_{\mathrm{max}} \left( \boldsymbol{C}^{\dagger} \boldsymbol{D} \right) = \frac{2}{\eta_{\mathrm{var}}^*} \ge \boldsymbol{v}^{\mathrm{T}} \boldsymbol{H} \boldsymbol{v} + p \frac{\frac{1}{n} \sum_{i=1}^{n} (\boldsymbol{v}^{\mathrm{T}} \boldsymbol{H}_i \boldsymbol{v} - \boldsymbol{v}^{\mathrm{T}} \boldsymbol{H} \boldsymbol{v})^2}{\boldsymbol{v}^{\mathrm{T}} \boldsymbol{H} \boldsymbol{v}} . \tag{176}
$$

Since this inequality holds for every $\boldsymbol{v} \notin \mathcal{N}(\boldsymbol{H})$, we can take the maximum to obtain (24).

## F.2 SETTING $\boldsymbol{u} = \mathrm{vec}\,(\boldsymbol{I})$

Let $\boldsymbol{u} = \mathrm{vec}(\boldsymbol{I}) \notin \mathcal{N}(\boldsymbol{D})$, then

$$
\begin{aligned}
\boldsymbol{u}^{\mathrm{T}} \boldsymbol{C} \boldsymbol{u} &= \frac{1}{2} \boldsymbol{u}^{\mathrm{T}} \left( \boldsymbol{H} \otimes \boldsymbol{I} + \boldsymbol{I} \otimes \boldsymbol{H} \right) \boldsymbol{u} \\
&= \frac{1}{2} \left( [\mathrm{vec}\,(\boldsymbol{I})]^{\mathrm{T}} \left( \boldsymbol{H} \otimes \boldsymbol{I} \right) \mathrm{vec}\,(\boldsymbol{I}) + [\mathrm{vec}\,(\boldsymbol{I})]^{\mathrm{T}} \left( \boldsymbol{I} \otimes \boldsymbol{H} \right) \mathrm{vec}\,(\boldsymbol{I}) \right) \\
&= \frac{1}{2} \left( \mathrm{Tr}(\boldsymbol{H}^{\mathrm{T}}) + \mathrm{Tr}(\boldsymbol{H}) \right) \\
&= \mathrm{Tr}(\boldsymbol{H}),
\end{aligned}
\tag{177}
$$

where we used (P4). Moreover,

$$
\boldsymbol{u}^{\mathrm{T}} \left( \boldsymbol{H} \otimes \boldsymbol{H} \right) \boldsymbol{u} = [\mathrm{vec}\,(\boldsymbol{I})]^{\mathrm{T}} \left( \boldsymbol{H} \otimes \boldsymbol{H} \right) \mathrm{vec}\,(\boldsymbol{I}) = \mathrm{Tr}(\boldsymbol{H}\boldsymbol{H}^{\mathrm{T}}) = \|\boldsymbol{H}\|_{\mathrm{F}}^2 .
$$

Similarly,

$$
\boldsymbol{u}^{\mathrm{T}} \left( \boldsymbol{H}_i \otimes \boldsymbol{H}_i \right) \boldsymbol{u} = [\mathrm{vec}\,(\boldsymbol{I})]^{\mathrm{T}} \left( \boldsymbol{H}_i \otimes \boldsymbol{H}_i \right) \mathrm{vec}\,(\boldsymbol{I}) = \mathrm{Tr}(\boldsymbol{H}_i \boldsymbol{H}_i^{\mathrm{T}}) = \|\boldsymbol{H}_i\|_{\mathrm{F}}^2 .
$$

$$
\begin{aligned}
\boldsymbol{u}^{\mathrm{T}} \boldsymbol{D} \boldsymbol{u} &= (1-p) \boldsymbol{u}^{\mathrm{T}} \left( \boldsymbol{H} \otimes \boldsymbol{H} \right) \boldsymbol{u} + p \frac{1}{n} \sum_{i=1}^{n} \boldsymbol{u}^{\mathrm{T}} \boldsymbol{H}_i \otimes \boldsymbol{H}_i \boldsymbol{u} \\
&= (1-p) \|\boldsymbol{H}\|_{\mathrm{F}}^2 + p \frac{1}{n} \sum_{i=1}^{n} \|\boldsymbol{H}_i\|_{\mathrm{F}}^2 .
\end{aligned}
\tag{178}
$$

Therefore,

$$
\eta_{\mathrm{var}}^* \leq 2 \frac{\boldsymbol{u}^{\mathrm{T}} \boldsymbol{C} \boldsymbol{u}}{\boldsymbol{u}^{\mathrm{T}} \boldsymbol{D} \boldsymbol{u}} = \frac{2 \mathrm{Tr}(\boldsymbol{H})}{(1-p) \|\boldsymbol{H}\|_{\mathrm{F}}^2 + p \frac{1}{n} \sum_{i=1}^{n} \|\boldsymbol{H}_i\|_{\mathrm{F}}^2} .
\tag{179}
$$

## G  PROOF OF THEOREM 5

In this section, we use the following result on the Moore–Penrose inverse of a sum of two matrices.

**Theorem 7** (Fill & Fishkind (2000), Thm. 3)**.** *Let* $\boldsymbol{X}, \boldsymbol{Y} \in \mathbb{R}^{p \times p}$ *with* $\mathrm{rank}(\boldsymbol{X} + \boldsymbol{Y}) = \mathrm{rank}(\boldsymbol{X}) + \mathrm{rank}(\boldsymbol{Y})$. *Then*

$$
(\boldsymbol{X} + \boldsymbol{Y})^{\dagger} = (\boldsymbol{I} - \boldsymbol{L}) \boldsymbol{X}^{\dagger} (\boldsymbol{I} - \boldsymbol{O}) + \boldsymbol{L} \boldsymbol{Y}^{\dagger} \boldsymbol{O},
\tag{180}
$$

*where*

$$
\boldsymbol{L} = \left( \boldsymbol{P}_{\mathcal{R}(\boldsymbol{Y}^{\mathrm{T}})} \boldsymbol{P}_{\mathcal{R}^{\perp}(\boldsymbol{X}^{\mathrm{T}})} \right)^{\dagger} \quad and \quad \boldsymbol{O} = \left( \boldsymbol{P}_{\mathcal{R}^{\perp}(\boldsymbol{X})} \boldsymbol{P}_{\mathcal{R}(\boldsymbol{Y})} \right)^{\dagger} .
\tag{181}
$$

Moreover, we use the following relations.

$$
\begin{aligned}
\mathcal{R}(\boldsymbol{D}) &= \mathcal{R}(\boldsymbol{P}_{\mathcal{N}^{\perp}(\boldsymbol{H})} \otimes \boldsymbol{P}_{\mathcal{N}^{\perp}(\boldsymbol{H})}), \\
\mathcal{R}(\boldsymbol{C}) &= \mathcal{R}(\boldsymbol{I} - \boldsymbol{P}_{\mathcal{N}(\boldsymbol{H})} \otimes \boldsymbol{P}_{\mathcal{N}(\boldsymbol{H})}), \\
\mathcal{R}(\boldsymbol{P}_{\mathcal{N}(\boldsymbol{D})} \boldsymbol{C}) &= \mathcal{R}(\boldsymbol{P}_{\mathcal{N}(\boldsymbol{H})} \otimes \boldsymbol{P}_{\mathcal{N}^{\perp}(\boldsymbol{H})} + \boldsymbol{P}_{\mathcal{N}^{\perp}(\boldsymbol{H})} \otimes \boldsymbol{P}_{\mathcal{N}(\boldsymbol{H})}).
\end{aligned}
\tag{182}
$$

The dynamics of $\boldsymbol{\mu}_t^{\perp}$ and $\boldsymbol{\Sigma}_t^{\perp}$ are given by (see (53))

$$
\begin{pmatrix} \boldsymbol{\mu}_{t+1}^{\perp} \\ \mathrm{vec}\left( \boldsymbol{\Sigma}_{t+1}^{\perp} \right) \end{pmatrix} = \boldsymbol{\Xi} \begin{pmatrix} \boldsymbol{\mu}_t^{\perp} \\ \mathrm{vec}\left( \boldsymbol{\Sigma}_t^{\perp} \right) \end{pmatrix} + \begin{pmatrix} \boldsymbol{0} \\ \mathrm{vec}\left( \boldsymbol{\Sigma}_{\boldsymbol{v}}^{\perp} \right) \end{pmatrix}.
\tag{183}
$$

where

$$
\begin{aligned}
\boldsymbol{\Xi} &= \begin{pmatrix} \boldsymbol{P}_{\mathcal{N}^{\perp}(\boldsymbol{H})} - \eta \boldsymbol{H} & \boldsymbol{0} \\ - \left( \boldsymbol{P}_{\mathcal{N}^{\perp}(\boldsymbol{H})} \otimes \boldsymbol{P}_{\mathcal{N}^{\perp}(\boldsymbol{H})} \right) \left( \mathbb{E}\left[ \boldsymbol{v}_t^{\perp} \otimes \boldsymbol{A}_t \right] + \mathbb{E}\left[ \boldsymbol{A}_t \otimes \boldsymbol{v}_t^{\perp} \right] \right) & \left( \boldsymbol{P}_{\mathcal{N}^{\perp}(\boldsymbol{H})} \otimes \boldsymbol{P}_{\mathcal{N}^{\perp}(\boldsymbol{H})} \right) \boldsymbol{Q} \end{pmatrix} \\
&\triangleq \begin{pmatrix} \boldsymbol{\Xi}_{1,1} & \boldsymbol{\Xi}_{1,2} \\ \boldsymbol{\Xi}_{2,1} & \boldsymbol{\Xi}_{2,2} \end{pmatrix}.
\end{aligned}
\tag{184}
$$

In App. B.8 we show that if $0 < \eta < \eta_{\text{var}}^*$ then the spectral radius of $\boldsymbol{\Xi}$ is less then one. Therefore, the dynamical system is stable, and the asymptotic values of $\boldsymbol{\mu}_t^\perp$ and $\boldsymbol{\Sigma}_t^\perp$ as $t \to \infty$ are given by

$$\lim_{t \to \infty} \begin{pmatrix} \boldsymbol{\mu}_t^\perp \\ \text{vec}\left(\boldsymbol{\Sigma}_t^\perp\right) \end{pmatrix} = (\boldsymbol{I} - \boldsymbol{\Xi})^{-1} \begin{pmatrix} \boldsymbol{0} \\ \text{vec}\left(\boldsymbol{\Sigma}_v^\perp\right) \end{pmatrix}. \tag{185}$$

Using the inversion formula for block matrix and the fact that $\boldsymbol{\Xi}_{1,2} = \boldsymbol{0}$ we have that

$$\begin{aligned}
(\boldsymbol{I} - \boldsymbol{\Xi})^{-1} &= \begin{pmatrix} \boldsymbol{I} - \boldsymbol{\Xi}_{1,1} & -\boldsymbol{\Xi}_{1,2} \\ -\boldsymbol{\Xi}_{2,1} & \boldsymbol{I} - \boldsymbol{\Xi}_{2,2} \end{pmatrix}^{-1} \\
&= \begin{pmatrix} \left(\boldsymbol{I} - \boldsymbol{\Xi}_{1,1} - \boldsymbol{\Xi}_{1,2}\left(\boldsymbol{I} - \boldsymbol{\Xi}_{2,2}\right)^{-1}\boldsymbol{\Xi}_{2,1}\right)^{-1} & \boldsymbol{0} \\ \boldsymbol{0} & \left(\boldsymbol{I} - \boldsymbol{\Xi}_{2,2} - \boldsymbol{\Xi}_{2,1}\left(\boldsymbol{I} - \boldsymbol{\Xi}_{1,1}\right)^{-1}\boldsymbol{\Xi}_{1,2}\right)^{-1} \end{pmatrix} \\
&\quad \begin{pmatrix} \boldsymbol{I} & \boldsymbol{\Xi}_{1,2}\left(\boldsymbol{I} - \boldsymbol{\Xi}_{2,2}\right)^{-1} \\ \boldsymbol{\Xi}_{2,1}\left(\boldsymbol{I} - \boldsymbol{\Xi}_{1,1}\right)^{-1} & \boldsymbol{I} \end{pmatrix} \\
&= \begin{pmatrix} (\boldsymbol{I} - \boldsymbol{\Xi}_{1,1})^{-1} & \boldsymbol{0} \\ \boldsymbol{0} & (\boldsymbol{I} - \boldsymbol{\Xi}_{2,2})^{-1} \end{pmatrix} \begin{pmatrix} \boldsymbol{I} & \boldsymbol{0} \\ \boldsymbol{\Xi}_{2,1}\left(\boldsymbol{I} - \boldsymbol{\Xi}_{1,1}\right)^{-1} & \boldsymbol{I} \end{pmatrix}. \tag{186}
\end{aligned}$$

Therefore,

$$\begin{aligned}
\lim_{t \to \infty} \begin{pmatrix} \boldsymbol{\mu}_t^\perp \\ \text{vec}\left(\boldsymbol{\Sigma}_t^\perp\right) \end{pmatrix} &= \begin{pmatrix} (\boldsymbol{I} - \boldsymbol{\Xi}_{1,1})^{-1} & \boldsymbol{0} \\ \boldsymbol{0} & (\boldsymbol{I} - \boldsymbol{\Xi}_{2,2})^{-1} \end{pmatrix} \begin{pmatrix} \boldsymbol{I} & \boldsymbol{0} \\ \boldsymbol{\Xi}_{2,1}\left(\boldsymbol{I} - \boldsymbol{\Xi}_{1,1}\right)^{-1} & \boldsymbol{I} \end{pmatrix} \begin{pmatrix} \boldsymbol{0} \\ \text{vec}\left(\boldsymbol{\Sigma}_v^\perp\right) \end{pmatrix} \\
&= \begin{pmatrix} (\boldsymbol{I} - \boldsymbol{\Xi}_{1,1})^{-1} & \boldsymbol{0} \\ \boldsymbol{0} & (\boldsymbol{I} - \boldsymbol{\Xi}_{2,2})^{-1} \end{pmatrix} \begin{pmatrix} \boldsymbol{0} \\ \text{vec}\left(\boldsymbol{\Sigma}_v^\perp\right) \end{pmatrix}. \tag{187}
\end{aligned}$$

Namely,

$$\lim_{t \to \infty} \boldsymbol{\mu}_t^\perp = \boldsymbol{0} \qquad \text{and} \qquad \lim_{t \to \infty} \text{vec}\left(\boldsymbol{\Sigma}_t^\perp\right) = \left(\boldsymbol{I} - \boldsymbol{P}_{\mathcal{N}^\perp(\boldsymbol{D})}\boldsymbol{Q}\right)^{-1} \text{vec}\left(\boldsymbol{\Sigma}_v^\perp\right). \tag{188}$$

Now,

$$\begin{aligned}
\left(\boldsymbol{I} - \boldsymbol{P}_{\mathcal{N}^\perp(\boldsymbol{D})}\boldsymbol{Q}\right)^{-1} &= \left(\boldsymbol{I} - \boldsymbol{P}_{\mathcal{N}^\perp(\boldsymbol{D})}\boldsymbol{Q}\right)^\dagger \\
&= \left(\boldsymbol{P}_{\mathcal{N}(\boldsymbol{D})} + \boldsymbol{P}_{\mathcal{N}^\perp(\boldsymbol{D})} - \boldsymbol{P}_{\mathcal{N}^\perp(\boldsymbol{D})}\boldsymbol{Q}\right)^\dagger \\
&= \left(\boldsymbol{P}_{\mathcal{N}(\boldsymbol{D})} + \boldsymbol{P}_{\mathcal{N}^\perp(\boldsymbol{D})}\left(\boldsymbol{I} - \boldsymbol{Q}\right)\right)^\dagger. \tag{189}
\end{aligned}$$

Let us apply Thm. 7 on $\left(\boldsymbol{P}_{\boldsymbol{D}} + \boldsymbol{P}_{\mathcal{N}^\perp(\boldsymbol{D})}(\boldsymbol{I} - \boldsymbol{Q})\right)^\dagger$. Here, $\boldsymbol{X}_1 = \boldsymbol{P}_{\mathcal{N}(\boldsymbol{D})}$ and $\boldsymbol{Y}_1 = \boldsymbol{P}_{\mathcal{N}^\perp(\boldsymbol{D})}(\boldsymbol{I} - \boldsymbol{Q})$. Note that $\mathcal{R}(\boldsymbol{X}_1) = \mathcal{R}^\perp(\boldsymbol{D})$ and $\mathcal{R}(\boldsymbol{Y}_1) = \mathcal{R}(\boldsymbol{D})$ and therefore $\text{rank}(\boldsymbol{X}_1 + \boldsymbol{Y}_1) = \text{rank}(\boldsymbol{X}_1) + \text{rank}(\boldsymbol{Y}_1)$. Additionally,

$$\boldsymbol{P}_{\mathcal{R}(\boldsymbol{Y}_1^\mathsf{T})} = \boldsymbol{P}_{\mathcal{N}^\perp(\boldsymbol{D})}, \quad \boldsymbol{P}_{\mathcal{R}^\perp(\boldsymbol{X}_1^\mathsf{T})} = \boldsymbol{P}_{\mathcal{N}^\perp(\boldsymbol{D})}, \quad \boldsymbol{P}_{\mathcal{R}^\perp(\boldsymbol{X}_1)} = \boldsymbol{P}_{\mathcal{N}^\perp(\boldsymbol{D})}, \quad \boldsymbol{P}_{\mathcal{R}(\boldsymbol{Y}_1)} = \boldsymbol{P}_{\mathcal{N}^\perp(\boldsymbol{D})}. \tag{190}$$

Hence,

$$\begin{aligned}
\boldsymbol{L}_1 &= \left(\boldsymbol{P}_{\mathcal{R}(\boldsymbol{Y}_1^\mathsf{T})}\boldsymbol{P}_{\mathcal{R}^\perp(\boldsymbol{X}_1^\mathsf{T})}\right)^\dagger = \left(\boldsymbol{P}_{\mathcal{N}^\perp(\boldsymbol{D})}\boldsymbol{P}_{\mathcal{N}^\perp(\boldsymbol{D})}\right)^\dagger = \boldsymbol{P}_{\mathcal{N}^\perp(\boldsymbol{D})}, \\
\boldsymbol{O}_1 &= \left(\boldsymbol{P}_{\mathcal{R}^\perp(\boldsymbol{X}_1)}\boldsymbol{P}_{\mathcal{R}(\boldsymbol{Y}_1)}\right)^\dagger = \left(\boldsymbol{P}_{\mathcal{N}^\perp(\boldsymbol{D})}\boldsymbol{P}_{\mathcal{N}^\perp(\boldsymbol{D})}\right)^\dagger = \boldsymbol{P}_{\mathcal{N}^\perp(\boldsymbol{D})}. \tag{191}
\end{aligned}$$

Therefore,

$$\begin{aligned}
\left(\boldsymbol{I} - \boldsymbol{P}_{\mathcal{N}^\perp(\boldsymbol{D})}\boldsymbol{Q}\right)^{-1} &= \left(\boldsymbol{P}_{\mathcal{N}(\boldsymbol{D})} + \boldsymbol{P}_{\mathcal{N}^\perp(\boldsymbol{D})}\left(\boldsymbol{I} - \boldsymbol{Q}\right)\right)^\dagger \\
&= (\boldsymbol{X}_1 + \boldsymbol{Y}_1)^\dagger \\
&= (\boldsymbol{I} - \boldsymbol{L}_1)\boldsymbol{X}_1^\dagger(\boldsymbol{I} - \boldsymbol{O}_1) + \boldsymbol{L}_1\boldsymbol{Y}_1^\dagger\boldsymbol{O}_1 \\
&= (\boldsymbol{I} - \boldsymbol{P}_{\mathcal{N}^\perp(\boldsymbol{D})})(\boldsymbol{P}_{\mathcal{N}(\boldsymbol{D})})^\dagger(\boldsymbol{I} - \boldsymbol{P}_{\mathcal{N}^\perp(\boldsymbol{D})}) \\
&\quad + \boldsymbol{P}_{\mathcal{N}^\perp(\boldsymbol{D})}\left(\boldsymbol{P}_{\mathcal{N}^\perp(\boldsymbol{D})}(\boldsymbol{I} - \boldsymbol{Q})\right)^\dagger \boldsymbol{P}_{\mathcal{N}^\perp(\boldsymbol{D})} \\
&= (\boldsymbol{I} - \boldsymbol{P}_{\mathcal{N}^\perp(\boldsymbol{D})})\boldsymbol{P}_{\mathcal{N}(\boldsymbol{D})}(\boldsymbol{I} - \boldsymbol{P}_{\mathcal{N}^\perp(\boldsymbol{D})}) \\
&\quad + \boldsymbol{P}_{\mathcal{N}^\perp(\boldsymbol{D})}\left(\boldsymbol{P}_{\mathcal{N}^\perp(\boldsymbol{D})}(\boldsymbol{I} - \boldsymbol{Q})\right)^\dagger \boldsymbol{P}_{\mathcal{N}^\perp(\boldsymbol{D})} \\
&= \boldsymbol{P}_{\mathcal{N}(\boldsymbol{D})} + \boldsymbol{P}_{\mathcal{N}^\perp(\boldsymbol{D})}\left(\boldsymbol{P}_{\mathcal{N}^\perp(\boldsymbol{D})}(\boldsymbol{I} - \boldsymbol{Q})\right)^\dagger \boldsymbol{P}_{\mathcal{N}^\perp(\boldsymbol{D})}, \tag{192}
\end{aligned}$$

where in the third step we used Thm. 7. Thus we get the following intermediate result

$$
\begin{aligned}
\lim_{t\to\infty} \operatorname{vec}\left(\boldsymbol{\Sigma}_t^{\perp}\right) &= \left(\boldsymbol{I} - \boldsymbol{P}_{\mathcal{N}^{\perp}(\boldsymbol{D})}\boldsymbol{Q}\right)^{-1} \operatorname{vec}\left(\boldsymbol{\Sigma}_{\boldsymbol{v}}^{\perp}\right) \\
&= \left(\boldsymbol{P}_{\mathcal{N}(\boldsymbol{D})} + \boldsymbol{P}_{\mathcal{N}^{\perp}(\boldsymbol{D})}\left(\boldsymbol{P}_{\mathcal{N}^{\perp}(\boldsymbol{D})}(\boldsymbol{I}-\boldsymbol{Q})\right)^{\dagger} \boldsymbol{P}_{\mathcal{N}^{\perp}(\boldsymbol{D})}\right) \operatorname{vec}\left(\boldsymbol{\Sigma}_{\boldsymbol{v}}^{\perp}\right) \\
&= \boldsymbol{P}_{\mathcal{N}^{\perp}(\boldsymbol{D})}\left(\boldsymbol{P}_{\mathcal{N}^{\perp}(\boldsymbol{D})}(\boldsymbol{I}-\boldsymbol{Q})\right)^{\dagger} \boldsymbol{P}_{\mathcal{N}^{\perp}(\boldsymbol{D})}\operatorname{vec}\left(\boldsymbol{\Sigma}_{\boldsymbol{v}}^{\perp}\right), \quad (193)
\end{aligned}
$$

where in the final step we used $\boldsymbol{P}_{\mathcal{N}(\boldsymbol{D})}\operatorname{vec}(\boldsymbol{\Sigma}_{\boldsymbol{v}}^{\perp}) = \boldsymbol{0}$. Now, note that $\mathcal{R}(\boldsymbol{P}_{\mathcal{N}(\boldsymbol{D})}\boldsymbol{C}) = \mathcal{R}(\boldsymbol{P}_{\mathcal{N}(\boldsymbol{H})} \otimes \boldsymbol{P}_{\mathcal{N}^{\perp}(\boldsymbol{H})} + \boldsymbol{P}_{\mathcal{N}^{\perp}(\boldsymbol{H})} \otimes \boldsymbol{P}_{\mathcal{N}(\boldsymbol{H})})$, whereas $\operatorname{vec}(\boldsymbol{\Sigma}_{\boldsymbol{v}}^{\perp}) \in \mathcal{R}(\boldsymbol{D}) = \mathcal{R}(\boldsymbol{P}_{\mathcal{N}^{\perp}(\boldsymbol{H})} \otimes \boldsymbol{P}_{\mathcal{N}^{\perp}(\boldsymbol{H})})$ and therefore $(\boldsymbol{P}_{\mathcal{N}(\boldsymbol{D})}\boldsymbol{C})^{\dagger}\operatorname{vec}(\boldsymbol{\Sigma}_{\boldsymbol{v}}^{\perp}) = \boldsymbol{0}$. Hence,

$$
\begin{aligned}
\lim_{t\to\infty} \operatorname{vec}\left(\boldsymbol{\Sigma}_t^{\perp}\right) &= \boldsymbol{P}_{\mathcal{N}^{\perp}(\boldsymbol{D})}\left(\boldsymbol{P}_{\mathcal{N}^{\perp}(\boldsymbol{D})}(\boldsymbol{I}-\boldsymbol{Q})\right)^{\dagger} \boldsymbol{P}_{\mathcal{N}^{\perp}(\boldsymbol{D})}\operatorname{vec}\left(\boldsymbol{\Sigma}_{\boldsymbol{v}}^{\perp}\right) \\
&= \left(\left(2\eta\boldsymbol{P}_{\mathcal{N}(\boldsymbol{D})}\boldsymbol{C}\right)^{\dagger} + \boldsymbol{P}_{\mathcal{N}^{\perp}(\boldsymbol{D})}\left(\boldsymbol{P}_{\mathcal{N}^{\perp}(\boldsymbol{D})}(\boldsymbol{I}-\boldsymbol{Q})\right)^{\dagger} \boldsymbol{P}_{\mathcal{N}^{\perp}(\boldsymbol{D})}\right) \operatorname{vec}\left(\boldsymbol{\Sigma}_{\boldsymbol{v}}^{\perp}\right). \quad (194)
\end{aligned}
$$

Let us apply again Thm. 7 but in the other direction. This time, $\boldsymbol{X}_2 = 2\eta\boldsymbol{P}_{\mathcal{N}(\boldsymbol{D})}\boldsymbol{C}$ and $\boldsymbol{Y}_2 = \boldsymbol{P}_{\mathcal{N}^{\perp}(\boldsymbol{D})}(\boldsymbol{I}-\boldsymbol{Q})$. Note that $\mathcal{R}(\boldsymbol{X}_2) = \mathcal{R}(\boldsymbol{P}_{\mathcal{N}(\boldsymbol{H})}\otimes\boldsymbol{P}_{\mathcal{N}^{\perp}(\boldsymbol{H})} + \boldsymbol{P}_{\mathcal{N}^{\perp}(\boldsymbol{H})}\otimes\boldsymbol{P}_{\mathcal{N}(\boldsymbol{H})})$ and $\mathcal{R}(\boldsymbol{Y}_2) = \mathcal{R}(\boldsymbol{P}_{\mathcal{N}^{\perp}(\boldsymbol{H})}\otimes\boldsymbol{P}_{\mathcal{N}^{\perp}(\boldsymbol{H})})$ and therefore $\operatorname{rank}(\boldsymbol{X}_2 + \boldsymbol{Y}_2) = \operatorname{rank}(\boldsymbol{X}_2) + \operatorname{rank}(\boldsymbol{Y}_2)$. Additionally,

$$
\begin{aligned}
\boldsymbol{P}_{\mathcal{R}(\boldsymbol{Y}_2^{\mathrm{T}})} &= \boldsymbol{P}_{\mathcal{N}^{\perp}(\boldsymbol{H})} \otimes \boldsymbol{P}_{\mathcal{N}^{\perp}(\boldsymbol{H})}, \\
\boldsymbol{P}_{\mathcal{R}(\boldsymbol{Y}_2)} &= \boldsymbol{P}_{\mathcal{N}^{\perp}(\boldsymbol{H})} \otimes \boldsymbol{P}_{\mathcal{N}^{\perp}(\boldsymbol{H})}, \\
\boldsymbol{P}_{\mathcal{R}^{\perp}(\boldsymbol{X}_2^{\mathrm{T}})} &= \boldsymbol{P}_{\mathcal{N}(\boldsymbol{H})} \otimes \boldsymbol{P}_{\mathcal{N}(\boldsymbol{H})} + \boldsymbol{P}_{\mathcal{N}^{\perp}(\boldsymbol{H})} \otimes \boldsymbol{P}_{\mathcal{N}^{\perp}(\boldsymbol{H})}, \\
\boldsymbol{P}_{\mathcal{R}^{\perp}(\boldsymbol{X}_2)} &= \boldsymbol{P}_{\mathcal{N}(\boldsymbol{H})} \otimes \boldsymbol{P}_{\mathcal{N}(\boldsymbol{H})} + \boldsymbol{P}_{\mathcal{N}^{\perp}(\boldsymbol{H})} \otimes \boldsymbol{P}_{\mathcal{N}^{\perp}(\boldsymbol{H})}. \quad (195)
\end{aligned}
$$

Hence,

$$
\begin{aligned}
\boldsymbol{L}_2 &= \left(\boldsymbol{P}_{\mathcal{R}(\boldsymbol{Y}_2^{\mathrm{T}})}\boldsymbol{P}_{\mathcal{R}^{\perp}(\boldsymbol{X}_2^{\mathrm{T}})}\right)^{\dagger} \\
&= \left(\boldsymbol{P}_{\mathcal{N}^{\perp}(\boldsymbol{H})} \otimes \boldsymbol{P}_{\mathcal{N}^{\perp}(\boldsymbol{H})}\left(\boldsymbol{P}_{\mathcal{N}(\boldsymbol{H})} \otimes \boldsymbol{P}_{\mathcal{N}(\boldsymbol{H})} + \boldsymbol{P}_{\mathcal{N}^{\perp}(\boldsymbol{H})} \otimes \boldsymbol{P}_{\mathcal{N}^{\perp}(\boldsymbol{H})}\right)\right)^{\dagger} \\
&= \boldsymbol{P}_{\mathcal{N}^{\perp}(\boldsymbol{H})} \otimes \boldsymbol{P}_{\mathcal{N}^{\perp}(\boldsymbol{H})} \\
&= \boldsymbol{P}_{\mathcal{N}^{\perp}(\boldsymbol{D})}, \\
\boldsymbol{O}_2 &= \left(\boldsymbol{P}_{\mathcal{R}^{\perp}(\boldsymbol{X}_2)}\boldsymbol{P}_{\mathcal{R}(\boldsymbol{Y}_2)}\right)^{\dagger} \\
&= \left(\left(\boldsymbol{P}_{\mathcal{N}(\boldsymbol{H})} \otimes \boldsymbol{P}_{\mathcal{N}(\boldsymbol{H})} + \boldsymbol{P}_{\mathcal{N}^{\perp}(\boldsymbol{H})} \otimes \boldsymbol{P}_{\mathcal{N}^{\perp}(\boldsymbol{H})}\right) \boldsymbol{P}_{\mathcal{N}^{\perp}(\boldsymbol{H})} \otimes \boldsymbol{P}_{\mathcal{N}^{\perp}(\boldsymbol{H})}\right)^{\dagger} \\
&= \boldsymbol{P}_{\mathcal{N}^{\perp}(\boldsymbol{H})} \otimes \boldsymbol{P}_{\mathcal{N}^{\perp}(\boldsymbol{H})} = \boldsymbol{P}_{\mathcal{N}^{\perp}(\boldsymbol{D})}. \quad (196)
\end{aligned}
$$

Moreover, since $\mathcal{R}(\boldsymbol{X}_2) = \mathcal{R}(\boldsymbol{P}_{\mathcal{N}(\boldsymbol{H})} \otimes \boldsymbol{P}_{\mathcal{N}^{\perp}(\boldsymbol{H})} + \boldsymbol{P}_{\mathcal{N}^{\perp}(\boldsymbol{H})} \otimes \boldsymbol{P}_{\mathcal{N}(\boldsymbol{H})})$ and $\mathcal{R}^{\perp}(\boldsymbol{Y}_2) = \mathcal{R}(\boldsymbol{P}_{\mathcal{N}(\boldsymbol{H})} \otimes \boldsymbol{P}_{\mathcal{N}^{\perp}(\boldsymbol{H})} + \boldsymbol{P}_{\mathcal{N}^{\perp}(\boldsymbol{H})} \otimes \boldsymbol{P}_{\mathcal{N}(\boldsymbol{H})} + \boldsymbol{P}_{\mathcal{N}^{\perp}(\boldsymbol{H})} \otimes \boldsymbol{P}_{\mathcal{N}^{\perp}(\boldsymbol{H})})$ we have that $\mathcal{R}(\boldsymbol{X}_2) \subseteq \mathcal{R}^{\perp}(\boldsymbol{Y}_2) = \mathcal{N}(\boldsymbol{D})$. Therefore,

$$
(\boldsymbol{I}-\boldsymbol{L}_2)\boldsymbol{X}_2^{\dagger}(\boldsymbol{I}-\boldsymbol{O}_2) = \left(\boldsymbol{I} - \boldsymbol{P}_{\mathcal{N}^{\perp}(\boldsymbol{D})}\right) \boldsymbol{X}_2^{\dagger} \left(\boldsymbol{I} - \boldsymbol{P}_{\mathcal{N}^{\perp}(\boldsymbol{D})}\right) = \boldsymbol{P}_{\mathcal{N}(\boldsymbol{D})}\boldsymbol{X}_2^{\dagger}\boldsymbol{P}_{\mathcal{N}(\boldsymbol{D})} = \boldsymbol{X}_2^{\dagger}. \quad (197)
$$

Therefore, applying Thm. 7 we get

$$
\begin{aligned}
\left(2\eta\boldsymbol{P}_{\mathcal{N}(\boldsymbol{D})}\boldsymbol{C}\right)^{\dagger} &+ \boldsymbol{P}_{\mathcal{N}^{\perp}(\boldsymbol{D})}\left(\boldsymbol{P}_{\mathcal{N}^{\perp}(\boldsymbol{D})}(\boldsymbol{I}-\boldsymbol{Q})\right)^{\dagger} \boldsymbol{P}_{\mathcal{N}^{\perp}(\boldsymbol{D})} \\
&= \boldsymbol{X}_2^{\dagger} + \boldsymbol{L}_2\boldsymbol{Y}_2^{\dagger}\boldsymbol{O}_2 \\
&= (\boldsymbol{I}-\boldsymbol{L}_2)\boldsymbol{X}_2^{\dagger}(\boldsymbol{I}-\boldsymbol{O}_2) + \boldsymbol{L}_2\boldsymbol{Y}_2^{\dagger}\boldsymbol{O}_2 \\
&= (\boldsymbol{X}_2 + \boldsymbol{Y}_2)^{\dagger} \\
&= \left(2\eta\boldsymbol{P}_{\mathcal{N}(\boldsymbol{D})}\boldsymbol{C} + \boldsymbol{P}_{\mathcal{N}^{\perp}(\boldsymbol{D})}(\boldsymbol{I}-\boldsymbol{Q})\right)^{\dagger} \\
&= \left(2\eta\boldsymbol{P}_{\mathcal{N}(\boldsymbol{D})}\boldsymbol{C} + 2\eta\boldsymbol{P}_{\mathcal{N}^{\perp}(\boldsymbol{D})}\boldsymbol{C} - \eta^2\boldsymbol{P}_{\mathcal{N}^{\perp}(\boldsymbol{D})}\boldsymbol{D}\right)^{\dagger} \\
&= \left(2\eta\boldsymbol{C} - \eta^2\boldsymbol{D}\right)^{\dagger}. \quad (198)
\end{aligned}
$$

where in the second step we used (197), and in the third step we used Thm. 7. Overall, together with (63) we get

$$
\begin{aligned}
\lim_{t \to \infty} \mathrm{vec}\left(\boldsymbol{\Sigma}_t^{\perp}\right) &= \left(2\eta \boldsymbol{C} - \eta^2 \boldsymbol{D}\right)^{\dagger} \mathrm{vec}\left(\boldsymbol{\Sigma}_{\boldsymbol{v}}^{\perp}\right) \\
&= \left(\frac{1}{\eta}\left(2\boldsymbol{C} - \eta \boldsymbol{D}\right)^{\dagger}\right)\left(\eta^2 p\, \mathrm{vec}\left(\boldsymbol{\Sigma}_{\boldsymbol{g}}^{\perp}\right)\right) \\
&= \eta p \left(2\boldsymbol{C} - \eta \boldsymbol{D}\right)^{\dagger} \mathrm{vec}\left(\boldsymbol{\Sigma}_{\boldsymbol{g}}^{\perp}\right).
\end{aligned}
\tag{199}
$$

## H    PROOF OF COROLLARY 1

From Thm. 5 we have that if $0 < \eta < \eta_{\mathrm{var}}^*$ then

$$
\lim_{t \to \infty} \mathrm{vec}\left(\boldsymbol{\Sigma}_t^{\perp}\right) = \eta p \left(2\boldsymbol{C} - \eta \boldsymbol{D}\right)^{\dagger} \mathrm{vec}\left(\boldsymbol{\Sigma}_{\boldsymbol{g}}^{\perp}\right),
\tag{200}
$$

Using this result, we prove Corollary 1.

**First statement.**    If $0 < \eta < \eta_{\mathrm{var}}^*$ then by Prop. 5

$$
\begin{aligned}
\lim_{t \to \infty} \mathbb{E}[\|\boldsymbol{\theta}_t^{\perp} - \boldsymbol{\theta}^{*\perp}\|^2] &= \left(\mathrm{vec}\left(\boldsymbol{I}\right)\right)^{\mathrm{T}} \lim_{t \to \infty} \mathrm{vec}\left(\boldsymbol{\Sigma}_t^{\perp}\right) \\
&= \left(\mathrm{vec}\left(\boldsymbol{I}\right)\right)^{\mathrm{T}} \left(\eta p \left(2\boldsymbol{C} - \eta \boldsymbol{D}\right)^{\dagger} \mathrm{vec}\left(\boldsymbol{\Sigma}_{\boldsymbol{g}}^{\perp}\right)\right) \\
&= \eta p \left(\mathrm{vec}\left(\boldsymbol{I}\right)\right)^{\mathrm{T}} \left(2\boldsymbol{C} - \eta \boldsymbol{D}\right)^{\dagger} \mathrm{vec}\left(\boldsymbol{\Sigma}_{\boldsymbol{g}}^{\perp}\right).
\end{aligned}
\tag{201}
$$

**Second statement.**    Similarly, let us compute the limit of the expected value of the loss function to obtain point 2.

$$
\begin{aligned}
\lim_{t \to \infty} \mathbb{E}\left[\tilde{\mathcal{L}}(\boldsymbol{\theta}_t)\right] - \mathcal{L}(\boldsymbol{\theta}^*) &= \frac{1}{2} \lim_{t \to \infty} \mathbb{E}\left[(\boldsymbol{\theta}_t - \boldsymbol{\theta}^*)^{\mathrm{T}} \boldsymbol{H} (\boldsymbol{\theta}_t - \boldsymbol{\theta}^*)\right] \\
&= \frac{1}{2} \lim_{t \to \infty} \mathbb{E}\left[(\boldsymbol{\theta}_t - \boldsymbol{\theta}^*)^{\mathrm{T}} \boldsymbol{P}_{\mathcal{N}^{\perp}(\boldsymbol{H})} \boldsymbol{H} \boldsymbol{P}_{\mathcal{N}^{\perp}(\boldsymbol{H})} (\boldsymbol{\theta}_t - \boldsymbol{\theta}^*)\right] \\
&= \frac{1}{2} \lim_{t \to \infty} \mathbb{E}\left[(\boldsymbol{\theta}_t^{\perp} - \boldsymbol{\theta}^{*\perp})^{\mathrm{T}} \boldsymbol{H} (\boldsymbol{\theta}_t^{\perp} - \boldsymbol{\theta}^{*\perp})\right] \\
&= \frac{1}{2} \mathrm{Tr}\left(\boldsymbol{H} \lim_{t \to \infty} \mathbb{E}\left[(\boldsymbol{\theta}_t^{\perp} - \boldsymbol{\theta}^{*\perp})(\boldsymbol{\theta}_t^{\perp} - \boldsymbol{\theta}^{*\perp})^{\mathrm{T}}\right]\right) \\
&= \frac{1}{2} \mathrm{Tr}\left(\boldsymbol{H} \lim_{t \to \infty} \boldsymbol{\Sigma}_t^{\perp}\right) \\
&= \frac{1}{2} \left(\mathrm{vec}\left(\boldsymbol{H}\right)\right)^{\mathrm{T}} \lim_{t \to \infty} \mathrm{vec}\left(\boldsymbol{\Sigma}_t^{\perp}\right) \\
&= \frac{1}{2} \left(\mathrm{vec}\left(\boldsymbol{H}\right)\right)^{\mathrm{T}} \left(\eta p \left(2\boldsymbol{C} - \eta \boldsymbol{D}\right)^{\dagger} \mathrm{vec}\left(\boldsymbol{\Sigma}_{\boldsymbol{g}}^{\perp}\right)\right) \\
&= \frac{1}{2} \eta p \left(\mathrm{vec}\left(\boldsymbol{H}\right)\right)^{\mathrm{T}} \left(2\boldsymbol{C} - \eta \boldsymbol{D}\right)^{\dagger} \mathrm{vec}\left(\boldsymbol{\Sigma}_{\boldsymbol{g}}^{\perp}\right).
\end{aligned}
\tag{202}
$$

**Third statement.**    Finally, we prove point 3. The gradient of the second-order Taylor expansion of the loss is given by

$$
\nabla \tilde{\mathcal{L}}(\boldsymbol{\theta}) = \boldsymbol{H}\left(\boldsymbol{\theta} - \boldsymbol{\theta}^*\right).
\tag{203}
$$

Therefore

$$
\begin{aligned}
\lim_{t\to\infty} \mathbb{E}\left[\left\|\nabla\tilde{\mathcal{L}}(\boldsymbol{\theta}_t)\right\|^2\right] &= \lim_{t\to\infty} \mathbb{E}\left[(\boldsymbol{\theta}_t - \boldsymbol{\theta}^*)^{\mathrm{T}}\boldsymbol{H}^2(\boldsymbol{\theta}_t - \boldsymbol{\theta}^*)\right] \\
&= \lim_{t\to\infty} \mathbb{E}\left[(\boldsymbol{\theta}_t - \boldsymbol{\theta}^*)^{\mathrm{T}}\boldsymbol{P}_{\mathcal{N}^\perp(\boldsymbol{H})}\boldsymbol{H}^2\boldsymbol{P}_{\mathcal{N}^\perp(\boldsymbol{H})}(\boldsymbol{\theta}_t - \boldsymbol{\theta}^*)\right] \\
&= \lim_{t\to\infty} \mathbb{E}\left[(\boldsymbol{\theta}_t^\perp - \boldsymbol{\theta}^{*\perp})^{\mathrm{T}}\boldsymbol{H}^2(\boldsymbol{\theta}_t^\perp - \boldsymbol{\theta}^{*\perp})\right] \\
&= \mathrm{Tr}\left(\boldsymbol{H}^2 \lim_{t\to\infty} \mathbb{E}\left[(\boldsymbol{\theta}_t^\perp - \boldsymbol{\theta}^{*\perp})(\boldsymbol{\theta}_t^\perp - \boldsymbol{\theta}^{*\perp})^{\mathrm{T}}\right]\right) \\
&= \mathrm{Tr}\left(\boldsymbol{H}^2 \lim_{t\to\infty} \boldsymbol{\Sigma}_t^\perp\right) \\
&= \left(\mathrm{vec}\left(\boldsymbol{H}^2\right)\right)^{\mathrm{T}} \lim_{t\to\infty} \mathrm{vec}\left(\boldsymbol{\Sigma}_t^\perp\right) \\
&= \left(\mathrm{vec}\left(\boldsymbol{H}^2\right)\right)^{\mathrm{T}}\left(\eta p\left(2\boldsymbol{C} - \eta\boldsymbol{D}\right)^\dagger \mathrm{vec}\left(\boldsymbol{\Sigma}_{\boldsymbol{g}}^\perp\right)\right) \\
&= \eta p(\mathrm{vec}\left(\boldsymbol{H}^2\right))^{\mathrm{T}}\left(2\boldsymbol{C} - \eta\boldsymbol{D}\right)^\dagger \mathrm{vec}\left(\boldsymbol{\Sigma}_{\boldsymbol{g}}^\perp\right).
\end{aligned} \tag{204}
$$

## I  RECOVERING GD'S STABILITY CONDITION

In this section we show how our stability condition for SGD reduces to GD's when $B = n$. In this case $p = 0$ and thus

$$
\boldsymbol{C} = \frac{1}{2}\boldsymbol{H}\oplus\boldsymbol{H}, \qquad \boldsymbol{D} = \boldsymbol{H}\otimes\boldsymbol{H}. \tag{205}
$$

For simplicity, here we assume that $\boldsymbol{H}$ has full rank, where the case that $\boldsymbol{H}$ has null space requires minor adjustments. Let $\boldsymbol{H} = \boldsymbol{V}\boldsymbol{\Lambda}\boldsymbol{V}^{\mathrm{T}}$ be the eigenvalue decomposition of $\boldsymbol{H}$, where $\boldsymbol{V}\boldsymbol{V}^{\mathrm{T}} = \boldsymbol{V}^{\mathrm{T}}\boldsymbol{V} = \boldsymbol{I}$, then

$$
\begin{aligned}
\boldsymbol{C} &= \frac{1}{2}\boldsymbol{H}\oplus\boldsymbol{H} \\
&= \frac{1}{2}\left(\boldsymbol{H}\otimes\boldsymbol{I} + \boldsymbol{H}\otimes\boldsymbol{I}\right) \\
&= \frac{1}{2}\left(\left(\boldsymbol{V}\boldsymbol{\Lambda}\boldsymbol{V}^{\mathrm{T}}\right)\otimes\left(\boldsymbol{V}\boldsymbol{V}^{\mathrm{T}}\right) + \left(\boldsymbol{V}\boldsymbol{V}^{\mathrm{T}}\right)\otimes\left(\boldsymbol{V}\boldsymbol{\Lambda}\boldsymbol{V}^{\mathrm{T}}\right)\right) \\
&= \frac{1}{2}\left((\boldsymbol{V}\otimes\boldsymbol{V})(\boldsymbol{\Lambda}\otimes\boldsymbol{I})\left(\boldsymbol{V}^{\mathrm{T}}\otimes\boldsymbol{V}^{\mathrm{T}}\right) + (\boldsymbol{V}\otimes\boldsymbol{V})(\boldsymbol{I}\otimes\boldsymbol{\Lambda})\left(\boldsymbol{V}^{\mathrm{T}}\otimes\boldsymbol{V}^{\mathrm{T}}\right)\right) \\
&= (\boldsymbol{V}\otimes\boldsymbol{V})\left(\frac{1}{2}\boldsymbol{\Lambda}\otimes\boldsymbol{I} + \frac{1}{2}\boldsymbol{I}\otimes\boldsymbol{\Lambda}\right)(\boldsymbol{V}\otimes\boldsymbol{V})^{\mathrm{T}}.
\end{aligned} \tag{206}
$$

Note that

$$
(\boldsymbol{V}\otimes\boldsymbol{V})^{\mathrm{T}}(\boldsymbol{V}\otimes\boldsymbol{V}) = \left(\boldsymbol{V}^{\mathrm{T}}\otimes\boldsymbol{V}^{\mathrm{T}}\right)(\boldsymbol{V}\otimes\boldsymbol{V}) = \left(\boldsymbol{V}^{\mathrm{T}}\boldsymbol{V}\right)\otimes\left(\boldsymbol{V}^{\mathrm{T}}\boldsymbol{V}\right) = \boldsymbol{I}\otimes\boldsymbol{I} = \boldsymbol{I}, \tag{207}
$$

*i.e.,* $(\boldsymbol{V}\otimes\boldsymbol{V})$ is an orthogonal matrix. Since $\frac{1}{2}(\boldsymbol{\Lambda}\otimes\boldsymbol{I} + \boldsymbol{I}\otimes\boldsymbol{\Lambda})$ is diagonal, then the last result in (206) is an eigenvalue decomposition of $\boldsymbol{C}$. Similarly,

$$
\begin{aligned}
\boldsymbol{D} &= \boldsymbol{H}\otimes\boldsymbol{H} \\
&= \left(\boldsymbol{V}\boldsymbol{\Lambda}\boldsymbol{V}^{\mathrm{T}}\right)\otimes\left(\boldsymbol{V}\boldsymbol{\Lambda}\boldsymbol{V}^{\mathrm{T}}\right) \\
&= (\boldsymbol{V}\otimes\boldsymbol{V})(\boldsymbol{\Lambda}\otimes\boldsymbol{\Lambda})\left(\boldsymbol{V}^{\mathrm{T}}\otimes\boldsymbol{V}^{\mathrm{T}}\right) \\
&= (\boldsymbol{V}\otimes\boldsymbol{V})(\boldsymbol{\Lambda}\otimes\boldsymbol{\Lambda})(\boldsymbol{V}\otimes\boldsymbol{V})^{\mathrm{T}}.
\end{aligned} \tag{208}
$$

We have that $\boldsymbol{C}$ and $\boldsymbol{D}$ have the same eigenvectors. Thus, set $\lambda_\ell = \boldsymbol{\Lambda}_{[\ell,\ell]} = \lambda_\ell(\boldsymbol{H})$, and define

$$
f(x) = \begin{cases} \frac{1}{x}, & x \neq 0, \\ 0, & x = 0. \end{cases} \tag{209}
$$

Then

$$\lambda_{\max}\left(\boldsymbol{C}^{\dagger}\boldsymbol{D}\right) = \max_{\ell,p\in[d]}\left\{\lambda_{\ell}\lambda_{p}f\left(\frac{1}{2}(\lambda_{\ell}+\lambda_{p})\right)\right\}. \tag{210}$$

Note that the objective vanishes whenever $\lambda_{\ell} = 0$ or $\lambda_{p} = 0$. Restricting to only positive eigenvalues gives

$$\lambda_{\max}\left(\boldsymbol{C}^{\dagger}\boldsymbol{D}\right) = \max_{\lambda_{\ell},\lambda_{p}>0}\left\{\frac{\lambda_{\ell}\lambda_{p}}{\frac{1}{2}(\lambda_{\ell}+\lambda_{p})}\right\}. \tag{211}$$

Additionally $\sqrt{\lambda_{\ell}\lambda_{p}} \leq \frac{1}{2}(\lambda_{\ell}+\lambda_{p})$ holds for all $\lambda_{\ell},\lambda_{p} > 0$, therefore

$$\begin{aligned}
\frac{\lambda_{\ell}\lambda_{p}}{\frac{1}{2}(\lambda_{\ell}+\lambda_{p})} &= \sqrt{\lambda_{\ell}\lambda_{p}}\frac{\sqrt{\lambda_{\ell}\lambda_{p}}}{\frac{1}{2}(\lambda_{\ell}+\lambda_{p})} \\
&\leq \sqrt{\lambda_{\ell}\lambda_{p}} \\
&\leq \lambda_{\max}.
\end{aligned} \tag{212}$$

Yet, for $\lambda_{\ell} = \lambda_{p} = \lambda_{\max}$ we have that

$$\frac{\lambda_{\ell}\lambda_{p}}{\frac{1}{2}(\lambda_{\ell}+\lambda_{p})} = \lambda_{\max}. \tag{213}$$

Hence we have

$$\lambda_{\max}\left(\boldsymbol{C}^{\dagger}\boldsymbol{D}\right) = \max_{\lambda_{\ell},\lambda_{p}>0}\left\{\frac{\lambda_{\ell}\lambda_{p}}{\frac{1}{2}(\lambda_{\ell}+\lambda_{p})}\right\} = \lambda_{\max}(\boldsymbol{H}). \tag{214}$$

## J    ADDITIONAL EXPERIMENTAL RESULTS AND DETAIL

In this section, we complete the technical detail of the experiment shown in Sec. 4. For the experiment, we used a single-hidden layer ReLU network with fully connected layers (with bias vectors). The number of neurons is 1024, and the total number of parameters is 807,940. We used four classes from MNIST, 256 samples from each class, with a total of 1024 samples. To get large initialization, we used standard torch initialization and multiplied the initial weights by a factor of 15. The maximal number of epochs was set to $4 \times 10^4$. If SGD did not converge within this number of epochs, then we removed this run from the plots.

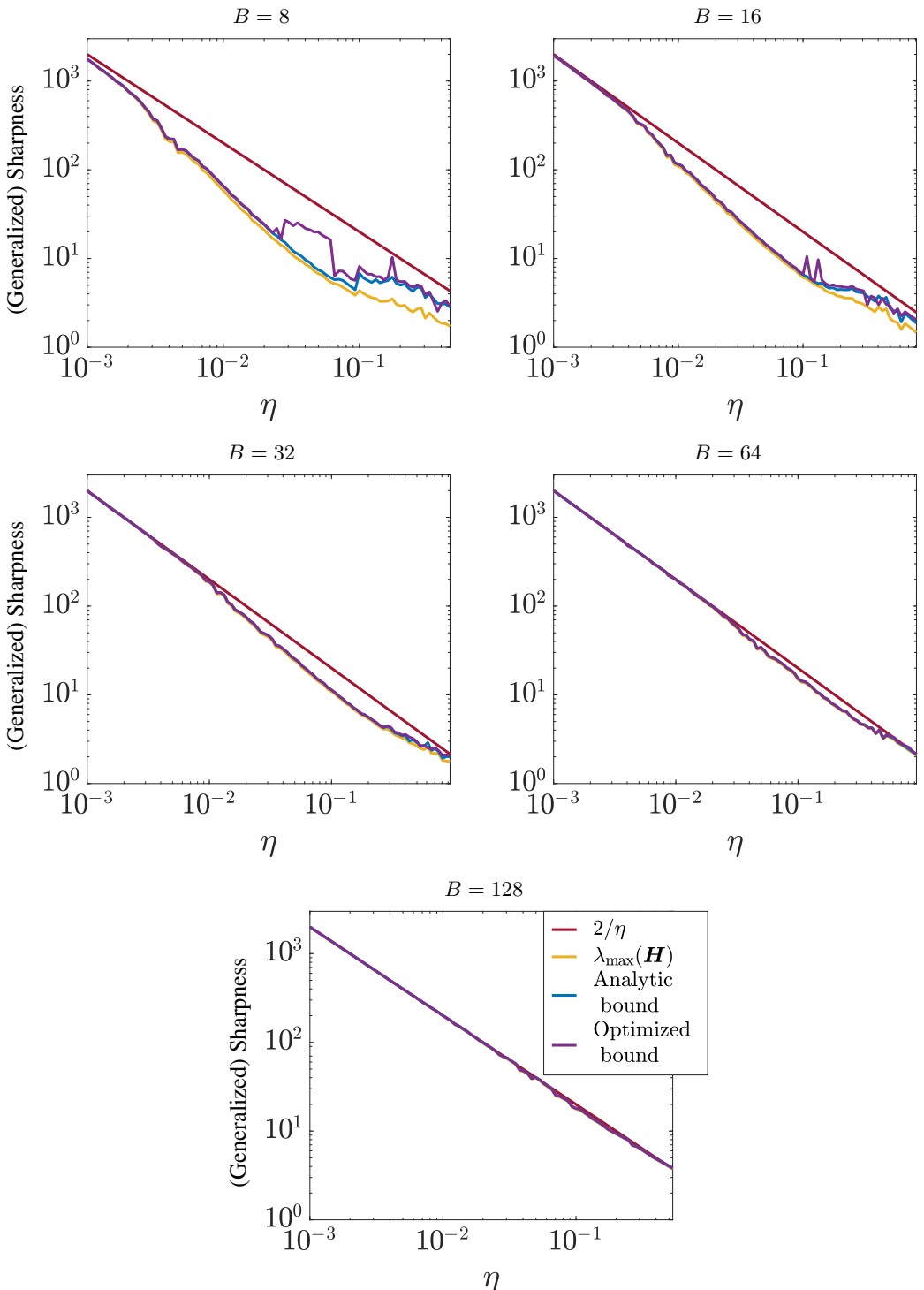

Figure 3: **Sharpness vs. learning rate.** Additional results for the experiment in Sec. 4. These five figures complete the results of Fig. 1. Here we see that SGD with big batch sizes behaves like GD.

