# OpenReview forum: "Exact Mean Square Linear Stability Analysis for SGD"
_ICLR.cc/2024/Conference — Submitted to ICLR 2024_

### Official Review · Reviewer_YQoF · 2023-10-24

**Soundness:** 3 good
**Presentation:** 3 good
**Contribution:** 3 good
**Rating:** 6
**Confidence:** 4

**Summary:**

This manuscript provides a precise condition for the mean square stability of SGD around a minimum, considering both interpolation and non-interpolation cases. The authors also discuss the implications for the influence of batch size and provide numerical experiments to support their theoretical findings.

**Strengths:**

- The precise condition presented, particularly the explanation provided in Proposition 2, is interesting and greatly intuitive. Proposition 2 is especially well-explained and appreciated by the reviewer.

- The discussion on how batch size affects stability and the finding that the maximum eigenvalue of the Hessian is close to that of GD are intriguing. The experimental validation, although limited to small nets for fitting MNIST, is solid.

- The experiments also suggest that SGD **may** operate on the edge of stability. This interesting observation deserves more investigations.

**Weaknesses:**

- The extension to non-interpolation minima is uninteresting and does not offer any new insights, as far as the reviewer can tell. Additionally, the definition of regular minima is peculiar, as it requires all sample Hessian matrices to be SPD. It is unclear why such a strange assumption is relevant.

- The experiments are limited to a very smal-scale setup: one-hidden-layer nets + subset of MNIST.

**Questions:**

None

---

> ### Author Response · Authors · 2023-11-15
> **Response to Reviews**
>
> Thanks for the feedback.
>
> &nbsp;
>
> **``The extension to non-interpolation minima is uninteresting and does not offer any new insights’’** \
> Although the stability threshold is the same in both cases, our theory reveals that the dynamics is significantly different. In interpolating minima, under the stability condition, SGD converges (namely reaches zero loss). This is in contrast to non-interpolating minima, where SGD randomly wanders around the minimum. Here we give a detailed characterization for its dynamics. Specifically, we derive a closed-form expression for the covariance matrix of the dynamics in the orthogonal complement of the null space of the Hessian for long training times (Theorem 5). As stated in the paper, from this we learn that when decreasing the learning rate, the loss level drops, and the dynamics gets closer to the minimum (Corollary 1). This explains the empirical behavior observed when decreasing the learning rate in neural network training,  which causes the loss level to drop. In the null space, we show that the dynamics perform a random walk (Theorem 4). This means that SGD constantly wanders in the loss valley formed by connected minima. These insights are specific to non-interpolating minima.

---

### Official Review · Reviewer_Dxwn · 2023-10-31

**Soundness:** 3 good
**Presentation:** 3 good
**Contribution:** 2 fair
**Rating:** 6
**Confidence:** 3

**Summary:**

This paper finds the explicit expression of the linear-stability threshold learning rate for SGD. This paper improves the existing result by deriving an explicit expression of the threshold learning rate from the implicit one. The new expression motivates linking the stability of the finite batch size SGD to a process of mixed online SGD and GD. Simpler bounds for the threshold learning rate are also found. Numerical experiments using MNIST confirm the analytical results.

**Strengths:**

This paper finds an explicit expression of the threshold learning rate and simplifies its computation by turning an optimization problem into an eigen-value problem, which I believe is a nice improvement over the existing results.  Also, this paper provides the insight that, in terms of linear stability, the SGD could be viewed as a mixture of GD steps and mini-batch SGD steps. The experiment results are well presented with highly informative figures.

**Weaknesses:**

Although explicit expression is always welcomed, the sufficient and necessary condition for linear stability involving learning rate, Hessian, and batch size already exists in previous works. So, this paper studies a well-understood problem. The writing could be improved. Inserting one subsection between theorem 3 and its proof may not be the best way of presenting them. Also, the notation $\theta^\parallel$, $\theta^\perp$, and $vec$ are not introduced in the main text.

**Questions:**

* (I didn’t check all the mathematical details so please excuse me if what I ask is already written in the paper.) I don’t see immediately why ${\bf C}$ is guaranteed to be PSD. Is there any relevant lemma in the paper? Or could the authors provide some intuition.
* It appears to me that getting relation (83) requires that ${\bf u^T Du}$ is positive. Is ${\bf D}$ also PSD?
* The authors mentioned analyzing the stability of SGD from a probabilistic point of view. I wonder if the quantitative results in the two papers can be compared.

---

> ### Author Response · Authors · 2023-11-15
> **Response to Reviews**
>
> Thanks for the feedback.
>
> &nbsp;
>
> **``The sufficient and necessary condition for linear stability involving learning rate, Hessian, and batch size already exists in previous works’’** \
> Many papers studied linear stability in the past. Most of these papers give conditions which are either only sufficient or only necessary, but not both. To the best of our knowledge, in the general case, there are exact explicit conditions only for the *one dimensional* case. Specifically, [R4] gave the exact condition for linear stability in the mean square sense (which our result generalizes to multiple dimensions), while [R1] gave the exact condition for linear stability in probability. Other exact conditions for stability in the multidimensional case are implicit. These include Theorem 1 in [R5] for moments, and Theorem 2 in [R1] for the probabilistic setting. Since these results are implicit, it is hard to draw from them conclusions on the stability of SGD w.r.t. the learning rate and batch size. In other words, although there are multiple papers on this subject, it is not a well-understood problem. The significance of our explicit expression can be seen by the subsequent results that we derived from it, each important in its own right: monotonicity of the stability threshold w.r.t. the batch size (Prop. 1), the analogous algorithm (Prop. 2), and two simple necessary conditions (Prop. 3). We would not have gotten to these results without our explicit condition.
>
> [R1] Ziyin, Liu, et al. "The probabilistic stability of stochastic gradient descent." arXiv preprint. \
> [R4] Wu, Lei, and Chao Ma. "How SGD selects the global minima in over-parameterized learning: A dynamical stability perspective." NeurIPS 2018. \
> [R5] Ma, Chao, and Lexing Ying. "On linear stability of sgd and input-smoothness of neural networks." NeurIPS 2021.
>
> **Why is $ \boldsymbol{C} $ guaranteed to be PSD?** \
> The Kronecker sum has the property that its eigenvalues are the pairwise sums of the spectrums of the summands [R6, Thm. 13.16]. In App. I, we derive this explicitly for $ \\boldsymbol{C} $ (see Eq. (206)). Specifically, we show that the eigenvalues of $ \\boldsymbol{C} = \\frac{1}{2} \\boldsymbol{H} \\oplus \\boldsymbol{H}  $ are $ \\frac{1}{2} \\big( \\lambda\_i(\\boldsymbol{H}) + \\lambda\_j(\\boldsymbol{H})  \\big) , \\  i=1,\\ldots,d ,\\, j=1,\\ldots,d $.  Recall that $ \\boldsymbol{H} $ is the Hessian of the loss at a minimum, and is therefore PSD. This means that all eigenvalues of $ \\boldsymbol{H} $ are nonnegative, and as a consequence the eigenvalues of $ \\boldsymbol{C} $ are nonnegative. Overall, we have that $ \\boldsymbol{C} $ is PSD. We’ll mention this in the main text, thanks.
>
> [R6] Laub, Alan J. “Matrix analysis for scientists and engineers.” Society for Industrial and Applied Mathematics, 2004.

---

> ### Author Response · Authors · 2023-11-15
> **Response to Reviews**
>
> **Is $\\boldsymbol{D}$ also PSD?** \
> Thank you for the careful reading of our proofs. The short answer is yes, in our setting $\\boldsymbol{D}$ is PSD. We will add this statement in relation to Eq. (83), together with the explanation below, thanks!
>
> The reason that $\boldsymbol{D}$ is PSD in our setting is rooted in the following property of the Kronecker product [R6, Thm. 13.12]: \
> Suppose that $\\boldsymbol{A}$ and $\\boldsymbol{B}$ are square matrices of size $n$ and $ m $ respectively. Let $\\lambda_1, \\ldots, \\lambda_n$ be the eigenvalues of $\\boldsymbol{A}$ and $ \\mu\_1, \\ldots, \\mu_m $ be those of $\\boldsymbol{B}$ (listed according to multiplicity). Then the eigenvalues of $\\boldsymbol{A}\\otimes\\boldsymbol{B}$ are $ \\lambda\_i \\mu_j, \\  i=1,\\ldots,n , \\, j=1,\\ldots,m$.
>
> This property asserts that for any PSD matrix $\\boldsymbol{A}$, the Kronecker product $\\boldsymbol{A}\\otimes\\boldsymbol{A}$ is PSD.
>
> Now, $ \\boldsymbol{D} $ is defined as
> $$\\boldsymbol{D}\\triangleq(1-p)\\boldsymbol{H}\\otimes\\boldsymbol{H}+p\\frac{1}{n}\\sum_{i=1}^n\\boldsymbol{H}\_i\\otimes\\boldsymbol{H}\_i.$$
> In our settings, *i.e.* regular and interpolating minima, we consider Hessian matrices $\\{\\boldsymbol{H}\_i\\} $ that are PSD. By the property above, all $\\{ \\boldsymbol{H}\_i\\otimes\\boldsymbol{H}\_i\\} $ are PSD, and also $ \\{\\boldsymbol{H}\\otimes \\boldsymbol{H}\\}$ is PSD. Therefore, $ \\boldsymbol{D} $ is a convex combination of PSD matrices, which is PSD.
>
> We point out that in the general case, *i.e.* for minima that are not regular, $ \\boldsymbol{D} $ may be not PSD. Here, one can work around this issue by the following strategy. Note that Eq. (82) boils down to the condition $ \\eta \\boldsymbol{u}^T \\boldsymbol{D} \\boldsymbol{u}\\leq2\\boldsymbol{u}^T\\boldsymbol{C}\\boldsymbol{u}$. As a consequence, whenever $\\boldsymbol{u}^T \\boldsymbol{D}\\boldsymbol{u}$ is negative, the stability condition is satisfied, since $\\boldsymbol{C}$ is PSD. Therefore, in terms of stability, we can consider only the set of $\\boldsymbol{u}$ for which $\\boldsymbol{u}^T\\boldsymbol{D}\\boldsymbol{u}>0$. This will change the constraint in the infimum of Eq. (83), and may add conditions downstream in the derivation. This is out of the scope for this paper, and we leave this to future study.
>
> [R6] Laub, Alan J. “Matrix analysis for scientists and engineers.” Society for Industrial and Applied Mathematics, 2004.
>
> **Comparing mean square and probabilistic analysis** \
> As mentioned in the main text, probabilistic stability is a relaxed notion of stability whereas mean square stability is a stricter notion. Specifically, every minimum that is stable in the mean square sense is also stable in probability, but not vice versa. In general, probabilistic stability allows the set of stable learning rates to be a collection of disjoint intervals [R1]. This is in sharp contrast to mean square, where here we show that the set of stable learning rates is in fact a single interval. (This is an additional contribution of our paper based on the explicit expression, which cannot be deduced trivially from the implicit condition). Moreover, probabilistic stability predicts that SGD can converge to minima with sharpness far beyond GD’s threshold [R1]. Here we note that this was not observed in any of the extensive experiments done in [R2] and [R3, App. G].
>
> Currently, there is a debate about which notion of stability best describes what is happening in real world applications. Quantitatively comparing these two notions in real world tasks is challenging, and deserves a separate work. The main difficulty lies in the exact condition for stability in probability [Thm. 2, R1], which is formulated as an optimization problem. In the general case, there is no effective way to solve this problem. Moreover, given a batch size, evaluating the probabilistic condition of stability for a single step size doesn’t reveal the set of stable learning rates, let alone its dependence on the batch size. For this, one should evaluate the condition numerous times which becomes computationally impractical. Therefore, overall, our work lays the foundation for future comparisons, but such comparisons are currently impossible for us to perform.
>
> [R1] Ziyin, Liu, et al. "The probabilistic stability of stochastic gradient descent." arXiv preprint. \
> [R2] Gilmer, Justin, et al. "A loss curvature perspective on training instabilities of deep learning models." ICLR 2021. \
> [R3] Cohen, Jeremy, et al. "Gradient Descent on Neural Networks Typically Occurs at the Edge of Stability." ICLR 2020.
>
> **The notations $ \\boldsymbol{\\theta}^{\\|} $ and $ \\boldsymbol{\\theta}^{\\perp} $** \
> The notation $ \\boldsymbol{\\theta}^{\\|} $ and $ \\boldsymbol{\\theta}^{\\perp} $ are introduced in the first paragraph of Sec. 3. However, for clarity we will repeat their meaning in more places within the text. In any case, they also appear in the table of notations in App. A.

---

### Official Review · Reviewer_GWmw · 2023-11-01

**Soundness:** 3 good
**Presentation:** 2 fair
**Contribution:** 3 good
**Rating:** 5
**Confidence:** 2

**Summary:**

The authors present a stability threshold on the stochastic gradient descent. The proposed threshold is a monotonically non-decreasing function of the batch size. It differs from the existing thresholds since it is a closed-form formulation rather than another optimization problem.

**Strengths:**

- The problem that the authors consider is broader. Hessian is still assumed to be positive semi-definite. However, the individual gradients are allowed to be arbitrary if the mean of the gradients over all samples goes to zero.

- Both interpolating and non interpolating (regular) minima are discussed.

**Weaknesses:**

- Figure 2 could be drawn by using different seeds for SGD and could be represented by an error bar. It is not obvious how would these results change with a different initialization.

- The conclusion that authors mentioned in Figure 2 is not very obvious, how would the authors explain the fluctuation, how is the evaluation done when  the authors are claiming "optimized bound coincides"? "We see that for small batch sizes B = 1 and B = 2, the optimized bound (24) coincides with 2/η, confirming that SGD converged at the edge of stability".

Minor:

The paper writing needs further polishing.

-There are some repetitions. The rates of the single sample gradient step and full batch gradient step are mentioned in the abstract, beginning of page 2, in the exact same words.

- The word "dynamics" is used quite frequently before mentioning what authors refer to as dynamics before mentioning the analysis of SGD's dynamics.

**Questions:**

- Can authors elaborate more on the mean-square sense?

A few questions are mentioned in the weaknesses section.

**Details Of Ethics Concerns:**

This paper does not require ethics review.

---

> ### Author Response · Authors · 2023-11-15
> **Response to Reviews**
>
> Thanks for the feedback.
>
> &nbsp;
>
> **Reproduce Fig. 2 under multiple random seeds and with variation bars** \
> We will add such an analysis to the final version. Kindly note that in this experiment we trained 800 models, and analyzed their sharpness after training. This requires significant compute power and consumes a lot of time. We will therefore not be able to repeat this experiment for a very large number of seeds. We will repeat it for a few seeds. From our experimentation with a single pair of learning rate and batch size, we predict that the full analysis will show that the yellow and blue curves do not fluctuate much between different seeds, yet we might see some fluctuations in the purple curve (optimized bound).
>
> **Fluctuation of the optimized bound in Fig. 2 and its evaluation** \
> The purple curves depict the optimized bound (Eq. (24)). The value of this bound is obtained by an optimization problem which we solved using GD for each pair of step size and batch size. It may definitely be that we did not find the global optimum for every step size, and got stuck at local maxima for some set of hyperparameters. This is one possible explanation for why the curve falls down and then comes up again at some of the learning rates.
>
> Another possible explanation for this behavior is the following. As stated in the paper above Eq. (24), the optimized bound is equivalent to restricting the optimization problem from Eq. (11) to rank one symmetric matrices. It is possible that for some minima of the loss, the optimal matrix of Eq. (11) is rank one and therefore the bound is tight, while for others the rank is higher and thus the bound is not tight (i.e. lower). We will add this discussion to the paper, thanks.
>
> From Eq. (25) we see that when the optimized bound equals  $2/ \\eta $ so does the general sharpness. Therefore, at least in these intervals we can conclude that SGD converged at the edge of stability. We will rephrase the sentence to ``We see that for the small batch sizes $B = 1$ and $B = 2$, the optimized bound (24) coincides with $2/ \\eta $ over certain intervals, indicating that SGD converged at the edge of stability over those intervals’’.
>
> **``Can the authors elaborate more on the mean-square sense?’’** \
> The dynamics of SGD $ \\{ \\boldsymbol{\\theta}\_t \\}\_{t=1}^{\\infty} $, given in Eq. (2), is a random process. Therefore, it may be that for some realizations it converges to a minimum, for some realizations it does not converge but remains within a bounded set around the minimum, and for some realizations it diverges. So how can we study its properties? Let’s talk about convergence for simplicity. Two popular options for convergence analysis of random processes rely on moments and on probability. Specifically, in the former option one considers some moment of the distance to the limit, and wants it to tend to zero. In the latter option one wants the probability that this distance is smaller than any positive threshold to tend to one. Here we chose to use the second moment which is the square mean distance. In general, convergence in moments is stricter than in probability, where higher moments lead to more stricter conditions. In our case, this means that every minimum which is stable in the mean square sense is also stable in high probability, but not vice versa. Currently, there is a discussion in the community about which notion of stability best describes the phenomena observed in practice [R1]. Here, it is important to note two things:
> * Our paper provides the exact threshold for mean square stability, which can be used in future work for comparison with other notions of stability. Namely, our work is needed to determine which notion of stability best describes the behavior of SGD in training neural networks.
> * The relaxed notion of stability ``in probability’’ predicts that SGD can converge to minima with sharpness far beyond GD’s threshold [R1], yet this was not observed in any of the extensive experiments done in [R2] and [R3, App. G].
>
> [R1] Ziyin, Liu, et al. "The probabilistic stability of stochastic gradient descent." arXiv preprint. \
> [R2] Gilmer, Justin, et al. "A loss curvature perspective on training instabilities of deep learning models." ICLR 2021. \
> [R3] Cohen, Jeremy, et al. "Gradient Descent on Neural Networks Typically Occurs at the Edge of Stability." ICLR 2020.
>
>
> **Further polishing of the paper** \
> We thank the reviewer for the comments. We will revise the writing of the paper.

---

### Meta-Review · Area_Chair_tJJh · 2023-12-14

**Metareview:**

In this paper, authors study the exact stability threshold for stochastic gradient descent (SGD) and derive a closed form expression. Authors provide an explicit condition on the learning rate and bathc size and prove that the stability threshold is monotonically non-decreasing in batch size.
They study different batch size regimes and discuss their implications in terms of stability. There are also experiments provided on MNIST dataset.

This paper was reviewed by 3 reviewers and received the following Rating/Confidence scores: 6/4, 6/3, 5/2.

None of the reviewers championed the paper and they do have some valid concerns. Some of the concerns include: 1- Ambiguous conclusions. 2- Better motivating the paper. 3- Highlighting novelty and limitations. 4- Insufficient experiments.

AC thinks that the paper has potential but requires significant revision, recommending reject for this ICLR.

**Justification For Why Not Higher Score:**

The reviewers pointed out several weaknesses which I agree with.

**Justification For Why Not Lower Score:**

n/a

---

### Decision · Program_Chairs · 2024-01-16

Reject